# The H₂Ours game to explore Water Use, Resources and Sustainability: connecting issues in two landscapes in Indonesia

Lisa Tanika[1], Rika Ratna Sari[2], Arief Lukman Hakim[1], Meine van Noordwijk[3,4], Marielos Peña-Claros[1], Beria Leimona[3], Edi Purwanto[5], Erika N. Speelman[6]

[1]Forest Ecology and Forest Management Group, Wageningen University & Research, 6708 PB Wageningen, The Netherlands
[2]Soil science department, Faculty of Agriculture, Brawijaya University. Jl. Veteran 1, Malang, East Java Indonesia
[3]Plant Production Systems, Plant Science, Wageningen University & Research, P.O. Box 430, 6700 AK Wageningen, The Netherlands
[4]World Agroforestry, ICRAF Southeast Asia, Bogor 16001, Indonesia
[5]Tropenbos Indonesia, Bogor, Indonesia
[6]Geo-Information Science and Remote Sensing, Wageningen University & Research, 6708 PB Wageningen, The Netherlands

*Correspondence to*: Lisa Tanika (lisa.tanika@wur.nl; lisa.tanika@gmail.com)

**Abstract.** Restoring hydrological functions affected by economic development trajectories faces social and economic challenges. Given that stakeholders often only have a partial understanding of how the functioning socio-hydrological systems, it is expected that knowledge sharing will help them to become more aware of the consequences of their land use choices and options to manage water collectively. To facilitate the collective learning tools are needed that represent the essential social and technical aspects parts of a social-hydrological system in simple terms. However, data-driven simplification can lead to very site specific models that are difficult to adapt to different conditions. To address these issues, this study aims to develop a highly adaptable serious game based on process-based understanding to make it easily applicable to any situation and to facilitate co-learning among stakeholders regarding complex socio-hydrological problems. We developed and tested a 'serious' game that revolves around a simple water balance and economic accounting, with environmental and financial consequences for the land-users. The game is based on process-based understanding of the system, allowing for both relevant site-specificity and generic replicability. Here, we describe the development of the Water: Use, Resources and Sustainability ('H₂Ours') game and explore its capacity to visualise, discuss and explore issues at landscape level. The H₂Ours game was designed using a combination of the Actors, Resources, Dynamics and Interaction (ARDI) and the Drivers, Pressure, State, Impact, and Responses (DPSIR) frameworks. The design steps for constructing the game led to a generic version, and two localised versions for two different landscapes in Indonesia: a mountain slope to lowland paddy landscape impacting groundwater availability in East Java, and a peatland with drainage-rewetting, oil palm conversion and fire as issues triggering responses in West Kalimantan. Based on evaluation referring to credibility, salience and legitimacy criteria, the H₂Ours game met its purpose as a tool for knowledge transfer, learning and triggering action. We discuss the steps that can lead to re-designing and adaptation of the game to other landscapes and their policy-relevant issues.

# 1 Introduction

A recent call for collective action by the Global Commission on the Economics of Water (Mazzucato et al., 2023) asked for turning the tide, shifting from exploitation, over-use and wastage of freshwater resources to stewardship, wise use and social-hydrological restoration. To achieve this shift, a better understanding is needed on the relations between the social and hydrological systems, and on how these relations vary over time and space (D'Odorico et al., 2019). For example, many locations are experience hydrological problems due to changes in the use of land and water to meet food production, and other domestic and industrial needs (Djuwansyah, 2018). These uses often affect negatively the ability of water systems to retain their hydrological functions, which results in an increase in the water demand (Rosa et al., 2018), leading ultimately to degradation of the water system. Consequently, hydrological restoration aims to re-establish or restore the hydrological functions, and to avoid further hydrological degradation by managing water resources sustainably and/or by eliminating the causal factors of degradation (Zhao et al., 2016).

Four interacting knowledge-to-action steps are needed to determine adequate strategies for social-hydrological restoration (van Noordwijk, 2018). These steps are understanding (technical agenda setting based on social relevance of environmental issues), commitment to goals (social understanding of urgency), operationalization of operationalization of means of implementation based on a common but differentiated responsibility (in its social-ecological context) and innovation for better solutions (through monitoring and learning). Consequently, the first step for any restoration planning is to develop a shared understanding of how the above- and belowground ecosystem structure and climate generate the hydrological functions and underpin the range of ecosystem services provided (van Noordwijk et al., 2022). Furthermore, the interactions between ecological-technical aspects and socio-economic conditions in a landscape (e.g., land tenure, the existence of regulations and incentive-disincentive mechanisms) make the socio-hydrological systems even more complex. Unfortunately, the lack transfer of knowledge between and within different groups of stakeholders often blocks the commitment, operationalization, and innovation stages of successful restoration (Creed et al., 2018).

Learning leads to gaining new information, knowledge, predictive ability, and ultimately to scenario development and knowledgeable decisions. However, providing information alone is not a catalyst that can trigger the associated knowledge to action chain (Marini et al., 2018). Therefore, 'services' that facilitate active learning and 'experiences' that provide a social context to technical aspects are needed for collective learning beyond knowledge transfer. In the 'learning' literature, there is a consensus that people learn more quickly through experiential learning where they can actively explore, engage with the process and then reflect on what happened during the exploration (Kolb and Kolb, 2005; Fanning and Gaba, 2007; Kolbe et al., 2015). Thus, we need tools that can show how a socio-hydrological system works as a whole and allow people to see and experience the consequences of the decisions made, to strengthen knowledge sharing and to facilitate collective decision-making. Two tools are being increasingly used in this context: hydrological modelling (Guo et al., 2021; Tsai et al., 2021) and serious gaming (Rossano et al., 2017; Feng et al., 2018; Ferguson et al., 2020). Hydrological modelling focuses on converting data to information, knowledge and understanding of technical aspects, and it is used to simulate various land-use change

scenarios and quantify the likely consequences of various water management practices (Singh and Kumar, 2017). In contrast,
serious gaming focuses on relating knowledge and understanding of social and technical aspects to enhance the credibility of
decisions made. It adopts the basic elements of gaming, such as challenges, rewards, experiences, strategies, emotions, to allow
stakeholders to safely explore management options (Fleming et al., 2014, 2016).

Although one can see all models as games, and all games as models, these conceptually related tools have developed as separate
communities of practice (van Noordwijk et al., 2020). Games are models as they are succinct and often stylised representations
of a more complex reality, and models are games as they allow the exploration of alternative strategies. In addition, both
approaches require breaking down a complex system into several pieces, which is challenging as not all elements in the real
conditions can and should be included in the models and/or games. Several considerations can serve as a guide in the
simplification process from reality to model and game simulation (Medema et al., 2019), such as what knowledge we want to
share with participants, what we want them to learn, and what changes/responses we expect from them.

Socially interactive games and models that explore larger spatial and temporal horizons have complementary strengths. As
reviewed in Villamor et al., (2023), games and models can 1) seek a conceptual triangulation of representing the processes
behind complex realties, 2) strive for numerical consistency between games and empirical models, 3) use games in the
development of scenario models, or 4) use models in the design of games that trigger players to learn by experiencing
manageable complexity. As an example of the letter, Lohmann et al. (2014) designed and tested model-based role plays with
Namibian land reform beneficiaries, simulating 10 years of rangeland management. In this paper, we explore the feasibility of
transforming a hydrological model into a serious game to provide socio-hydrological dynamics to stakeholders with diverse
backgrounds to develop restoration plans.

Simplifying the complexity of the system and highlighting the socio-hydrological issues from a hydrological model into a
socio-hydrological game will facilitate knowledge transfer among stakeholders and offer a better decision-making tool (Savic
et al., 2016). But such a simplification process can lead to serious games that are very specific to a given local context, making
it difficult for the game to be applied to other places. For that reason, the elements and rules in the game should be easily
adapted to other locations, or at least there should be guidelines on how the game can be applied elsewhere.

Therefore, the objectives of this study are to develop a serious game that is adaptable to different socio-hydrological contexts
and issues, and to evaluate the quality of the game in terms of credibility, salience and legitimacy. To achieve our objectives
we developed a generic game with two adaptations to two different locations in Indonesia differing largely in hydrological
characteristics. First, we developed the $H_2$Ours game based on the socio-hydrological characteristics of the Rejoso watershed
in East Java. Then, we modified the $H_2$Ours game according to the conditions of the Pawan-Kepulu peatland, West Kalimantan.
The qualities of the game were assessed based on several criteria representing credibility, salience and legitimacy which were
included in the game development process and post-game assessment

We organised the paper by presenting as method the stages of how we prepared, designed, tested, implemented and evaluated
the two variants of the $H_2$Ours game. The game itself is the primary 'result', illustrated by the game dynamics during test

settings and early applications with local stakeholders. Feedback by game participants is presented as an evaluation of the current games. We close by discussing the simplification process from reality to game, effectiveness of the game to achieve the goals set, and the lessons learned.

## 2 Methodology

This study consists of four stages from the diagnosis of the study area to the evaluation of the game (Fig. 1). The different stakeholders involved in each stage are also provided.

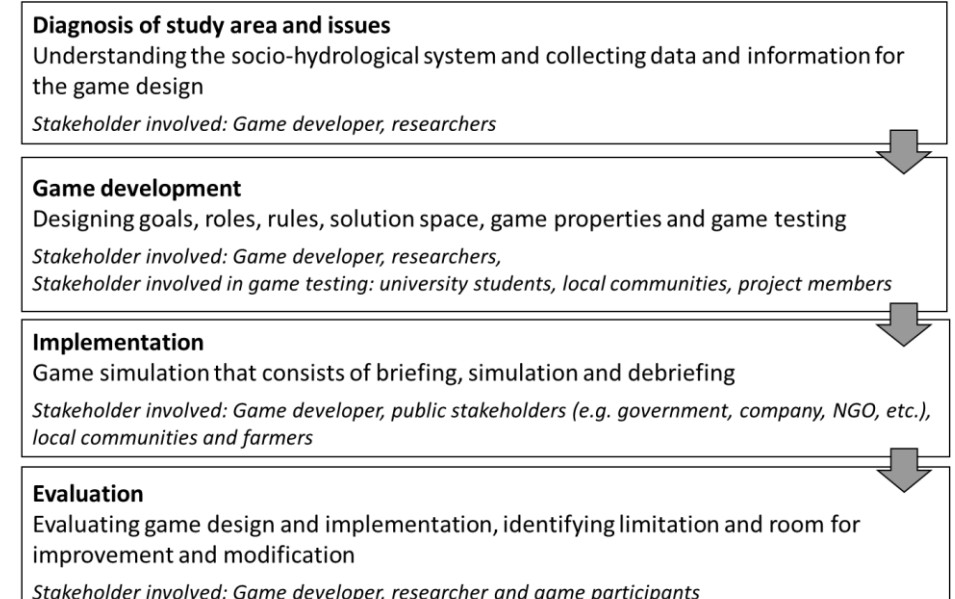

**Figure 1: Stages undertaken from the preparation to the evaluation of the H₂Ours game, including stakeholder involvement across the different stages of this study**

### 2.1 Study areas

The two study areas used in this research, namely the Rejoso watershed and the Pawan-Kepulu peatlands (Fig. 2), differ in physical characteristics (hydrological system, land cover, soil type), but they experience similar socio-hydrological problems (lack of coordination and collective action). In the Rejoso watershed, the hydrological restoration was conducted under '*Rejoso Kita*' project in which World Agroforestry (ICRAF) was responsible for research and development of conservation and restoration strategies, while in the Pawan-Kepulu peatland, the hydrological restoration was conducted by Tropenbos Indonesia through the '*Working Landscape*' project and '*Fires*' project. Both areas have environmental problems because of the disruption of the buffering peak flow that contribute to floods due to lack of infiltration, which in turn is key to the supply of groundwater. To restore those hydrological functions, understanding about the relationship between land-use and (surface-

ground) water management and water balance at the landscape level is crucial before developing a joint strategy (IPBES, 2018).

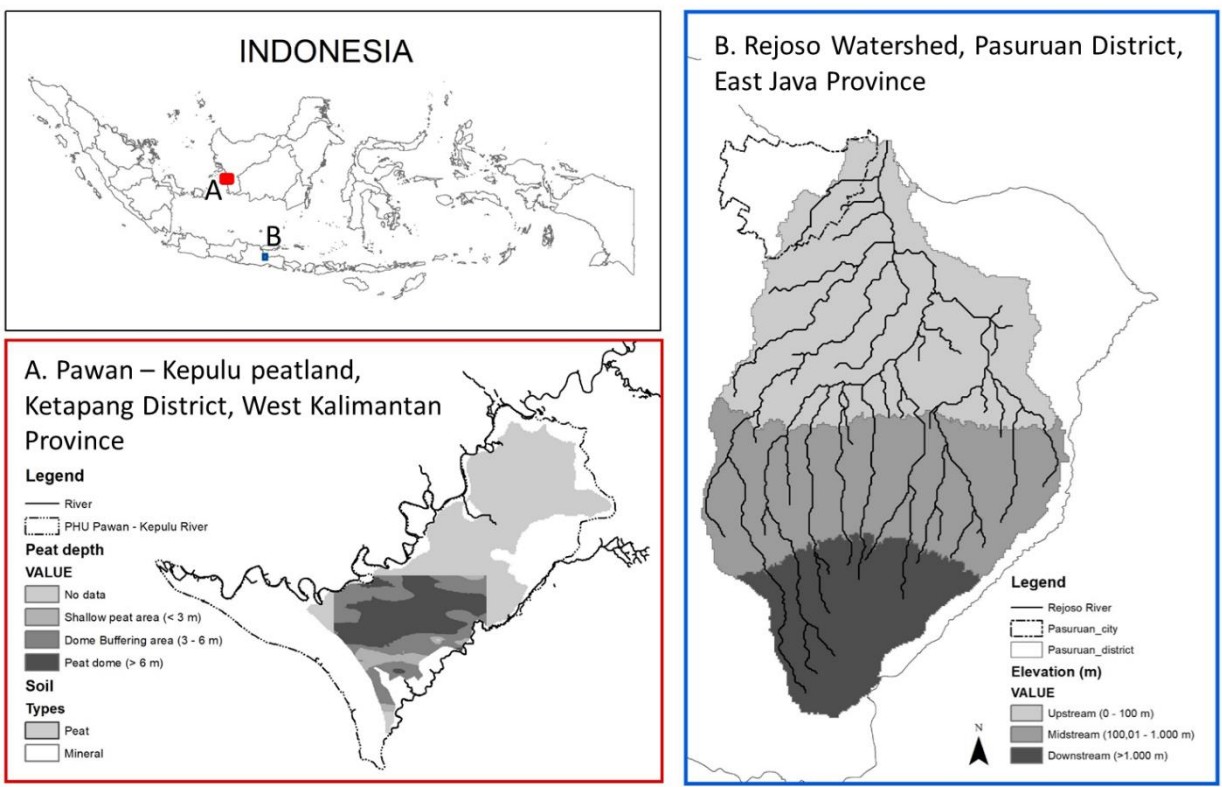

Figure 2: The two study areas of this study: A. Rejoso watershed that consists of upstream (elevation >1000 m above sea level (m a.s.l.)), midstream (elevation 100–1000 m a.s.l.) and downstream (elevation < 100 m a.s.l.), and B. Pawan-Kepulu peatland that
consists of peat dome (peat depth > 6 m), peat buffering dome (peat depth 3–6 m) and shallow peat (peat depth <3 m).

The Rejoso watershed (1600 km$^2$) is in the Pasuruan district, East Java Province, Indonesia. Based on the elevation and hydrological system, we can divide the Rejoso watershed into three areas: downstream (<100 m a.s.l. (meter above sea level), midstream (100-1000 m a.s.l.) and upstream (>1000 m a.s.l.). This watershed is a national priority because the Umbulan spring is used, through a recent pipeline, to supply water to 1.1 million people in the surrounding metropolitan area. Land conversion
from agroforestry to intensive agriculture in the recharge areas (>700 m a.s.l. upstream and midstream area) and massive groundwater extraction for rice fields using artesian wells in the downstream area were understood to cause the reduced average discharge of the Umbulan spring, from 5 m$^3$/s (1980s) to 3.5 m$^3$/s (2020) (Leimona et al., 2018; Amaruzaman et al., 2018; Toulier, 2019; Khasanah et al., 2021). As the declining spring discharge is disrupting the water supply for drinking water, agriculture and industries, stakeholders in the Rejoso watershed need to develop strategies to restore the hydrological function
of their watershed through land-use management in the recharge area and groundwater utilization in the downstream to maintain the continuity of water supply in the Umbulan spring (Khasanah et al. 2021).

The Pawan-Kepulu peatland is located in the Ketapang district, West Kalimantan. This area is between the Pawan and Kepulu rivers, functioning as a unified hydrological system (Fig. 2A). Based on the mapped peat depth, we divided the Pawan-Kepulu peatland into relatively shallow peat area (peat depth <3 m), the dome buffering area (peat depth 3–6 m) and the dome (peat depth >6 m). In the 2000s, local communities and oil palm companies started to build canals for artificial drainage to facilitate timber extraction and for facilitating the management of oil palm and other forms of agriculture (Carlson et al., 2012). However, during the dry season, the canals cause a decrease in the groundwater level so that the peatland becomes drier and more vulnerable to fire. Land fires are detrimental to both the local area and at the global level with the haze and carbon emissions (Widayati et al., 2021). Therefore, there is interest to restore the hydrological function of peatlands to prevent or reduce land fires (Murdiyarso et al., 2021).

## 2.2 Diagnosis of the study areas and issues

In developing the H$_2$Ours game, the system diagnosis relied on hydrological information (e.g., hydrological boundaries, hydrological problems and efforts that my control the causes and overcome impacts), climate condition (e.g., rainfall, potential evapotranspiration), land cover information (e.g., typology, main locally relevant types, recent land cover change and life-cycle profitability estimates), and socio-economic information (e.g., village conditions, socio-economic issues, alternative livelihood options, institutional conditions). These information types were collected using the Rapid Hydrological Appraisal (RHA) approach, which has been used and tested in a number of Southeast Asian countries (van Noordwijk et al., 2013; Jeanes et al., 2006)(van Noordwijk et al., 2013; Jeanes et al., 2006). In this approach, the information were grouped based on local ecological knowledge (LEK), public ecological knowledge (PEK) and modeller/scientist ecological knowledge (MEK). Mapping these different knowledge systems showed overlap, gaps and contrasts that provided starting points for further exploration.

To make it easier to describe the interactions between components in a socio-hydrological system, we structured the socio-hydrological condition of the study area based on the Dynamic, Pressure, State, Impacts and Responses (DPSIR) and Actors, Resources, Dynamic and Interaction (ARDI) frameworks. The DPSIR framework is widely used to carry out hydrological assessments because of its comprehensive connections between various components in a socio-hydrological system (Sun et al., 2016; Lu et al., 2022). We used DPSIR to trace the causes of problems, including interactions and relationships between social and hydrological components and to further explore various responses to socio-hydrological problems (Sun et al., 2016). The ARDI framework is widely used in companion modelling approaches to guide system diagnosis as a first step in designing serious games (Etienne et al., 2011). We used ARDI framework to identify main stakeholders involved in water management, main resources, main processes that affect changes in resources, and interaction between stakeholders and resources (Villamor et al., 2019).

## 2.3 Game development

In this step, we transformed the information from the DPSIR and ARDI analyses into components needed in the game design: goals, roles, rules, and solution space (Fig. 1).

### 2.3.1 Scope and objective

The first stage in designing a serious game is to determine the scope and objective of the game (Silva, 2020; Mitgutsch and Alvarado, 2012). The scope of the game refers to the problem or issues to be addressed. The objective of the game refers to what kind of knowledge, new insight or impacts are expected to be obtained by players after participating in the game. We determined the scope and the objective of the game based on the socio-hydrological problem defined in the previous stage (Sect. 2.2).

### 2.3.2 Roles

According to the ARDI framework (Sect. 2.2), we defined the roles based on the main stakeholders involved in water management in each study area. Most of the players were asked to be a villagers, representing the largest stakeholder group, but others had specific roles as agents trying to influence villager decisions. Related to these roles, we designed goals that players must achieve during each simulation based on discussions and interviews with the related stakeholders according to their actual goal. Before the game started, we asked each group to choose a leader to facilitate discussion within the internal team and represent the group in communicating with other groups.

### 2.3.3 Rules

According to the ARDI and DPSIR frameworks (Sect. 2.2), we transformed the interaction between actors and resources as the rules of the H$_2$Ours game. To show the dynamics of change in resources and the impact of human decisions, the game rules consist of a set of values attached to each decisions type of land-use and water infrastructure that describes both the economic and the water balance component. The economic component consists of the production costs/capital required to manage a certain land-use type and the income derived from that land-use. The water balance component consists of surface flow and infiltration of each land-use type and water infrastructures. The values used as rules for the economic component referred to research findings by ICRAF and Tropenbos Indonesia (Sec. 2.1). For the water balance component, the Rejoso watershed data were obtained from the hydrological modelling and field measurement (Leimona et al., 2018; Suprayogo et al., 2020), while the Pawan-Kepulu peatland data was based on field measurement (Tanika et. al, manuscript in prep.). Several local communities then validated the values through a process of discussion and game testing (Sect. 2.3.6). We simplified the values for each land-use type as a ratio between land-uses to make the quantification process easier during the simulation process. A simple guideline for developing or modifying rules can be seen in the Appendix A.

There are two conditions that are used to mark the position of the participants towards their goals in the game, namely economic and environmental conditions. We derived the economic conditions based on a simple profit calculation equation, where profit is revenue minus all financial expenses (taxes, cost, incidental cost, etc.). The underlying economic analysis applied a life-cycle perspective to the various land-use systems, annualizing discounted future cost and benefit flows. At the sub-landscape level (e.g., upstream, dome), total profit is the difference between total revenue and total production costs. While the environmental indicators were derived based on a simple water balance model implemented in the Generic River Flow (GenRiver) model (https://www.worldagroforestry.org/output/genriver-generic-river-model-river-flow) (van Noordwijk et al., 2017). Consequently, the relationship between the two conditions allowed us to describe the socio-hydrological system of each study area.

### 2.3.4 Game solution space analysis

The purpose of game solution space is to define the envelope of possible outcomes within the rules of the game, considering all possible choices made by players in the game (Speelman et al., 2014). In a random-walk any sequence of steps has equal probability, blind to where it may lead. The solution space of the $H_2$Ours game was explored based on the average of economic and environmental outcomes obtained with a computer simulates random-generator deciding choices for every step. We mapped the estimated solution space after 3, 10, 30, 100, 300 and 1000 random-walk iterations to obtain a reference for the trajectories observed in a limited number of actual, real-player games. The random-walk conditions were generated in R, then simulated using an Excel spreadsheet representation of the $H_2$Ours game and its economic and environmental performance indicators. The 1000 random- walk data set was used to assess the probability density function of outcomes within the solution space. The economic and environmental performance indicators of actual game implementation refer to player's land use decisions from four different game session in the Rejoso watershed which are calculated using the same Excel spreadsheet.

### 2.3.5 Game properties

The purpose of game development is to bring the game design into a real form that players can play or touch such as a game board, various required tokens, and other attributes that support the simulation of the game. We developed the game to be close to the perceived reality, so that players can relate their decisions with the consequences obtained during the game session with the impacts that they have experienced or will experience with the similar decisions. The game board, the game's land-use options, and water simulation miniature are the key elements of recognition for players. Therefore, we adapted these elements to the conditions of each study area.

### 2.3.6 Game testing

The purpose of game testing is to assess the game's playability and dynamics. We tested the game in two ways: checking all the quantification systems using an Excel spreadsheet and the complexity through role playing testing. In the role-playing testing, we tested the game several times with different participants: members of the project, undergraduate students and non-

targeted farmer groups. During the role-playing testing with project members, we checked the suitability and the game elements with the reality; with the students, we calibrated and validated the rules and feedback system in the game; and finally with the farmer groups, we checked if the rules of the game were sufficiently clear.

## 2.4 Game implementation

In this study, we executed ten game sessions which a total of 93 people participating, with five sessions in each of the study areas. All game sessions in the Rejoso watershed were held in October 2021, while in the Pawan-Kepulu peatland were held in August 2022. In each study area, a first one game session was organised with members of a multi-stakeholder forum consisting of representative of governments, NGOs, private sectors, and universities to get ideas on regulations and programs that would be offered to farmer communities, and four game session were organised with farmer groups to explore the implementation of the regulations and programs resulting from the game session with the multi-stakeholder forum.

For each game session, we invited in total of 9-12 representatives of farmer groups from the upstream, midstream and downstream village of the Rejoso watershed, and 12-16 representatives of four villages in the Pawan-Kepulu peatland. In the invitation, we let the group determine who would attend the simulation, provided that the group representatives were willing to hold discussions and exchange information with participants from other villages. For the four sessions with farmer groups, we grouped participants according to different criteria to get a variety of decisions. For the Rejoso watershed, we conducted two sessions with participants who had experience with a recent Payment for Ecosystem Services (PES) program (Leimona et al., 2018) and two sessions with participants from neighbouring villages where the PES program was not active. For the Pawan-Kepulu peatland, we conducted a game session with members of the village forest management unit, a session with members of an active farmer field school, and two sessions with people who are not members of village forest management unit and farmer field school. Game sessions took place in a central location in each of the landscapes to allow easy access for all participants. During the game session, the participants were asked we asked to play the game with the role of a farmers from their location within the landscape.

Each game session required half a day of implementation (briefing, simulation and debriefing), excluding game preparation and participant surveys for further research. We started the session with a briefing of around 10–15 minutes to help participants connect with the game by introducing the environment, setting goals, and clarifying the roles and rules of the game (Rudolph et al., 2014). At the end of playing the game, we did a debriefing of around 30–40 minutes to allow participants to reflect on what they experienced and learned during the game(Crookall, 2023; Kim and Yoo, 2020). To maintain consistency of the $H_2Ours$ for different game sessions, we used the game session guideline provided in Appendix B.

The game explores the trade-off space between economic and environmental outcomes, with the responses from players during the debriefing adding further insights. The economic and environmental outcomes was calculated based on the average economic and environmental conditions as a result of decision making regarding land use combinations during a game simulation over 10 rounds. We present these results together with the results of the solution space analysis to show the position

of players' decisions compared to random decision-making. During the debriefing, we asked participants several questions such as whether they enjoyed the game, what knowledge they gained from the game, how they responded to government regulations of the type included in the game, how they felt seeing other group decisions and (for study case Pawan-Kepulu peatland) their strategies as a member of multi-stakeholder forum

## 2.5 Game evaluation

The aim of the evaluation stage is to assess the game session process and the quality of the game as the basis for the game's performance to fulfil its objectives. The game session process was evaluated based on game performances criteria in the form of rules that can be understood, fun and playability over time. While the quality of the game is assessed based on the scientific logic and reliable knowledge used to build the game (credibility), its relevance to the societal issues (salience) and the acceptance by the game participants (legitimacy) (Cash et al., 2002; van Voorn et al., 2016). For the effectiveness of the assessment, we followed input-output assessment process, which evaluated the input used in the game during development process and the output after the game session (Bedwell et al., 2012). We followed the latter approach and carried out the evaluation based on several criteria that refer to credibility, salience, and legitimacy (Table C1 in Appendix C), using some criteria developed by Belcher et al. (2016).

Because Belcher's long list of criteria (Belcher et al., 2016) originally was used to assess the quality of research, for this study we chose several criteria that were relevant to game quality. Each of these criteria were measured during the game design process and after the game implementation. We measured these criteria by how it was associated with the condition and diagnosis of the study area (Section 2.1 and 2.2) and game development process (Section 2.3). Please see Table C1 to see the parameters and sections associated with each criteria. A rapid evaluations were conducted after the game session to assess the process and the quality of game session. We converted those game performace criteria and creadibility, salience and legitimay criteria into Likert used questions and asked all game participants to fill in the survey. In the Likert survey, we used five-point scales (strongly disagree, disagree, neutral, agree, and strongly agree) on six statements to ask participants about their feeling during the game, their understanding of the rules of the game, the length of the game simulation, new knowledge that they got from the game, and implementation the game to their reality.

## 3 Results

We organized the results section by presenting the Rejoso watershed version and the Pawan-Kepulu peatland version site by site to make it easier to see the similarities and differences between the two applications even though the Pawan-Kepulu peatland version of the $H_2$Ours game was developed after the Rejoso watershed version.

## 3.1 Diagnosis of the study areas and issues

Based on the results from the DPSIR and ARDI analyses, we found that the Rejoso watershed and the Pawan-Kepulu peatland showed similarities in their socio-hydrology context (Table 1). Stakeholder's expectations for improved economic conditions led local communities to change land cover, and extract excessive amounts of water resources (groundwater) causing disruption of the water balance. This disruption resulted in local communities and multi-stakeholder forum experiencing various hydrological problems, such as water shortages (or decreasing the groundwater level) and flooding. However, the hydrological context of these two sites also differ regarding their hydrological contexts, such as hydrological boundaries, topography, and water management, and interactions among stakeholders and landscape (Fig. 3, Fig. D1). Two proposed solutions (responses) were identified by ICRAF and Tropenbos Indonesia based on their research findings to restore hydrological functions in watersheds and peatlands, namely better land use management and (ground) water management (Table 1; component 7-Response).

**Table 1. Framing problem definition for the Rejoso watershed and Pawan-Kepulu peatland, Indonesia. Problem definition was done the using Driver, Pressure, State, Impact and Response (DPSIR) and Actor, Resource, Dynamic and Interaction (ARDI) frameworks, based on ICRAF and Tropenbos research findings**

| | COMPONENTS | REJOSO WATERSHED | PAWAN-KEPULU PEATLAND |
|---|---|---|---|
| 1 | Hydrological boundary/ landscape | Watershed (and/or groundwater catchment) | Peatland hydrological unit |
| 2 | Zone partition | (1) Upstream: elevation >1000 meter above sea level (m a.s.l.) <br> (2) Midstream: elevation 100-1000 m a.s.l. <br> (3) Downstream: elevation <100 m a.s.l. | (1) Dome: peat depth > 6 m <br> (2) Buffering area: peat depth 3-6 m <br> (3) Shallow peat area: peat depth <3 m |
| 3 | **D**river | To get a better household income and livelihood | |
| 4 | **P**ressure | (1) Land-use conversion into non-tree-based system in the recharge area (upstream and midstream) <br> (2) Massive artesian well construction for paddy field (downstream area) | (1) Land-use conversion into oil palm (dome and buffering area) <br> (2) Massive canal construction to drain peatland water |
| 5 | **S**tate | (1) Increasing runoff and reducing infiltration (upstream and midstream) <br> (2) Increasing groundwater uptake (downstream) | Increasing water outflow from peatland and decreasing peatland water level. This condition makes peatland become drier during the dry season |
| 6 | **I**mpact | (1) Decreasing groundwater supply in the Umbulan spring <br> (2) Floods (during rainy season) | (1) Peat fires (during the dry season) <br> (2) Floods (during the rainy season) |
| 7 | **R**esponse | (1) Land-use/cover management <br> (2) Better groundwater management through artesian well management | (1) Land-use/cover management |

| | | | | |
|---|---|---|---|---|
| | | | (2) | Better groundwater level through canal blocking management/ distribution |
| 8 | **A**ctors | Multi-stakeholder forum and farmers/local communities | | |
| 9 | **R**esources | (1) Money (2) Water balance (especially groundwater and surface water) | (1) Money (2) Water balance (especially groundwater and surface water) | |
| 10 | **D**ynamic | (1) Land-use/cover change (2) Water management (artesian well management) | (1) Land-use/cover change (2) Water management (canal blocking management) | |
| 11 | **I**nteraction | Fig. 3 | Fig. D1 | |

The interaction between stakeholders and the landscape is represented by the type of decisions regarding their landscape taken by the multi-stakeholder forum and local communities. Local communities (farmer from upstream, midstream, and downstream villages in the Rejoso watershed and farmers from neighbouring villages: Village 1-Village 4 in the Pawan-Kepulu peatland) have the authority to make decisions regarding their land including land-use and water management decisions (artesian wells in the Rejoso watershed and canal blocking in the Pawan-Kepulu peatland). Multi-stakeholder forums have authority over regulations and programs applied to local communities to achieve their goals. Multi-stakeholder forum can refer to their existing or potential regulations and programs.

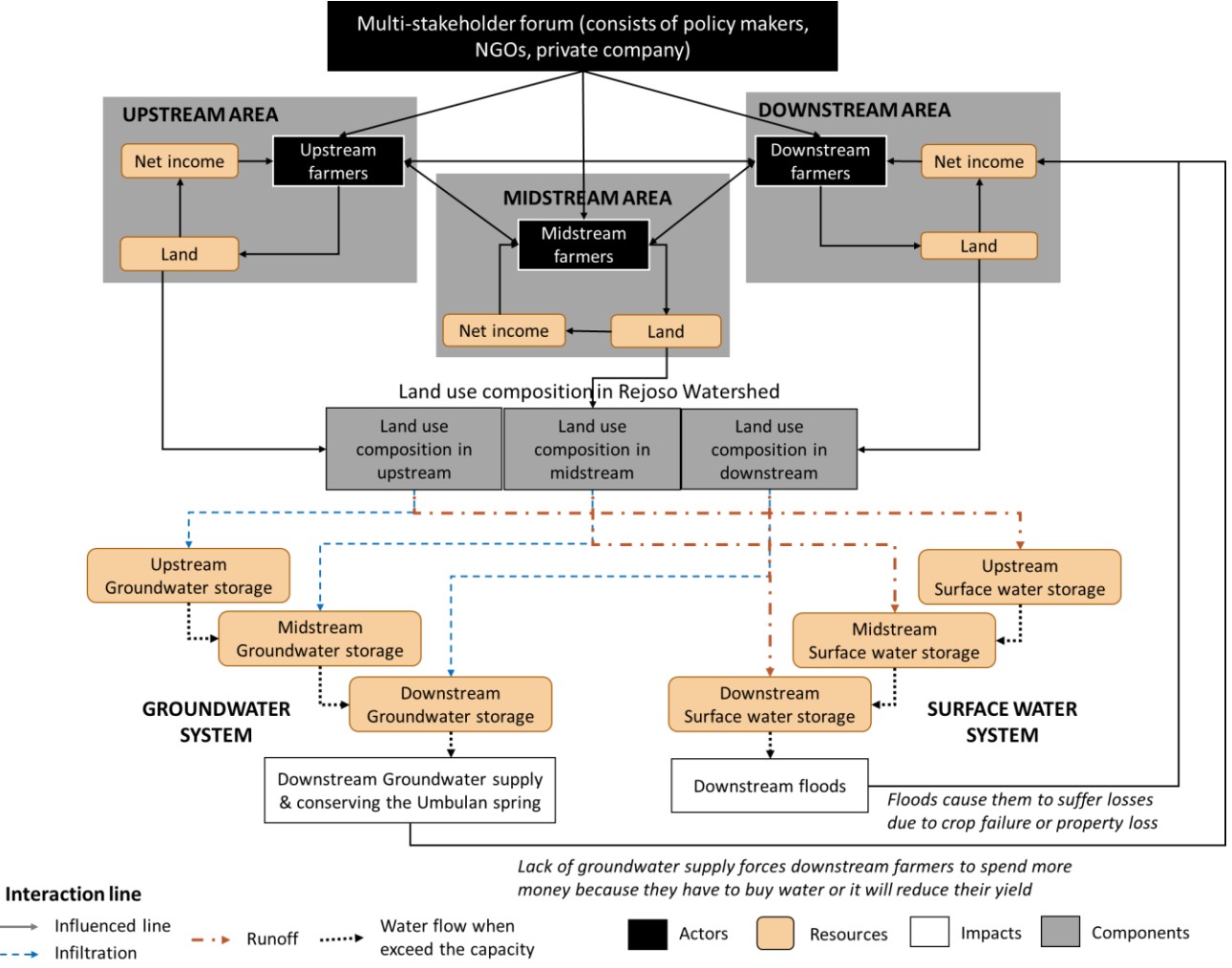

Figure 3: Socio-hydrological model for the Rejoso watershed, defined using the ARDI framework. Interactions among actors and between actors to landscape influence land-use composition. The land composition affects the hydrological and economic situation, which influences back to the interactions. A similar socio-hydrological model with some adjustments for Pawan-Kepulu peatland was also developed (Appendix D).

### 3.2 Game development: H₂Ours game

#### 3.2.1 Scope and objective of the game

The H₂Ours game has the objective to help sharing knowledge and building collaboration among stakeholders to restore hydrological functions in a landscape. We determined the goal for the H₂Ours game simulation in the two study areas to be knowledge sharing and facilitating collaboration, specifically for groundwater water restoration and flood prevention (Table 1). However, the H₂Ours game in the Rejoso watershed addressed the supply and utilization of deep groundwater, while in

Pawan-Kepulu peatland it addressed peatland's groundwater as an indicator of the wettability of peatlands and its vulnerability to land fires.

### 3.2.2 Roles

Based on the stakeholder identification survey in the Rejoso watershed and the Pawan-Kepulu peatland, we defined two key roles in this game, namely a multi-stakeholder forum and local (or farmer) communities. The goal of the multi-stakeholder forum is to prevent natural disasters specifically water scarcity and floods in the Rejoso watershed, and fires and floods in Pawan-Kepulu peatland. In the Rejoso watershed, local communities can be grouped into people who live in the upstream village, midstream village and downstream village based on the village elevation. Meanwhile in the Pawan-Kepulu peatland, local communities can be grouped into four groups of people living in four neighbouring villages (Village 1 – Village 4). Local communities represent landowners. Their goal is to fulfil their household needs (to produce sufficient food and to raise sufficient funds to pay taxes). The H$_2$Ours game brings the various interests of these actors together and shows how they make their decisions regarding the management of land and water resources to meet their economic and environmental expectations.

### 3.2.3 Rules

At the start of the game, players (i.e. multi-stakeholder forum or local communities) received a limited amount of play money. Community members were asked to manage their land to meet their household needs by arranging the land-use type combination and water management in their area with the play money provided, while multi-stakeholder forum was asked to run programs or to help reduce the local community's financial problems. Once players decided on how they would manage their land or community programs, the economic and environmental rules linked to those land-use decisions were applied (Table 2). These rules then defined the dynamics of the economic and environmental conditions (Table 2, and Table D1 and D2 for the Pawan-Kepulu peatland).

When during the rainy season the total of surface water in the downstream area of the Rejoso watershed and in the shallow peat of Pawan-Kepulu peatland exceeds its capacity (>800 ml), flooding occurs. When the groundwater exceeds its capacity (>700 ml), excess water flows to the Umbulan springs in the Rejoso watershed and to sea in the Pawan-Kepulu peatland. But, when the groundwater was less than <200 ml, it caused water shortages for agriculture in the Rejoso watershed and made peat soil dry which in turn triggered fires in the Pawan-Kepulu peatland. These environmental impacts decreased the overall community income. As a consequence of this, the players might not have enough money to manage their land, buy food or pay taxes in the next round of the game. The multi-stakeholder forums with their limited budget could then choose to help them by providing financial help or making regulations/programs to prevent these environmental problems. Through this gameplay, we aimed to stimulate players to collaborate to achieve their goals.

**Table 2. Economic and environmental impacts as the rules of the H₂Ours game in the Rejoso watershed. The variation of environmental components resulting from different land-use options in the upstream and midstream depends on the ability of the land-use options to infiltrate water, while the variation of environmental components downstream depends on the use of water based on farmers' perceptions. The rules of the H₂Ours game in the Pawan-Kepulu peatland are in the Appendix D. (AF= agroforestry).**

| Land-use | Production cost (unit money) | Income/year (unit money) | | Environment impacts during wet year (ml) | | | Environment impacts during dry year (ml) | | |
|---|---|---|---|---|---|---|---|---|---|
| | | Wet year | Dry year | Runoff | Infiltration | Water use | Runoff | Infiltration | Water use |
| **UPSTREAM AND MISTREAM** | | | | | | | | | |
| All crop | 12 | 25 | 13 | 40 | 0 | 0 | 0 | 0 | 5 |
| Mixed AF low density | 9 | 17 | 9 | 30 | 10 | 0 | 0 | 0 | 5 |
| Mixed AF moderate density | 6 | 9 | 6 | 20 | 20 | 0 | 0 | 0 | 0 |
| Mixed AF high density | 3 | 6 | 4 | 10 | 30 | 0 | 0 | 0 | 0 |
| All trees | 1 | 0 | 0 | 0 | 40 | 0 | 0 | 0 | 0 |
| **DOWNSTREAM** | | | | | | | | | |
| Paddy | 12 | 12 | 25 | 0 | 0 | 10 | 0 | 0 | 15 |
| Maize | 9 | 15 | 18 | 0 | 0 | 5 | 0 | 0 | 10 |
| Orange | 7 | 11 | 15 | 0 | 0 | 0 | 0 | 0 | 5 |
| Cucumber | 9 | 15 | 13 | 0 | 0 | 2.5 | 0 | 0 | 7.5 |
| Banana | 5 | 10 | 10 | 0 | 0 | 0 | 0 | 0 | 0 |

In addition, the economic and environmental conditions during the game are also influenced by the yearly weather that could be either wet or dry year. In each round, participants decided on land-use not knowing whether the next round would be a 'dry' or 'wet' year.

### 3.2.4 Game solution space analysis

From the comparison results between 3, 10, 30, 100, 300 and 1000 computer simulated random-walk iterations, we found that the shape and distribution of economic and environmental outcomes began to stabilize at 300 iterations. Therefore we used 300 computer simulated game sessions with randomly selected land and water use options as the basis for creating the solution space of this research, as reference for the player-based game sessions. In 300 computer simulated game runs with a random decision making process, the groundwater distribution varied depending on the location, while the distribution of surface water in the upstream and midstream remained almost the same, and in the downstream was wider (Fig. 4A and Fig. 4B). Upstream

and midstream had almost the same frequency distribution of surface water flows while runoff from the upstream and

midstream areas was dominated by wet years, which then may potentially cause flooding downstream in the same year. Contributions to groundwater from upstream and midstream areas also responded to wet years, while groundwater utilization occurs mostly during the dry years by downstream stakeholders. Therefore, the frequency distribution of groundwater contributions were wider than those for surface water.

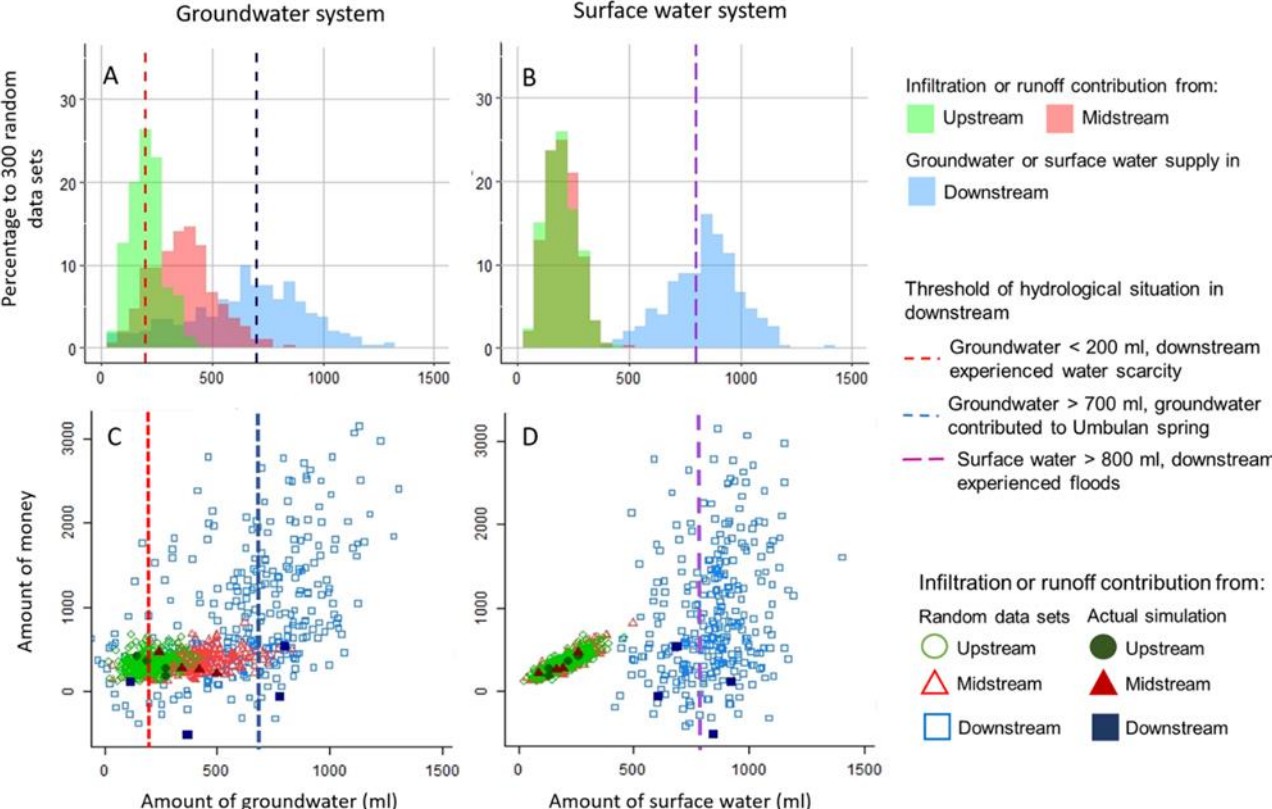

**Figure 4: Simulation of hydrological and economic situation in the H$_2$Ours game using random value (N = 300) and game actual simulation (obs.) results (N = 4) for the Rejoso watershed. A. Distribution of infiltration contribution from upstream and midstream and groundwater supply in downstream based on simulation with the random value; B. Distribution of runoff contribution from upstream and midstream and surface water accumulation downstream based on simulation with the random value; C. Groundwater situation and economic situation based on random value simulation and actual simulation; D. Runoff situation and economic situation based on random value simulation and actual simulation. Appendix E provides a further analysis of the solution space**

Related to the economic outcomes (Fig. 4C and Fig. 4D), efforts to increase infiltration in the upstream and midstream did not contribute much to increasing the income of the community. However, the efforts of farmers in the upstream and midstream areas to improve their economic conditions resulted in increased runoff, which caused flooding in the downstream areas. Therefore, for the downstream area, the relationship between environmental and economic conditions varies because of the influence from upstream and midstream conditions.

The presence of relationship values between humans and nature and humans and other humans (relational values) influences decision making regarding natural resource management (van Noordwijk et al., 2023, 2020). Therefore, the decisions made

by players during the game are influenced by various factors (e.g. interactions between players, game settings, level of player ecological knowledge, etc.) (Rodela and Speelman, 2023), while computer simulated random decision making is used to build the game's solution space. For example, when the upstream and midstream groups decided to maintain and improve their economic conditions, they caused a reduction in groundwater supply and increase flooding for downstream area, which caused the downstream group to pay for the losses the experienced. Apart from that, during the game session the facilitator also

provided PES scenarios (Appendix B, Game Play number 9: repeat step 6 for the rest of the rounds with additional scenarios such as providing payment for ecosystem services). This scenario offered downstream groups to contribute a certain amount of money to maintain more trees in the upstream and midstream. Therefore, the downstream player groups always spend more money than the mid- and upstream player groups either as a loss due to the environmental consequences (floods or water scarcity) or due to their efforts to prevent negative impacts by joining the PES program.

**3.2.5 Game properties**

To make the game engaging, we prepared game materials such as a game board to represent the landscape, land-use tiles according to the existing and future land use types, play money token, and water infrastructures token (Fig. 5). We also created, a stylised miniature water balance (Fig. 6) to demonstrate how surface water flows and can cause floods and how water infiltration increases ground water supply. Each round after calculating the economic condition and environmental conditions

based on Table 2, we asked players to pay production costs, taxes, etc. and get income, incentives, etc. using play money. The water balance was shown via a miniature with real water according to the produced surface and groundwater.

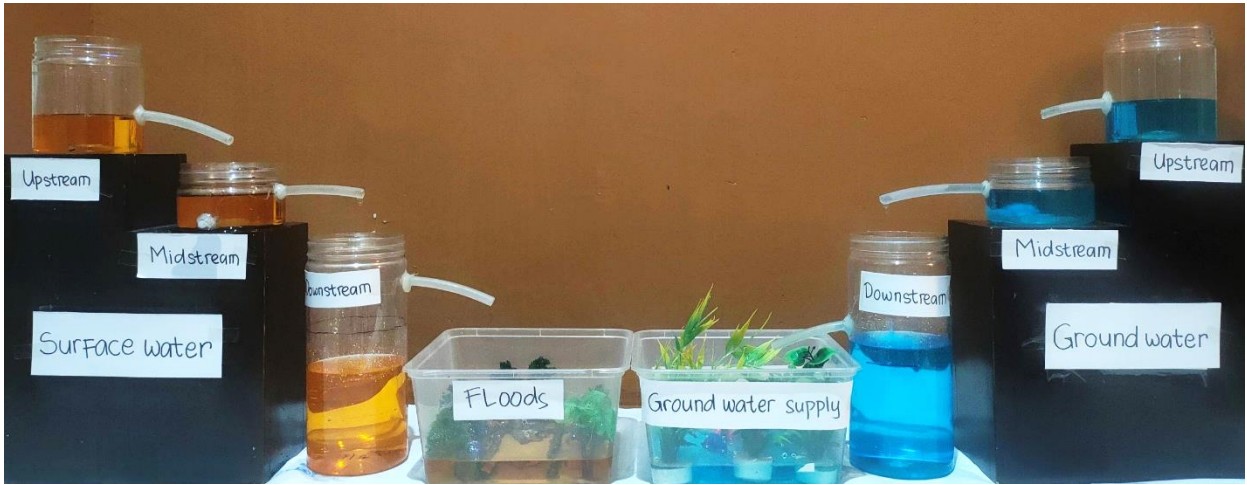

Land cover tiles for upstream village

All crops | Agroforestry with low density tree | Agroforestry with moderate density tree | Agroforestry with high density tree | All trees

Land cover tiles for midstream village

All crops | Agroforestry with low density tree | Agroforestry with moderate density tree | Agroforestry with high density tree | All trees

Land cover tiles for downstream village

Paddy | Maize | Orange | Cucumber | Banana

Village 1 — Upstream

Village 2 — Mistream

Village 3 — Downstream

**Figure 5: Game board and land use/cover tiles of the H₂Ours game in the Rejoso watershed. The land cover options in the upstream and midstream area varies based on their ability to infiltrate water, while in the downstream area varies based on farmer's perception on water utilization. See appendix D for the game materials for the Pawan-Kepulu peatland.**

Upstream — Midstream — Downstream — Surface water — FLoods — Ground water supply — Upstream — Midstream — Downstream — Ground water

**Figure 6: Simple water balance model of H₂Ours game in thw Rejoso watershed to show the dynamics of changes in hydrological conditions because of land-use change and water utilization. See appendix D for the simple water balance model for the Pawan-Kepulu peatland**

### 3.2.6 Game testing

From the results of checking the game calculation in excel, we adjusted the values used in the rules to ensure that these values are sensitive enough to changes in strategy by players, i.e., the initial money given to players, as well as the initial water for groundwater and surface water. The role-playing testing with project members allowed us to validate the game scenarios that would be applied in the game implementation; with the university students, we adjusted the flow of the game, and we set the number of rounds to 8-10 rounds, and the length of simulation time to two hours; and with the local communities (non-targeted participants), we checked the terminology used during the game session.

### 3.3 Game implementation

The game session with the $H_2$Ours game takes approximately two hours (excluding briefing and debriefing). For the Rejoso watershed version, the 2 hour game session consisted of 10 rounds with 6-12 players divided into 3 groups (or 2-4 people per group) acting as local communities: upstream, midstream, and downstream. The Pawan-Kepulu peatland version, the 2 hour game session consisted of 8 rounds with 8-16 players divided into 4 groups, and players are asked to select their village name as first step of creating ownership. In both versions, an additional group of players consisting of 2-4 people can act as public stakeholders (government, companies, NGOs) and interact with the villages.

During the game session, players in their role of farmers or local community tried to improve their household income and livelihood to at least manage until the next year. The results of the game implementation showed that there was a trade-off between economic and environmental conditions, and between the upstream, midstream and downstream groups (Fig. 4, below). In the Rejoso watershed, the efforts of the upstream and midstream communities to improve their economic situation by increasing crop area brought a negative environmental impact for downstream communities as flooding and water scarcity. The efforts of upstream and midstream communities to reduce these problems resulted in a reduction of their economic outcomes. This situation led to negotiation among communities. In contrast, the negotiation process in the Pawan-Kepulu peatland was related to the canal blocking construction among villages and between villages with the multi-stakeholder forum. To achieve a closed hydrological system to maintain the wetness of the peatland, the construction of canal blocking must be carried out collectively by all villages according to the location as suggested by the multi-stakeholder forum. The construction of canal blocking reduced the income of farmers/local communities due to decreased yield or increased harvesting costs. Furthermore, the multi-stakeholder forum also persuaded the community by providing compensation for maintaining more trees to protect the peat dome area.

During the debriefing of the sessions, the participants in the Rejoso watershed and the Pawan-Kepulu peatland mentioned that the game showed that any decision at plot level impacted the hydrological function at the landscape level. They also mentioned that if they had not met their economic needs, the economic conditions became their priority. In addition, they indicated that they would accept any regulation or program from other stakeholders as long as their income would not significantly reduced. But, if that would happen, they would expect some financial compensation. From the multi-stakeholder forum's perspective,

they said that it would be easier if the village knew what they wanted in advance, so that the programs and assistances would be able to match their needs. In addition, regulations should also be complemented by supporting schemes, such as compensation or incentive schemes, not just regulations issued by the government. Further analysis to these different perspectives will be presented in follow-up manuscripts (Tanika et al, in prep).

## 3.4 Game evaluation

After playing the game, the participants of both study areas were asked to fill out a survey to assess the credibility, salience and legitimacy of the game (Appendix C, Table C1). For the credibility of the game, the survey showed that on average 87% of the participants from both study areas indicated that they understood the rules of the game well or even very well, while 78% of participants indicated to know the purpose of the game. For the salience and legitimacy of the game, the survey showed that 92% of participants gained new understanding and 87% said that they could apply the knowledge that they took away from the game to their real life. Besides the credibility, salience and legitimacy criteria, we also asked the participants about their opinion regarding the game session process. From the survey, 87% of the participants enjoyed the simulation and 79% of them felt that the length of game session was fair.

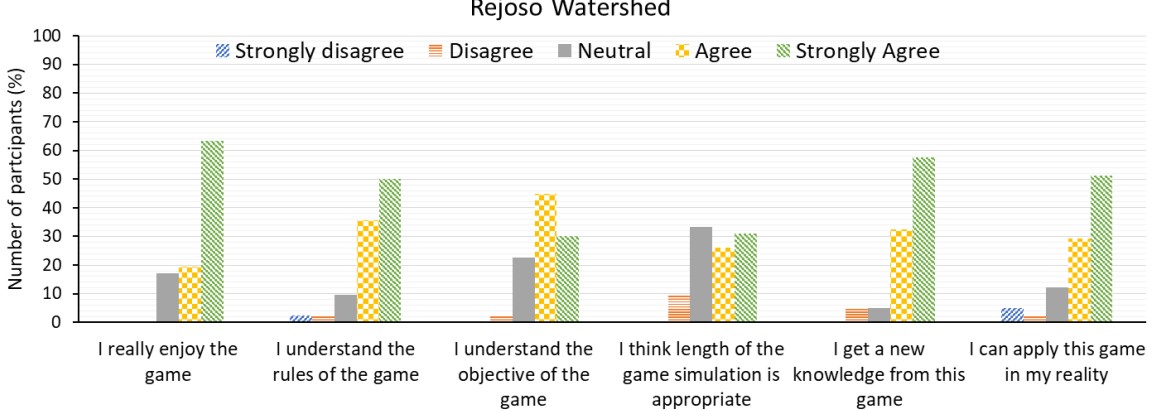

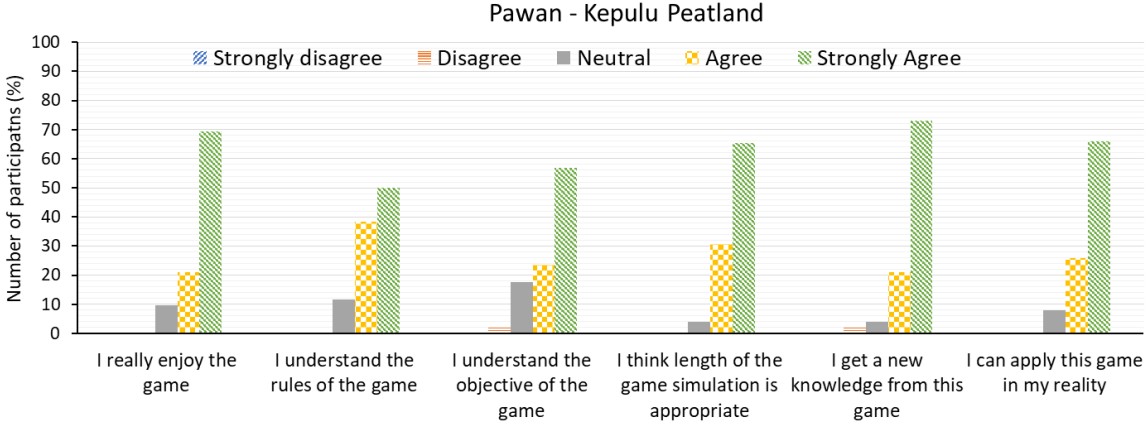

**Figure 7: Game evaluation from the participants in the Rejoso watershed (N = 41 people) Watershed and Pawan-Kepulu peatland (N= 52 people)**

## 4 Discussion

To meet the first objective of this paper to develop an adaptable serious game that can represent the socio-hydrological system,
we presented the generic version of the $H_2$Ours game as the result of the development and modification process in two different landscapes in Indonesia (Sect. 4.1). Then, to assess whether the $H_2$Ours game can facilitate the knowledge transfer and knowledge sharing regarding water use and management and can support negotiation and coordination among various stakeholders as the second objective, we evaluated $H_2$Ours game based on input-output assessment according to evaluation criteria (Sect. 4.2).

## 4.1 The adaptability of the H₂Ours game allows simplifying complex socio-hydrological systems

The complexity of a system is closely related to the number of interdependent information and interactions between elements in the system (Vidal and Marle, 2008; Rumeser and Emsley, 2019). Serious game and the associated conceptual models help to simplify this complexity by reducing the amount of information and interactions, and by only showing the most relevant information from a holistic perspective (Strait and Dawson, 2006; Rumeser and Emsley, 2019). For the development of the H₂Ours game, we used the DPSIR and ARDI frameworks to identify the components and the interconnections of the complex socio-hydrological system of the Rejoso watershed (Table 1, column 3). The rejoso watershed version of the H₂Ours game was then adapted to the Pawan-Kepulu peatland version by modifying the socio-hydrological condition (Table 1, column 4). Therefore, these well-established frameworks act as the generic version of the H₂Ours game, which can easily be modified according to other socio-hydrological realities.

The two study sites experience more complex socio-hydrological problems than represented in the H₂Ours game. In our game, the water quantity issues were represented in line with national priority issues in that location, which resulted in groundwater scarcity and floods for the Rejoso watershed, (Fig. 3) and fire and floods for the Pawan-Kepulu peatland (Fig. D1). In reality, the Rejoso watershed also experiences other hydrological problems, such as erosion and landslides in the upstream areas, water quality degradation due to high amount of chemical fertilizer (Amaruzaman et al., 2018; Leimona et al., 2018), while the Pawan-Kepulu peatland also experiences land degradation and water contamination because of mining in the upper area of peatland (Widayati et al., 2021). The complexity of a socio-hydrological system is formed due to many relationships and inter-connection of the various components (aggregate complexity), therefore self-organization through gradual learning is the key to a better transformation (Manson, 2001). If all the real life problems would have been included at once in the game, the risk of confusing people, especially those without a technical educational background (Gomes et al., 2018), which would preclude their understanding of the causes and effects of the problem. Therefore, by unravelling each individual problem and showing its causes and associated-impact, players were able to expand their understanding gradually. We believe that the generic game H₂Ours creates the opportunity to explore different problems, allowing the players to gain a deeper understanding and start building connections among various problems. In this way, it is possible to create opportunities to build overall socio-hydrological understanding in the future.

By comparing the H₂Ours game in the two study areas, we found that there were game elements that could remain the same while others had to be adjusted to the local situation. Game elements related to the interaction among humans or between humans and the environment (relational value) are similar in the two study areas (e.g. land-use management to maximize profits, effort scenarios to restore hydrological functions, the need for coordination and negotiation among stakeholders). As such these elements maintained the same between the locations (Driver and Pressure in Table 1). However, the environmental response to the drivers and pressures generally requires technical adjustments to local conditions e.g., hydrological boundaries, land-use types and composition, water infrastructures, hydrological systems (Table 1, State and Impact). Therefore, our generic H₂Ours game (defined using the components of Table 1) showed to be easy to adapt to other problems and/or other locations.

In addition, it is expected to overcome the complexity of a system as we can choose what the most important and most influential socio-hydrological problems that want to be addressed.

## 4.2 Game evaluation and lesson learned

During the game design, we evaluated the H$_2$Ours game using the input-output assessment process (Bedwell et al., 2012). Here, credibility, salience and legitimacy were assessed throughout the different stages of the H$_2$Ours game development (Fig. 1). During the game development of the Rejoso watershed, we assessed the credibility of the H$_2$Ours game by relaying on the biophysical and hydrological research, including hydrological modelling through the GenRiver model (Leimona et al., 2018; Suprayogo et al., 2020), while in the Pawan-Kepulu peatland based on the biophysical measurement and hydrological modelling (Tanika et. al, manuscript in prep.). For the salience and legitimacy, we relied on results of participatory research executed involving various stakeholders in the Rejoso watershed and Pawan-Kepulu peatland (Amaruzaman et al., 2018; Leimona et al., 2018; Widayati et al., 2021; Leimona and Khasanah, 2022). By considering the criteria of credibility, salience and legitimacy since the data and information collection, it was easier for the H$_2$Ours game to fulfil these criteria during the evaluation after the simulation.

We limit the evaluation in this study only to the quality of the game as a product. As a serious game, the H$_2$Ours carries certain goals that it wants to fulfil (Rodela et al., 2019), namely as a tool that can facilitate the transfer and sharing of knowledge from its players to support the coordination and negotiation process (Section 3.2.1). Evaluating the game in fulfilling its objectives is more complicated than evaluating the game session process. Ideally, the evaluation of the game in achieving its objective can be evaluated after several simulations at various levels of simulation, and should be conducted before, during and after the game sessions (Oprins et al., 2015). The evaluation of the game to meet the objective will be carried out in the next manuscript by providing evidence of changes in participant's perceptions

As hydrological problems are usually complex and fundamental, any potential solution requires ample time for integrated planning, and all relevant stakeholders to understand the dynamics of the system at large scale (Medema et al., 2019). The H$_2$Ours game tries to provide a simple representation of the landscape so that it makes it easier for players to be aware of the conditions of neighbouring players and to gain system level perspective of socio-hydrological issues. Improving player knowledge by looking at socio-hydrological problems in a broader context encourages responsible behaviour towards the environment which is directly proportional to commitment (Keles et al., 2023). The evaluation of the game after the simulation (Fig. 7) indicated that most of the participants gained new knowledge from the game which they could apply in real life. Transparency of the rules of the H$_2$Ours game allowed players to see the interdependent connections between elements in the complex socio-hydrological system more clearly and made it easier for players to explore various possibilities and to gain lessons from the reflection results (Kolbe et al., 2015; Kolb and Kolb, 2005; Fanning and Gaba, 2007). During the game session, after the players began to understand how the H$_2$Ours game works, the players started to initiate communication in the form of negotiations or coordination between groups or with external parties such as multi-stakeholder forum. This is in

accordance with the four interacting knowledge to action steps in restoration strategies, which commitment begins after the mutual understanding have been made (van Noordwijk, 2018). One of the advantages of a serious game is that participants interact directly with the environment and get feedback as quickly as possible so that they can immediately analyse and correct inappropriate strategies (Bartolome et al., 2011; Feng et al., 2018). Moreover, during the H$_2$Ours game session, players were also faced with the game situation that resemble actual situation, so they are indirectly encouraged to find possible solutions together as two last parts of restoration strategies related to operationalization and innovation.

There are several lessons learned from the H$_2$Ours game development and simulation process in this study. First, setting up the game material with attributes of the local context helped participants to build emotions during the simulation. Second, to maintain participant commitment to restoration efforts after the game session, it is important to show that their collaborative and collective actions really worked in achieving their goals at the end of the game simulation. Third, based on the evaluation and debriefing results, even if they stated they could apply the ideal collaborative actions that were explored in the game session, in real life, the enabling conditions needed to support this still required to be build (e.g. regulation, integrated planning strategies, etc.). As the game is a simplification of the real-life system, forms of collaborative action can be discussed directly by the players. In real life, the parties that are needed for successful collaboration may not easily meet each other to discuss issues openly. Therefore, it is necessary to create a condition where stakeholders can meet and explore collaboration options to jointly address issues and achieve goals. Without such encounters, the commitment that referred to in the four knowledge-to-action chains cannot be attained.

The H$_2$Ours game clearly showed the trade-off between the economy and the environment by calculating economic and environmental performance indicators in each round after the players change the land use combination and water management. As a result, the relational value between humans and human with nature (e.g. trees and water being inherited from their predecessors and will be a legacy for their descendants, the use of certain woods in religious rituals) sometime becomes blurred. A very clear trade-off between the economic and environmental conditions have led players to make decisions based solely on economic value. Therefore, the cost-benefit calculation of conservation activities needs to be done carefully in this game or include social values as part of the scenario in the game.

In this research, we invited participants from upstream, midstream, and downstream to play from the perspective from their location in the landscape. We expect that this impacted how the game was played. We intend to explore the impacts of role switching by asking farmers to play the role of a farmer in another location in the landscape.

**5 Conclusion**

The generic version of the H$_2$Ours game allows for the exploration of the complexity of a socio-hydrological system. The game can be easily modified according to different needs and conditions. The complexity of the socio-hydrological system can be applied separately and/or simultaneously depending on the knowledge level of the intended participants. With an

adaptable game as the one developed, the game designer can adjust the level of complexity included in the game, and even include an advanced simulation that combines all possible problems and interactions found in a socio-hydrological system.

The H$_2$Ours game showed to facilitate knowledge transfer and knowledge sharing, and triggered collaborative actions by simplifying in time and space. The H$_2$Ours game save the time because the transparency of the rules allows players to see that
the restoration target is something that can be achieved in the future with a clearer perspective by exploring various strategies and scenarios during the game sessions. Space simplification allowed players to see the entire landscape and the relationship between components that influence each other. In addition, they can also easily see the various enabling conditions needed to make the strategies in the game can be implemented in real terms (e.g. need of for multi-stakeholder collaboration, restoration masterplan).

**Appendix A. H$_2$Ours rule development**

One of the challenges in developing or modifying the H$_2$Ours game is providing values for the economic and environmental impact components for each type of land-use. Here is a simple guidance to modifying the H$_2$Ours game rules:

1. Determine the types of land-use in the landscape. If the land-use types are varied enough, take the 4-6 most dominant land covers, including the new land-use types that might be intervened.
2. For each type of land-use, determined the amount of the economic value (production costs and income) and environmental value (runoff, infiltration, water use/utilization). The value used as a rule does not have to be the actual value. You can only use the ratio value between land-use types after setting up the maximum and minimum value. A simple method to collect this information is by conducting survey to several farmers and ask them to rank or make score the land-use type based on their economic and environmental impacts (Fig. A1).
3. Determine infrastructures that will be used in games that might affect economic and environmental conditions (e.g. artesian wells for irrigation, canal blocking, water storage, etc.).
4. Determine how each of these infrastructures affects economic and environmental conditions (e.g. artesian wells: construction cost, threat, amount of groundwater extraction, etc.). You can conduct a survey to collect that information, then normalize the value following the economic and environmental value.
5. During the game testing, evaluate those values with the participant whether it is reasonable and represents their actual condition

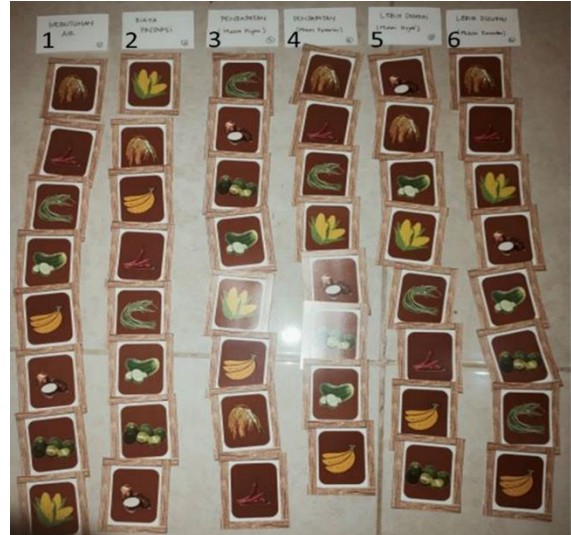 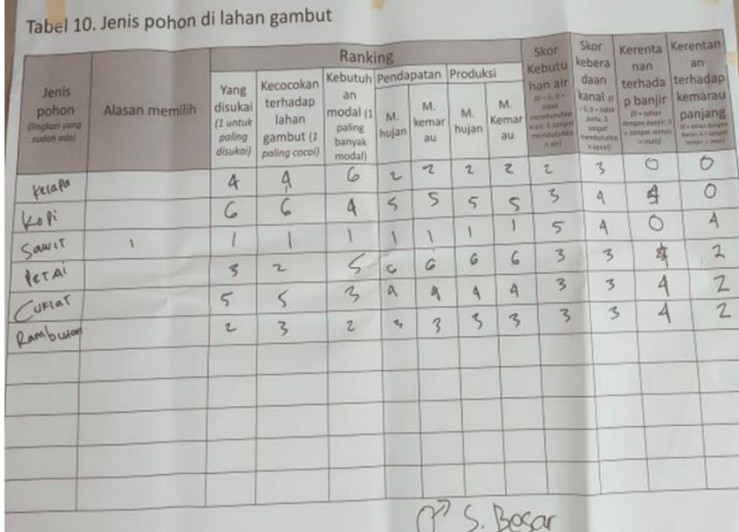

**Figure A1. Left: an example of the results of sorting the types of land-use in the Rejoso watershed by one of the local farmers, respectively: 1. Water use, 2. production cost, 3. income during wet season, 4. income during dry season, 5. preferences during wet season and 6. Preferences during dry season. For the water balance component, we derived from hydrological model parameterization. Right: example results of scoring of land-use type during focus group discussion with some farmers in the Pawan-Kepulu peatland to collect information about preferences, suitable to peat soil, production cost, income during wet and dry season, yield during wet and dry season, water use, and dependence on the present of canal, vulnerability rate to floods, and vulnerability rate to drought.**

## Appendix B. Guideline for facilitating H₂Ours game

| | |
|---|---|
| Overview | Simulation of the impact of land-use/cover change and water management on hydrological situation (water balance) |
| Objective | Knowledge sharing and decision making to support collaborative and collective actions among stakeholders |
| Benefits | 1. Players can explore many scenarios of land-use/cover and water management and see its impact to their hydrological situation |
| | 2. Players can feel the trade-off between economic and environment and explore the solutions |
| | 3. Players can learn about negotiation and collaboration |
| Duration | 2 hours (or around 8 – 10 rounds) |
| Number of players | 6 – 16 players |
| Material | 1. Board of the game |
| | 2. Land-use tokens |

3. Money tokens
4. Mini water balance simulation model
5. Water infrastructure token (optional)

**Game play**
1. Welcoming all the players and give a general introduction about the workshop and game/simulation
2. Selecting 2-3 people from players to act as public stakeholders whose role is responsible for the management of the whole watershed or peatlands by providing regulations or programs to prevent various environmental problems (optional)
3. Grouping the remaining the players into 3 groups (for watershed version) or 4 groups (for peatland version) to represent the farmers from different villages. During the game simulation, their goals are to live happily by fulfilling their needs.
4. Briefing players by giving explanations/definitions about the terminology that is often used in the game and building connection between the game properties with their actual situation so the decision made by the players can be very 605 close to their reality.
5. Introducing co-facilitator for each group who help calculation of economic resources  (optional)
6. Giving initial money to players (300 – 450 per group) and initial groundwater and surface water into the water balance simulation model
7. Starting round by asking player to decide their land-use system, then calculation of the economic and environmental 610 impact based on the (random) weather situation in that round
8. Repeat step 6 for round 2 and 3 as the warming up
9. Repeat step 6 for the rest of the rounds with additional scenarios, such as announcing regulation by government, providing payment for ecosystem program, etc. You can develop the scenarios based on the stakeholder perceptions of what they should do to restore the hydrological function through discussion or interview.
10. Debriefing session, by asking the player their strategies to achieve their goal and their feeling during the game simulation

**Appendix C. Criteria of Credibility, Salience and Legitimacy**

In this study, we refer to the criteria of credibility, salience and legitimacy by Belcher et al. (Belcher et al., 2016) in the 620 development and evaluation process of the H$_2$Ours game. Table C1 shows the criteria that we consider most relevant to represent the objective of the H$_2$Ours game to facilitate the transfer and sharing knowledge to support negotiation and collaboration among stakeholders. To use these criteria, we adjusted the definition of each criterion from the original definition (column 3) to a definition that meets the objectives of the H$_2$Ours (Column 4). Then, how we include each criterion in the development and evaluation process of the H$_2$Ours game is shown in columns 5 and 6.

 **Table C1: Criteria used to measure the credibility, salience and legitimacy of the H₂Ours game (adapted from Belcher et al. (2016).** **The criteria included were used to assess effectiveness in sharing understanding and encouraging collaboration for the H₂Ours game development and simulation.**

| No | Criteria | Original definition according to (Belcher et al., 2016) | Adjustment of to meet the objective of the H₂Ours game | How to include the criteria during the game design | Evaluation after game implementation |
|---|---|---|---|---|---|
| **CREDIBILITY** | | | | | |
| 1 | Clear problem definition | The research problem is clearly defined, researchable, grounded In the academic literature and relevant to the context | The issues handled in the H²Ours game are relevant to the actual situation | In diagnosis of the study area and issues using ARDI and DPSIR (Sect. 2.2) | Likert question: the possibility to apply the knowledge from the game in the reality |
| 2 | Clear objective | Research objectives are clearly stated | The objective of the H₂Ours game is clearly stated | In scope and objective (Sect. 2.3.1) | Likert question: understanding the objective of the game |
| 3 | Appropriate methods | Methods are fit to purpose and well-suited to answering the research questions and achieving the objectives. | Methods used are scientifically proven | The data and method used scientifically proven with some publications (Sect. 2.1 and Sect. 2.2) | There was no evaluation for this criterion after the game because we used scientifically proven method |
| 4 | Clearly presented argument | The movement from analysis through interpretation to conclusions is | The rules, dynamics, and interactions in the H₂Ours game built | Component interaction analysis based on ARDI and | Likert question: Understanding the rules of the game |

| | | | | | |
|---|---|---|---|---|---|
| | | transparently and logically described. Sufficient evidence is provided to clearly demonstrate the relationship between evidence and conclusions | based on logical interpretation supported by scientific data and methods | DPSIR (Sect. 2.2 and Sec. 2.3) | |

**SALIENCE/RELEVANCE**

| | | | | | |
|---|---|---|---|---|---|
| 5 | Socially relevant research problem | Research problem is relevant to the problem context | The problems/issues raised in the H$_2$Ours game are in accordance with the issues/problems in actual conditions | The information used based on participatory approach (referring some publication in Sect. 2.1) | Likert question: The possibility to apply the knowledge from the game in the reality |
| 6 | Engagement with problem context | Researchers demonstrate appropriate breadth and depth of understanding of and sufficient interaction with the problem context | The H$_2$Ours game is built by demonstrating the interaction of various elements (physical and social, interaction between stakeholders) that are shown in actual conditions. | Problem analysis based on DPSIR (Sect. 2.2) | Likert question: The possibility to apply the knowledge from the game in the reality |
| 7 | Explicit theory of change | The research explicitly identifies its main intended outcomes and | The H$_2$Ours game was built explicitly to facilitate | Set the purpose of the game in the game | Likert question: Gaining new knowledge from |

| | | | | | |
|---|---|---|---|---|---|
| | | how they are intended/expected to be realized and to contribute to longer-term outcomes and/or impacts | knowledge sharing and knowledge transfer to trigger collaborative action among various stakeholder | development proses (Sect. 2.3.1) | the game simulation |
| 8 | Relevant research objective and design | The research objectives and design are relevant, timely, and appropriate to the problem context, including attention to stakeholder needs and values | The objectives and design of the H$_2$Ours game are relevant to the problem context, including considering what the stakeholder needs and values | Based on ARDI and DPSIR analysis (Sect. 2.2 and 2.3) | Likert survey: 1. understanding the objective of the game 2. the possibility to apply the knowledge from the game in the reality |
| 9 | Appropriated project implementation | Research execution is suitable to the problem context and the socially relevant research objectives | The solutions in the H$_2$Ours game is generated based on activities that can be implemented in the actual condition | The solutions based on the multidisciplinary research (Sect. 2.1) | Likert question: the possibility to apply the knowledge from the game in the reality |
| **LEGITIMACY** | | | | | |
| 10 | Effective collaboration | Appropriate processes are in the place to ensure effective collaboration (e.g. clear and explicit roles and responsibility agreed upon, transparent and appropriate | The H$_2$Ours game shows transparency of rules, responsibilities, decision-making between game participants, so the | Simple game rules based on actual condition to facilitate participant game understanding (Sect. 2.3) | Using before and after survey using q-methodology to identify the change in |

| 11 | Genuine and explicit inclusion | Inclusion of diverse actors in the research process is clearly defined. Representation of actors' perspectives, values, and unique contexts is ensured through adequate planning, explicit agreements, Communal reflection, and reflexivity. | Involvement of various stakeholders during the process of $H_2$Ours game preparation, design, implementation, and evaluation to accommodate various perspectives, knowledge, values, interests of stakeholders | Involvement of various stakeholders in this study (Fig. 1) | Likert survey: the possibility to apply the knowledge from the game in the reality |

**Appendix D. $H_2$Ours game for peatland version (case study Pawan-Kepulu peatland)**

Based on some references, focus ground discussion and interview with various stakeholders in Pawan-Kepulu peatland, we found that this area experiences land and forest fires during the dry year (season) and flood during the wet year (season). Land cover conversion from forest to oil palm plantation and crop season has led massive canal construction to get better production. This situation makes this landscape drier during the dry year and vulnerable to fires.

The hydrological boundary of peatland is a Peatland Hydrological Unit (PHU) as an area between two rivers. Usually in this

landscape, there is a peat dome (the deepest peat area), an area surrounding the peart dome (i.e. buffering dome area) and an area with shallow peat. Villages are spread over the peat dome and the buffer zone with villages having different proportions of peat dome and buffer zone areas. However, for simplification, peat depth (including that of the peat domes) was distributed evenly between villages (Figure D1). However, for future game adaptations, the peat depth distributions in each village can be adjusted on the game board.

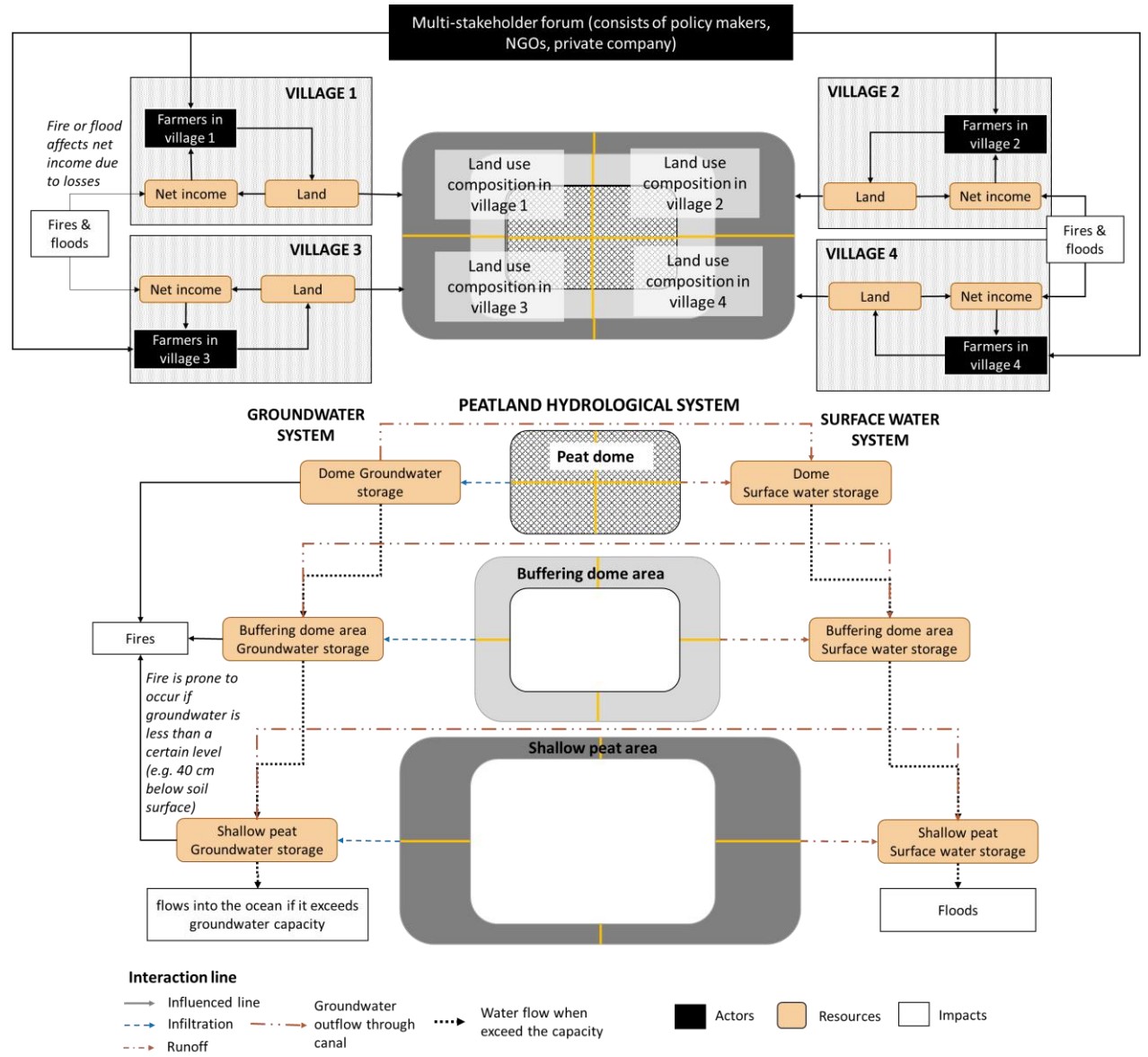

**Figure D1. Socio-hydrological model defined using the ARDI framework that was used to design the H₂Ours game for Pawan-Kepulu peatland. Interaction among actors and between actors to landscape influence land-use composition which affect the hydrological and economic situation, then its influences back to interaction.**

**Rules of game**

Based on measurement data, focus group discussion with local farmers and some references, we design the rules of the H₂Ours game for peatland version by combining six land-use options (all trees, all oil palm, oil palm + trees, oil palm + crop, all crop, and shrub/burned area) and three canal density options (without canal, low- and high-density canal) (Table D1 and D2).

**Table D1. Economic impacts in Pawan-Kepulu peatland version, the production cost in dome area +2/plot and in the buffering dome area +1/plot**

| Land-use options | Canal density options | Production cost/year in the shallow peat area (unit: money) | Income/year (unit: money) Wet Year | Dry year |
|---|---|---|---|---|
| All tree | Without | 1 | 0 | 3 |
| | Low | 1 | 0 | 3 |
| | High | 1 | 0 | 3 |
| All Oil palm | Without | 6 | 6 | 9 |
| | Low | 9 | 9 | 17 |
| | High | 12 | 17 | 25 |
| Oil palm + trees | Without | 3 | 4 | 6 |
| | Low | 4 | 6 | 9 |
| | High | 6 | 9 | 17 |
| Oil palm + seasonal crop | Without | 5 | 4 | 8 |
| | Low | 7 | 7 | 15 |
| | High | 10 | 12 | 20 |
| Crop | Without | 4 | 3 | 7 |
| | Low | 5 | 5 | 13 |
| | High | 8 | 7 | 15 |
| Shrub | Without | 0 | 0 | 0 |
| | Low | 0 | 0 | 0 |
| | High | 0 | 0 | 0 |

**Table D2. Environmental impacts in Pawan-Kepulu peatland version, we assumed during the dry year there is no runoff or infiltration (a = dome area, b = buffering dome area, c = shallow peat area, x = runoff (unit: ml), y = infiltration (unit: ml) and z = groundwater out flow through canal (unit: ml))**

| Land-use | Canal density | Dry year a | b | c | Wet Year a x | a y | a z | b x | b y | b z | c x | c y | c z |
|---|---|---|---|---|---|---|---|---|---|---|---|---|---|
| | | z | | | x | y | z | x | y | z | x | y | z |
| All tree | Without | 0 | 0 | 0 | 0 | 20 | 0 | 2.5 | 17.5 | 0 | 5 | 15 | 0 |
| | Low | 7.5 | 5 | 2.5 | 2.5 | 17.5 | 10 | 5 | 15 | 7.5 | 7.5 | 12.5 | 5 |
| | High | 15 | 10 | 5 | 5 | 15 | 20 | 7.5 | 12.5 | 15 | 10 | 10 | 10 |
| All Oil palm | Without | 0 | 0 | 0 | 10 | 7.5 | 0 | 12.5 | 5 | 0 | 15 | 5 | 0 |

| | | | | | | | | | | | | |
|---|---|---|---|---|---|---|---|---|---|---|---|---|
| | Low | 7.5 | 5 | 2.5 | 12.5 | 5 | 10 | 15 | 2.5 | 7.5 | 17.5 | 2.5 | 5 |
| | High | 15 | 10 | 5 | 15 | 2.5 | 20 | 17.5 | 1 | 15 | 17.5 | 1 | 10 |
| Oil palm + trees | Without | 0 | 0 | 0 | 5 | 15 | 0 | 7.5 | 12.5 | 0 | 10 | 10 | 0 |
| | Low | 7.5 | 5 | 2.5 | 7.5 | 12.5 | 10 | 10 | 10 | 7.5 | 12.5 | 7.5 | 5 |
| | High | 15 | 10 | 5 | 10 | 10 | 20 | 12.5 | 7.5 | 15 | 15 | 5 | 10 |
| Oil palm + seasonal crop | Without | 0 | 0 | 0 | 15 | 2.5 | 0 | 17.5 | 1 | 0 | 17.5 | 1 | 0 |
| | Low | 7.5 | 5 | 2.5 | 17.5 | 1 | 10 | 19 | 1 | 7.5 | 19 | 1 | 5 |
| | High | 15 | 10 | 5 | 17.5 | 1 | 20 | 19 | 1 | 15 | 19 | 1 | 10 |
| Crop | Without | 0 | 0 | 0 | 19 | 1 | 0 | 19 | 1 | 0 | 19 | 1 | 0 |
| | Low | 7.5 | 5 | 2.5 | 19 | 1 | 10 | 19 | 1 | 7.5 | 19 | 1 | 5 |
| | High | 15 | 10 | 5 | 19 | 1 | 20 | 19 | 1 | 15 | 19 | 1 | 10 |
| Shrub | Without | 0 | 0 | 0 | 20 | 0 | 0 | 20 | 0 | 0 | 20 | 0 | 0 |
| | Low | 7.5 | 5 | 2.5 | 20 | 0 | 10 | 20 | 0 | 7.5 | 20 | 0 | 5 |
| | High | 15 | 10 | 5 | 20 | 0 | 20 | 20 | 0 | 15 | 20 | 0 | 10 |

**Game Properties**

The component of the H$_2$Ours game for peatland version is similar with watershed version with modification in the board as the landscape and the land-use options (Fig. D2). The board is designed in such a way that it resembles a PHU with a dome in the middle, a buffering area around the dome and shallow peat on the outside. In the real simulation, we can add river and road to help player have a connection with their real situation.

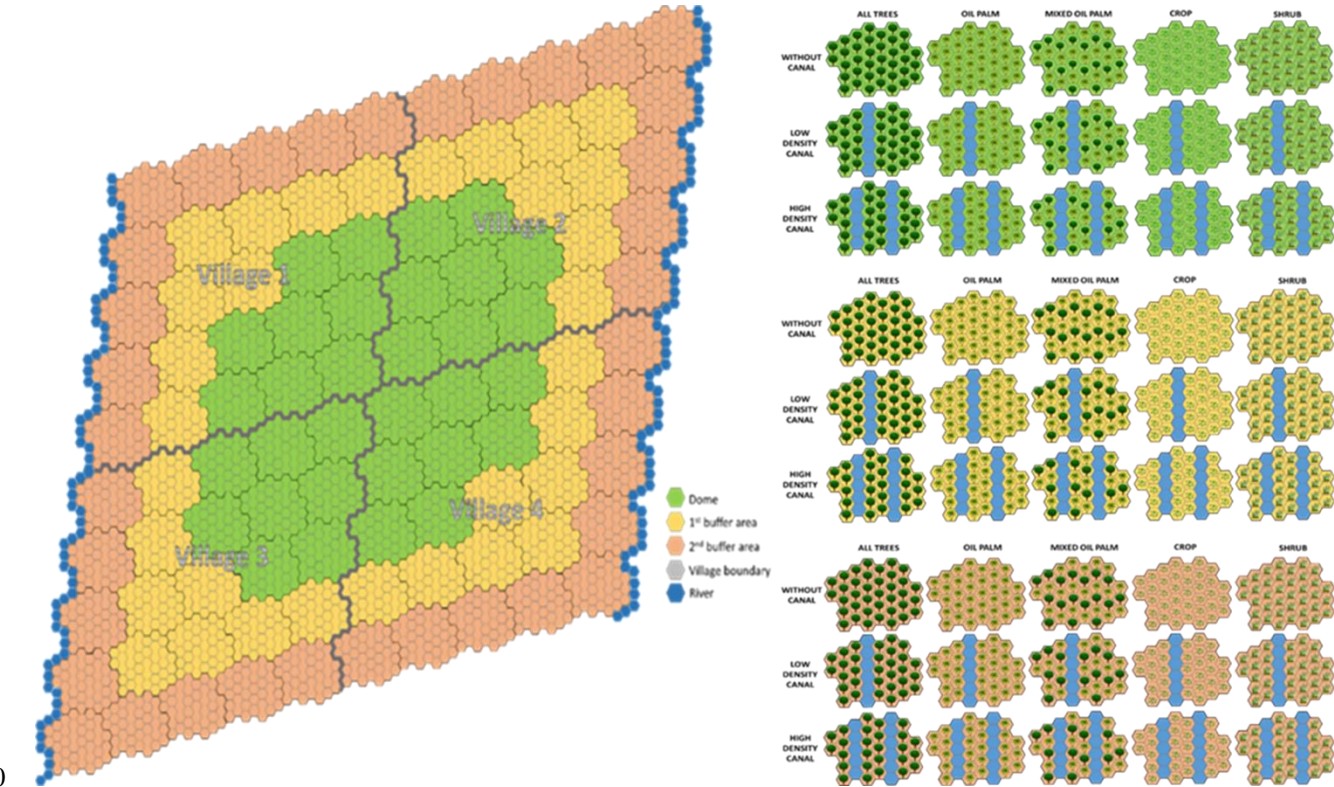

**Figure D2. Board of the H$_2$Ours game for peatland version that consist of dome area, buffering dome area and shallow peat area (left); and land-use option (all trees, all oil palm, oilpalm+trees, all crop and shrub) with various canal density (without canal low canal density and high canal density) for Pawan-Kepulu peatland area (right)**

Similar with the Rejoso watershed, the H$_2$Ours game for peatlands also has the same water balance miniature (Fig. D3). This

water balance model follows the hydrological system in Fig. D1. In the groundwater system, each tank has a fire vulnerable threshold. This threshold represents 40 cm below soil surface in its actual condition as stipulated by government regulations. If the groundwater in each zone is below this limit, then the area has the potential for fires which causes harm to the local community.

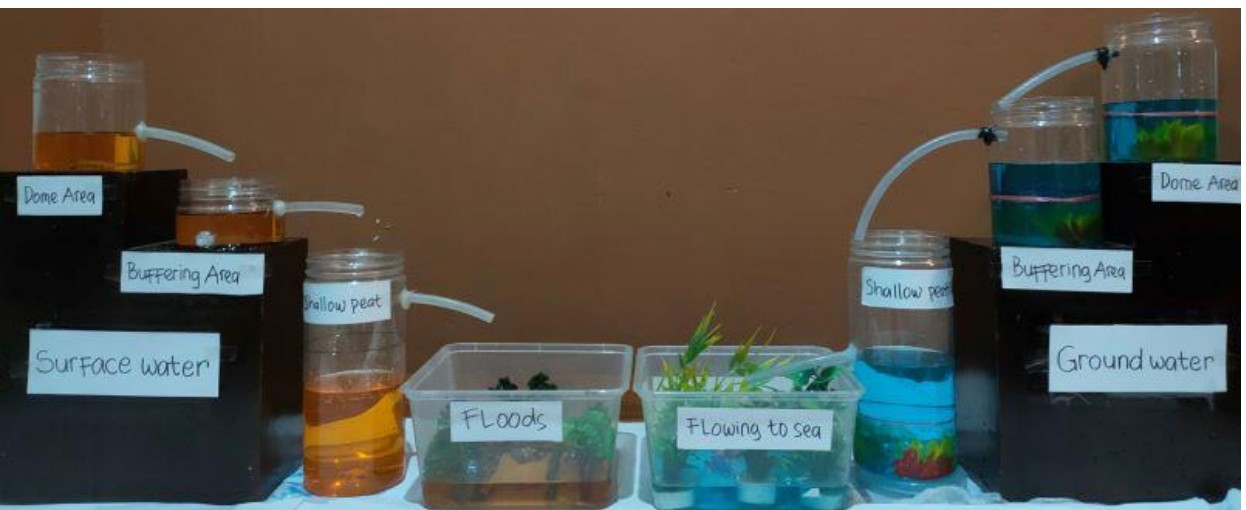

**Figure D3. Simple water balance model of the H₂Ours game in Pawan-Kepulu peatland to show the dynamics of changes in hydrological conditions as a result of the changes in land-use and canal density. The red line in each tank in the groundwater system represent the fire vulnerable threshold.**

In addition to the H₂Ours game for the peatland version, there is a peat infrastructure token in the form of canal blocking and fire fighters (Fig. D4). In the reality, the canal blocking blocks the canal to reduce/stop the groundwater outflow. In this game

simulation the canal blocking changes the land-use from high to low density canal or from low to without canal. The firefighter helps to prevent plot from fires during the dry year/season. However, providing canal blockings and firefighters cost some money.

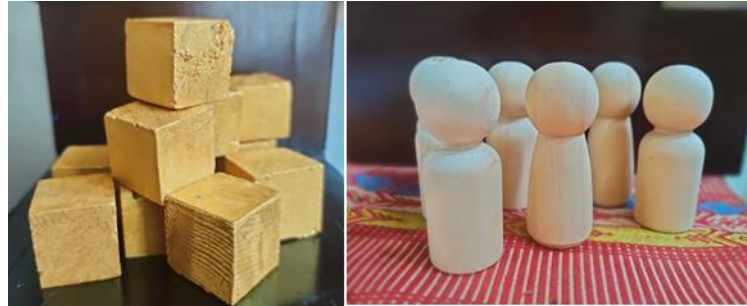

**Figure D4. Additional token as canal blocking (left) and firefighters (right) for H₂Ours game**

## Appendix E. Solution space of H₂Ours game in Rejoso watershed

The rules of the game determine the possible outcomes or 'solution space', within which the specific choices made by game participants are located. If all choices would be random (equal probability of all choices available), without response to the outcomes so far, a substantial variation in outcomes is possible. The primary outcomes of interest are the surface water flows

(rainfall not used as canopy interception evaporation or infiltration into the soil), and the groundwater flows (water infiltrating and not used for subsequent evapotranspiration), all depending on both land cover and rainfall.

A first question in defining this solution space is the number of random series that need to be evaluated to accurately estimate the frequency distributions of outcomes in various response parameters. We present data for 3, 10, 30, 100, 300 and 1000 iterations (Fig. E1 – E4) (each including 10 rounds and three zones, thus 30 land-use choices and 10 weather conditions (dry

or wet)). The actual game simulation was only done 4 times; therefore, the closest solution space is with 3 or 10 random values, which is have not sufficiently representative the distribution.  Based on Fig. E1 and E2, the solution space distribution pattern starts to appear in 30 random data sets. Therefore, to see the actual distribution of the farmer's decision making, at least we need 30 game simulations. Figure E3 and E4 show the relationship between economic conditions (money) and environment (groundwater and surface water) in the downstream area is more scatter compared to upstream and midstream.  However,

related to groundwater supply in downstream (Fig. E3), the more groundwater supply, and the higher the economic benefits obtained. On the contrary, the more runoff obtained from upstream and midstream (Fig. E4), it will decrease their economic benefits.

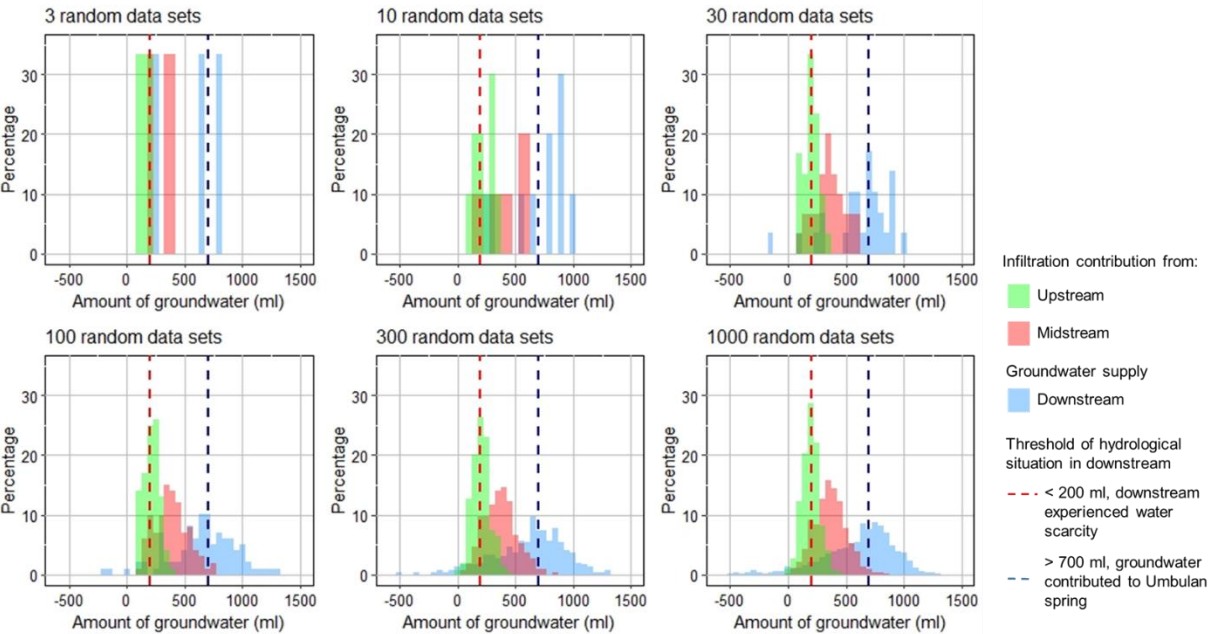

**Figure E1. Distribution of infiltration contribution from upstream and midstream and groundwater supply in downstream based**
**on simulation with the random value**

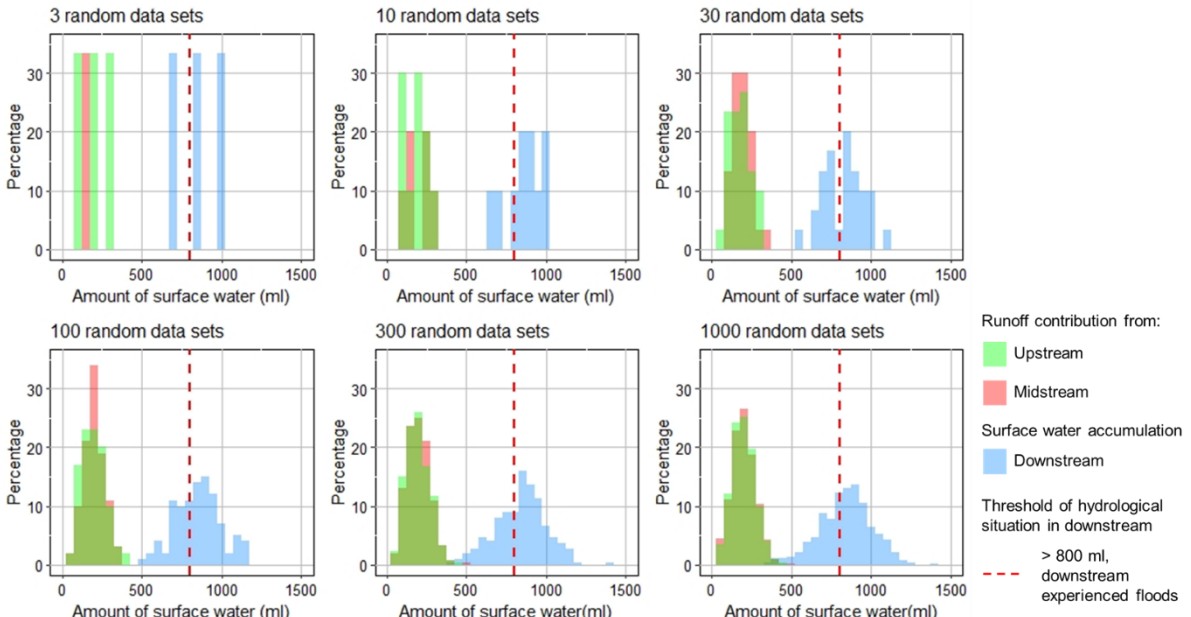

**Figure E2. Distribution of runoff contribution from upstream and midstream and surface water accumulation in downstream based on simulation with the random value**

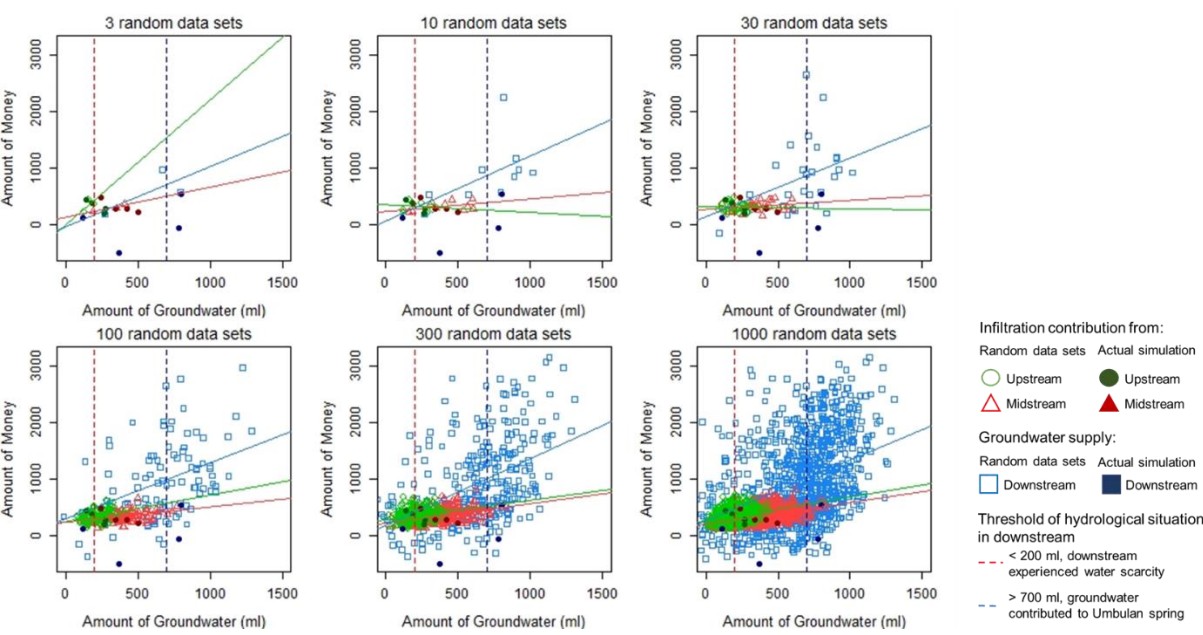

**Figure E3. Groundwater and economic conditions based on random value simulation and actual simulation**

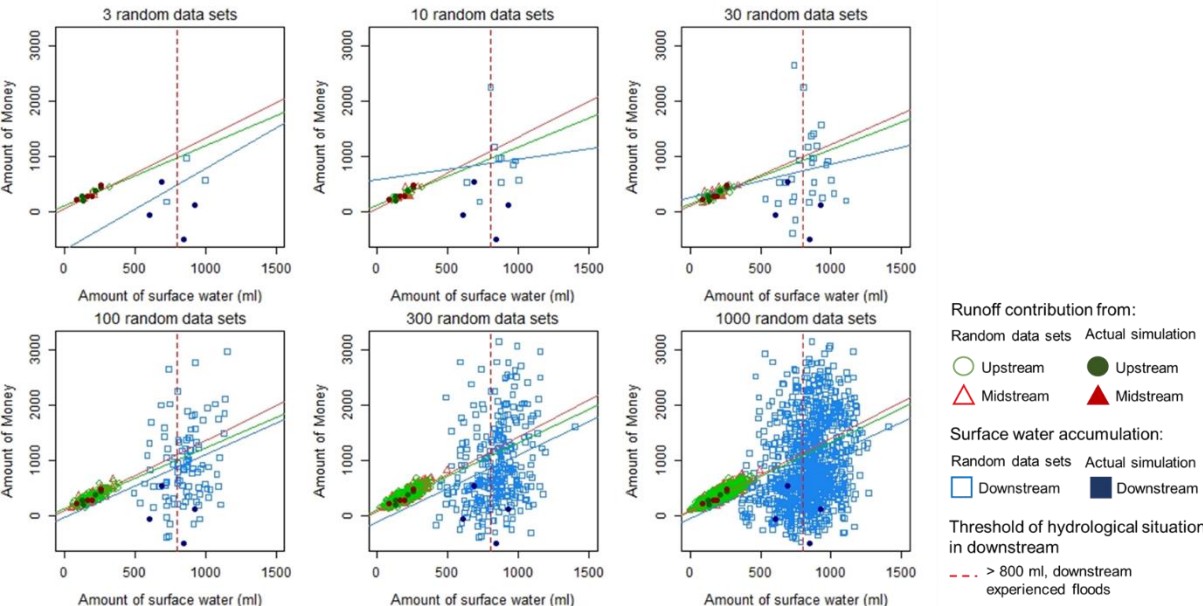

**Figure E4. Runoff and economic conditions based on random value simulation and actual simulation**

**Data availability**

All raw data can be provided by the corresponding authors upon request

**Competing interests**

The authors declare that they have no conflict of interest.

**Author contribution:**

LT, RRS and ALH designed the research project, LT, RRS and ENS designed the game, LT performed the game simulation and game analysed, LT, MvN, MPC and ENS wrote the manuscript, EP and BL gave input on the performance of game simulation in each case study area.

**Acknowledgment**

We appreciate the valuable input from colleagues of World Agroforestry (ICRAF) and Tropenbos Indonesia during the H₂Ours game development and game implementation. We would like to express our deep gratitude to students of the Merdeka University, Pasuruan, East Java and Tanjung Pura University, Pontianak, West Kalimantan, and Brawijaya University, Malang,

East Java, who have participated in H$_2$Ours game testing. This study was funded by Tropenbos Indonesia as part of the Working landscape and fires project, ICRAF as part of the Rejoso Kita project and INREF as part of the SESAM project.

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
