# Peer review of "The H2Ours game to explore Water Use, Resources and Sustainability: connecting issues in two landscapes in Indonesia"

_Hydrology and Earth System Sciences, 2023_

## Author Comment (AC1)

| No | Comments and Answers |
|---|---|
| 1 | This manuscript provides a very interesting review of the process used to develop "serious" games for social learning among stakeholders in two different socio-hydrological systems in Indonesia: a mountain slope leading to lowland paddies, and a peatland "dome". The games were developed based on hydrological studies using the Drivers, Pressure, State, Impact, and Responses (DPSIR) framework as well as analysis of the actors and stakeholders as well as the Actors, Resources, Dynamics, and Interaction (ARDI) framework. The introduction provides a good explanation of how games can influence action around water. It is particularly interesting how the study use the credibility, salience, and legitimacy framework to evaluate the two games.

Answer:

Thank you very much for your comments |
| 2 | The abstract concludes that "We provide clear steps in designing and adapting the game to another area…". This is the area where more is needed to deliver on this promise.  As currently laid out, there is not enough information about what kind of hydrological or socio-economic study is needed to adapt such a game to new contexts. Was this based on quick assessment or multi-year study of the two areas described?

Answer:

Thank you for sharing your concern about this.

The socio-hydrological studies as the basis for the game development were a combination of previous studies and rapid assessment during the game development process to obtain information that is not yet available.

We will add list of minimum information requirement and approach for game adaptation in the section 2.1 study area: "In order to diagnose issues and develop the $H_2$Ours game, the minimum information required from a socio-hydrological study includes: hydrological study (boundaries of the hydrological system, hydrological problems and efforts that should be done to overcome the causes and impacts of the problems), land cover study (types, changes and profitability), socio-economic study (village conditions, socio-economic issues, alternative livelihood options, institutional conditions). The approach taken to obtain this information is grouped based on local ecological knowledge (LEK), public ecological knowledge (PEK) and modeler/scientist ecological knowledge (MEK). The method was described as 'rapid hydrological appraisal' and tested in a number of Southeast Asan countries, and is part of a 'Negotiation-support' toolkit for learning landscape' (van Noordwijk et al., 2013; Jeanes et al., 2006).

Reference:

Jeanes, K., van Noordwijk, M., Joshi, L., Widayati, A., Farida, and Leimona, B.: Rapid Hydrological Appraisal in the Context of Environmental Service Rewards, World Agroforestry, Bogor, 56 pp., 2006.

Leimona, B., Khasanah, N., Lusiana, B., and ...: A business case: co-investing for ecosystem service |

provisions and local livelihoods in Rejoso watershed, 2018.

van Noordwijk, M., Lusiana, B., Leimona, B., Dewi, S., and Wulandari, D.: Negotiation-support toolkit for learning landscapes, edited by: van Noordwijk, M., Lusiana, B., Leimona, B., Dewi, S., and Wulandari, D., World Agroforestry Southeast Asia Regional Program, 2013.

What number of game simulations and what number of actual players are needed?

Answer:

We will add explanations in section 3.3 Game implementation to give an idea of the number of simulations and players: "In general, the simulation of $H_2$Ours game takes approximately two hours (excluding briefing and debriefing). For the Rejoso watershed version, the $H_2$Ours game consisted of 10 rounds with 6-12 players divided into 3 groups acting as local communities: upstream, midstream and downstream. The PHU Pawan-Kepulu version consisted of 8 rounds with 8-16 players divided into 4 groups, and players are asked to select their village name as first step of creating ownership. In both versions, an additional group of players consisting of 2-4 people can act as public stakeholders (government, companies, NGOs) and interact with the villages."

The other gap in the paper is a description of who played the game, and whether there were differences in how different types of players responded in the game, or in their interactions with each other.  For example, were all the players men?

Answer:

We will elaborate Section 2.4 Game implementation line 198: "In this study, we executed ten game sessions consisted of five sessions at each study areas. The five game sessions consisted of a session with a multi-stakeholder forum…… and four sessions with  farmer groups……While in each simulation in Pawan-Kepulu peatland, we invited 12-16 representatives of farmer group from four villages in that landscape. In the invitation, we let the group determine who would attend the simulation, provided that the group representatives were willing to hold discussions and exchange information with participants from other villages. During the game simulation, we asked……."

"For the four sessions with farmer groups we selected participants according to different criteria. For Rejoso watershed, we conducted two sessions with participants who had experience with a recent Payment for Ecosystem Services (PES) program (Leimona et al., 2018) and two sessions with participants from neighboring villages where the PES program was not active. Meanwhile, at PHU Pawan-Kepulu we conducted a game session with members of the village forest management unit, a session with members of an active farmer field school, and two sessions with people who are not members of village forest management unit and farmer field school. Follow-up manuscripts are planned that will provide further analysis of these contrasts in player background. (Tanika et al, in prep)"

Reference:

Leimona, B., Khasanah, N., Lusiana, B., and ...: A business case: co-investing for ecosystem service provisions and local livelihoods in Rejoso watershed, 2018.

Did the players from upstream play differently than those from downstream areas, even if they were not playing the parts of their own area?

Answer

We will integrate in the discussion section 4.2 Game evaluation and lessons learned:

"In this research, we invited people from upstream, midstream and downstream to play according to their location. We did not yet have the opportunity to conduct simulations with role-switching players, but this can be done and can provide further insights. Recommendations for further research that makes use of the $H_2$Ours game are to allow players to switch roles to see how responses and perceptions depend on such shifts."

| 3 | Line 173 says "profit is total income minus total capital". But if income is on an annual or seasonal basis, shouldn't that be the annualized cost of the capital (e.g. if there is a major outlay for pumps)? Or should that be "minus total costs" (which is what it says in the next sentence). In economic terms, there is a difference. |
| --- | --- |
| | Answer: |
| | Thank you for your correction. We will revise Line 173: "profit is revenue minus all financial expenses (taxes, cost, incidental cost, etc.)". |
| 4 | Figure 4B X axis is labelled Amount of groundwater), but shouldn't that be amount of surface water, or runoff? |
| | Answer: |
| | Yes, it should be 'amount of Surface water (ml)'. Thank you for your correction. I revised Figure 4B |
| 5 | Figure 4C and D, what does it say that the actual choices by the participants were so much below the simulated income, and mostly lower groundwater and runoff? |
| | Answer: |
| | We will provide a clearer explanation about the comparison between solution space and simulation results in the results section 2.3.4 Game solution space analysis and in the discussion. |
| | "The presence of relationship values between humans and nature and humans and other humans (relational values) influences decision making regarding natural resource management (van Noordwijk et al., 2020). Therefore, the decisions made by players during the game are influenced by various factors (e.g. interactions between players, game settings, level of player ecological knowledge, etc.) (Rodela R and Speelman, E.N., 2023, manuscript in review), whereas |

| | |
|---|---|
| | random decision making is used to build solution space. For example, when the upstream and midstream groups decided to maintain and improve their economic conditions, they caused a reduction in groundwater supply and increase flooding for downstream area, which caused the downstream group to pay for the losses it experiences. Apart from that, during the simulation the facilitator also provided Payment for Ecosystem Services (PES) scenarios (Appendix A, Game Play number 9: repeat step 6 for the rest of the rounds with additional scenarios such as providing payment for ecosystem services). This scenario offers downstream groups to contribute a certain amount of money to maintain more trees in the upstream and midstream. Therefore, the downstream groups economically always spend more money either as a loss due to the environmental consequences (floods or water scarcity) or as a prevention effort by joining the PES program.

Reference:

Rodela, R and Speelman, E.N., 2023, Serious game in natural resource management: steps toward assessment of their contextualized impacts, Current Opinion in Environmental Sustainability (under review process)

van Noordwijk, M., Speelman, E., Hofstede, G. J., Farida, A., Kimbowa, G., Geraud, G., Assogba, C., Best, L., and Tanika, L.: Sustainable Agroforestry Landscape Management :, Land, 9, 1–38, 2020. |
| 6 | Figure C1: how do the villages match the peat dome?

Answer:

Thank you for your question. Referring to Figure 2A (PHU Pawan-Kepulu), in reality the positions of peat domes are spread across several villages with different distribution. There are villages dominated by peat domes and buffering areas, and some of them are dominated by shallow peat. But, in the game board design, the distribution of peat depth (including peat domes) is distributed evenly in all villages. This is intended to facilitate replication for other locations.

We will add further explanation regarding peat dome distribution across the village in Appendix C line 505: "The hydrological boundary of peatland in peatland hydrological unit (PHU) as an area between two rivers. Usually inside this landscape we can find a dome (the deepest peat area), area surrounding dome (buffering dome area) and shallow peat area. The peat depth are spread across villages with different distribution. However, to facilitate the replication of the game, we designed the distribution of peat depth (including peat domes) is distributed evenly in all villages. Figure C1 shows the conceptual game model of H2Ours game for peatland version. However, for further simulation, we can design a game board with different peat depth distributions for each village" |
| 7 | The paper needs a good copy editor throughout ,including the appendix.

Answer:

Thank you for pointing this out. We will scrutinize the next version to be submitted. |

---

## Author Comment (AC2)

| No | Comments and Answers |
|---|---|
| 1 | In the manuscript, the authors provide a description of the H2Ours serious game developed and tested on two locations in Indonesia with an opinion on the possibility of adapting this game to other areas and conditions. While the reasons for developing such a game is clearly described and explained, the rules of the game and the flow of the game are not so clear for the reader. I found especially hard to follow so many subtitles in sections 2 and 3 (Methods and Results) that interrupt the reading flow and consequently the understanding of the game. Moreover, it is not clearly stated who should be the target group of players (students, farmers, general public etc.).

Answer,

Thank you for your comments. We have added more information according to your comments. We hope our response addresses your concerns and makes this manuscript clearer.

We provided more detailed explanations of roles (section 3.2.2) and rules (section 3.2.3). Please see

We added more information about the participants in the Section 2.4 (Game implementation)

"In this study, we executed ten game sessions with different participant groups with a total of 93 participants. The ten game sessions consisted of five sessions at each study areas. The five game sessions consisted of a session with a multi-stakeholder forum consisting of representative of governments, NGOs, private sectors, and universities to get ideas on regulations and programs that would be offered to local communities/farmers, and four session with farmer groups to implement the regulations and programs resulting from the game simulation with the multi-stakeholder forum. In each session with farmer groups in Rejoso watershed, we invited a total 9-12 representatives of farmer groups from upstream, midstream and downstream village to a meeting hall where all participants could still reach it. While in each simulation in Pawan-Kepulu peatland, we invited 12-16 representatives of farmer group from four villages in that landscape. In the invitation, we let the group determine who would attend the simulation, provided that the group representatives were willing to hold discussions and exchange information with participants from other villages. During the game simulation, we asked the invited farmers to behave as farmers in line with the position of their village in the landscape.

For the four sessions with farmer groups we selected participants according to different criteria. For Rejoso watershed, we conducted two sessions with participants who had experience with a recent Payment for Ecosystem Services (PES) program (Leimona et al., 2018) and two sessions with participants from neighboring villages where the PES program was not active. Meanwhile, at PHU Pawan-Kepulu we conducted a game session with members of the village forest management unit, a session with members of an active farmer field school, and two sessions with people who are not members of village forest management unit and farmer field school." |
| 2 | For me as the hydrologist, the content of section 2.3 about game solution space analysis is not described clearly enough, more specifically, how did you produce random choices (e.g., using some software, etc.). |

| | |
|---|---|
| | Answer:

Thank you for your input. We added more information about the process of producing solution space in section 2.3.4

"Solution space is defined as a set of all possible decisions made by players. The solution space of the game was explored based on the average of economic and environmental conditions obtained from 3, 10, 30, 100, 300 and 1000 games with random-choice. One random-choice game consisted of 10 rounds in which climate conditions and land use decisions made by players are completely random. The random-choice of land use and climate condition were generated in R, then simulated using Excel spreadsheet as an imitation of the real H2Ours game to calculate the economic and environmental conditions. In addition, we assessed the probability of outcomes within the solution space under random decision-making as a point of reference for the actual game implementation." |
| 3 | Also related to the rules and flow of the game, it is not clearly described how and when the models shown in Figure 6 and Figure C3 take place in the game. Please clarify.

Answer
Thank you for your concern. We revised text in the section 3.2.5 (Game Properties) to add more explanation about Figure 6:
"To make the game more interesting and stimulate engagement, we prepared some game materials such as a game board to represent the landscape, land-use tiles according to the existing and future land cover types, play money token, and water infrastructures token (Fig. 5). We also created water balance miniatures (Fig. 6) to demonstrate how surface water flows and becomes flood and water infiltration become ground water supply. Each round after calculating the economic condition and environmental conditions based on Table 3, we asked players to pay production costs, taxes, etc. and get income, incentives, etc. using play money and fill the water balance miniature with real water according to the produced surface water and groundwater. " |
| 4 | In line 119 it is not clear what kind of values represent discharges 5 and 3,5 m$^3$/s (average in the mentioned year, some long-term average, something else?). Please clarify.

Answer:

Thank you for question. The value represent the average throughout a year of the daily discharge. We changed the text to:
"Land conversion from agroforestry to intensive agriculture in the recharge areas (> 700 masl. upstream and midstream area) and massive groundwater extraction using artesian wells in the downstream area for rice field were thought to cause the reduced average (during a hydrological year) of daily discharge of the Umbulan spring, from 5 m$^3$/s (1980s) to 3.5 m$^3$/s (2017) " |
| 5 | Table 1 in my opinion is too long/big and similarly as multiple subsections break the reading flow. I would suggest adding some summary into the text and moving the table into appendices.

Answer:
Thank you for your suggestion. We moved table of criteria of credibility, salience and legitimacy to Appendix C, and replaced that table with short summary in the sub-section 2.5 p. 3) |

| | |
|---|---|
| | "From the long list of criteria (Belcher et al., 2016), we chose four credibility criteria, five salience criteria and two legitimacy criteria which we considered to be the most relevant for evaluating the H2Ours game to meet the objective of the game. Each of these criteria were included during the game design process and the evaluation after game implementation. We included those criteria during the game design using ARDI and DPSIR frameworks to structure socio-hydrological data and information based on the research findings from ICRAF and Tropenbos Indonesia (both already and to be published) to meet the criteria during the game development process. For evaluation after game implementation, we converted those criteria into several question ans statements for the q-method and Likert survey and asked all game participants to fill in the survey." |
| 6 | In line 231, a brief explanation about both methods, i.e. Likert scale and q-method, needs to be added.

Answer:
Thank you for your input. We added further explanations related to Likert scale and q-method. We elaborate the text in section 2.5 (p.3)
"In the Likert scale survey, we used five-point scales (strongly disagree, disagree, neutral, agree, and strongly agree) on six statements to ask about their feeling during the game, their understanding of the rules of the game, the length of the game simulation, new knowledge that they got from the game, and how close the game to their reality. We used q-method to capture participants' subjectivity regarding the relationship between vegetation, water and humans, the causes of socio-hydrological problems in their region and the factors that determine the success of hydrological condition restoration activities. To capture changes in their perceptions regarding these three questions, participants conducted the q-method before and after the game using the same questions and q sort statements. The results of the q-method will be presented in another paper along with their decision making, preferences, vision, collaborative and collective action. " |
| 7 | In Figure 4, Figure D1, Figure D2, and Figure D3 it is not clear what are presenting solid blue, green, and red lines. Additionally, "ml" in legends should be replaced with "masl". In relation to these figures, why are there different thresholds used (e.g., 200, 800) than explained in lines 114–116?

Answer:
Thank you for your suggestion. The legend is correct with 'ml' because it describes the amount of surface water and ground water (>200 ml). We will make the legend clearer.
We added more explanation about the threshold value (e.g. 200, 800) in the section 3.2.3 because this is related to flooding, water shortages and land fires as a result of the impacts from Table 3 and Table D2.
"When the total of surface water in the downstream of Rejoso watershed and in the shallow peat of Pawan-Kepulu peatland exceeding its capacity (>800 ml) during the rainy season, it caused flooding. When the groundwater exceeding its capacity (>700 ml), it flowed to springs in Rejoso watershed and to sea in Pawan-Kepulu peatland. But, when the groundwater was less than its requirement (<200 ml), it caused water shortages in the Rejoso Watershed and potential fires in Pawan-Kepulu peatland. These environmental impacts decreased the community income. As the consequence of this situation, they might not have enough money to manage their land, |

| | |
|---|---|
| | buy food or pay taxes in the next round of the game. The multi-stakeholder forums with their limited budget could then choose to help them by providing financial help or making regulations/programs to prevent these environmental problems. Through this gameplay, we expected to promote all actors to work together and collaborate to achieve their goals." |
| 8 | In Figure 7, there are missing y-axis titles. Please add.

Answer:
Thank you for your correction. we revised the Figure 7 |
| 9 | Please use en-dash throughout the manuscript in case of ranges and periods. E.g., 0–100 masl instead 0 – 100 masl.

Answer:

Yes, Thank you. we checked all the manuscript and revised it |
| 10 | Appendices should be mentioned in text in the order in which they appear, e.g., Appendix A before Appendix B.

Answer:

Thank you for your input. We have change the order of the Appendices |

Reference:

Belcher, B. M., Rasmussen, K. E., Kemshaw, M. R., and Zornes, D. A.: Defining and assessing research quality in a transdisciplinary context, Res. Eval., 25, 1–17, https://doi.org/10.1093/reseval/rvv025, 2016.

Cash, D., Clark, W. C., Alcock, F., Dickson, N., Eckley, N., and J�ger, J.: Salience, Credibility, Legitimacy and Boundaries: Linking Research, Assessment and Decision Making, SSRN Electron. J., https://doi.org/10.2139/ssrn.372280, 2002.

Jeanes, K., van Noordwijk, M., Joshi, L., Widayati, A., Farida, and Leimona, B.: Rapid Hydrological Appraisal in the Context of Environmental Service Rewards, World Agroforestry, Bogor, 56 pp., 2006.

Keles, H., Yayla, O., Tarinc, A., and Keles, A.: The Effect of Environmental Management Practices and Knowledge in Strengthening Responsible Behavior: The Moderator Role of Environmental Commitment, Sustain., 15, https://doi.org/10.3390/su15021398, 2023.

Leimona, B., Khasanah, N., Lusiana, B., and ...: A business case: co-investing for ecosystem service provisions and local livelihoods in Rejoso watershed, 2018.

Medema, W., Mayer, I., Adamowski, J., Wals, A. E. J., and Chew, C.: The potential of serious games to solve water problems: Editorial to the special issue on game-based approaches to sustainable water governance, Water (Switzerland), 11, https://doi.org/10.3390/w11122562, 2019.

van Noordwijk, M., Lusiana, B., Leimona, B., Dewi, S., and Wulandari, D.: Negotiation-support toolkit for learning landscapes, edited by: van Noordwijk, M., Lusiana, B., Leimona, B., Dewi, S., and Wulandari, D., World Agroforestry Southeast Asia Regional Program, 2013.

van Noordwijk, M., Speelman, E., Hofstede, G. J., Farida, A., Kimbowa, G., Geraud, G., Assogba, C., Best, L., and Tanika, L.: Sustainable Agroforestry Landscape Management :, Land, 9, 1–38, 2020.

Rodela, R and Speelman, E.N., 2023, Serious game in natural resource management: steps toward assessment of their contextualized impacts, Current Opinion in Environmental Sustainability (under review process)

van Noordwijk, M., Speelman, E., Hofstede, G. J., Farida, A., Kimbowa, G., Geraud, G., Assogba, C., Best, L., and Tanika, L.: Sustainable Agroforestry Landscape Management :, Land, 9, 1–38, 2020.

---

## Author Comment (AC3)

| No | Comments and Answers |
|---|---|
| 1 | Thank you for sharing this interesting work with me. First of all, please let me apologize if I have not properly understood those elements of the paper on which I give comments, or if my perception was wrong.

The paper documents the process of designing and applying a serious game based on a system simulation model. The application of serious games as a contribution to sustainable development receives increasing attention also raising high expectations. The presented work documents a game design process and intends to provide evidence on outcomes from playing the game. It can help to better understand the potentials and limitations of the approach and can meaningfully inform future serious game design processes. I strongly support the publication of the paper. Having said this, I want to share some thoughts which I hope can help to improve the paper. |
| 2 | *The study has two objectives:*

1. *Develop an adaptable serious game*

2. *Assess if game facilitates knowledge transfer and sharing, and supports negotiation and coordination.*

*I am not sure whether the paper really addresses these two objectives. Either rethink your objectives or structure the paper more strongly around the ones you formulated. You documented the game development process which I consider an important contribution. It is difficult for me to judge how easy it is to adapt the developed game to a different context. The paper does not really make this clear.*

Answer:

Thank you. We rephrased the two objectives to be closer to what we actually discuss
1. Develop a serious game that is adaptable to different socio-hydrological contexts and issues
2. Clarify how such game can facilitate knowledge transfer and sharing, and can support negotiation and coordination by context-specific use

*I am even more concerned about the second research question. The methods section does not explain the approach in sufficient detail. An input-output process is mentioned as methodology without explaining it. You state that information was collected throughout the game development and implementation process. How was this information collected and analyzed? You mention a post-game survey. Please provide more information on: 1) who was interviewed and how have respondents been selected, 2) how many respondents were interviewed, 3) how was the survey implemented (also addressing the strong risk of social desirability bias), 4) when was it conducted?*

Answer: |

You have similar question with reviewer 2 (RC 2). We add more information about how we collect the data for the evaluation survey in the section 2.5 about the game evaluation.

"From the long list of criteria (Belcher et al., 2016), we chose four credibility criteria, five salience criteria and two legitimacy criteria which we considered to be the most relevant for evaluating the H2Ours game to meet the objective of the game. Each of these criteria were included during the game design process and the evaluation after game implementation. We included those criteria during the game design using ARDI and DPSIR frameworks to structure socio-hydrological data and information based on the research findings from ICRAF and Tropenbos Indonesia (both already and to be published) to meet the criteria during the game development process. For evaluation after game implementation, we converted those criteria into several question ans statements for the q-method and Likert survey and asked all game participants to fill in the survey.

In the Likert scale survey, we used five-point scales (strongly disagree, disagree, neutral, agree, and strongly agree) on six statements to ask about their feeling during the game, their understanding of the rules of the game, the length of the game simulation, new knowledge that they got from the game, and how close the game to their reality. We used q-method to capture participants' subjectivity regarding the relationship between vegetation, water and humans, the causes of socio-hydrological problems in their region and the factors that determine the success of hydrological condition restoration activities. To capture changes in their perceptions regarding these three questions, participants conducted the q-method before and after the game using the same questions and q sort statements. The results of the q-method will be presented in another paper along with their decision making, preferences, vision, collaborative and collective action."

*For answering the second research question, can you share information on the decisions of players in the game? Do you see patterns and changes over the rounds? Can you share information on what has been discussed during the game sessions?*

Answer:

As the objective of this manuscript is to document the development and adaptation process of the H2Ours game, so in this paper we focus on the development process.
We plan to make a series of publications about H2Ours game in Rejoso watershed and Pawan-kepulu peatland:
1. H2Ours game development and adaptation process (this manuscript)
2. Result of the H2Ours game.
We decided to present the results of the H2Ours game in different manuscript to have more space to present the conditions of the research location which are closely related to how players make decisions regarding land and water management, including changes in perception and knowledge.
We mentioned in section 2.5 (game evaluation), we mentioned: "The results of the q-method will be presented in another paper along with their decision making, preferences, vision, collaborative and collective action"

| 3 | *In this context, I recommend giving more weight to Section 3.3. Interesting statements are made regarding the game dynamics. It could be combined with some statements included in the* |

| | |
|---|---|
| | *discussion (e.g. that setting up the game has built emotions). The value of the information depends, however, on the transparency of how information was generated.*

Answer:

In accordance with the objective of this manuscript which focuses on the H2Ours development process, the results shown in this manuscript are more about the form of the H2Ours game itself. Implementation results are presented in section 3.3 is only indicative result. We believe that the dynamics of the game results correspond to the character of the socio-hydrological system and changes in participants' perceptions can be an interesting article which provide more space for explanations. So we will deliver more detailed result in another paper. |
| 4 | *On a conceptual level, I am not convinced that collecting data on credibility, salience/relevance, and legitimacy can answer the second research question. This would at least require stronger justification. Going a step further, I do not think that the survey questions are adequate indicators for Belcher's et al. criteria. For instance, asking a player whether she sees the possibility to apply the knowledge acquired in the game does not provide information on how inclusive the process has been. I would propose a more differentiated approach regarding how you provide evidence on different criteria. Players will not be able to give meaningful answers to some of them. For instance, by playing the game, a player does not need to be aware of the underlying theory of change. I would move Table 1 to an appendix to make space for a better explanation of the methodology and the key results.*

Answer:

In the theory of change related to this game, the desired changes refer to the 4 stages of the knowledge to action chain (understanding, commitment, operationalization and innovation). Based on the definitions of credibility, salinity and legitimacy in section 2.5: "From the game development perspective, credibility refers to whether a game is built based on scientifically reliable knowledge, including the data and methods used to build the game. Salience refers to how far the game can show the relevance of goals, rules and finding to the actual situation. Finally, legitimacy refers to how the participant can accept the game by relating the game simulation to their actual situations (Cash et al., 2002)." In order to achieve the second objective of this paper, to clarify how games can facilitate the sharing and transfer of knowledge, we want to ensure relevance of game situations to the reality (salience), the acceptance that game conditions resemble their conditions (legitimacy), and supported by appropriated data and methods (credibility).

Thank you for your suggestion. We moved table of criteria of credibility, salience and legitimacy to Appendix C, and replaced that table with short summary in the sub-section 2.5, please see our response to Reviewer 2 number 5. |
| | *A clear theory of change is one of the evaluation criteria and it needs to be expressed in a more explicit way. In lines 42ff, the authors argue that knowledge supports commitment (intention?) supports responsibility, supports innovation. It is not clear which type of knowledge, commitment, responsibility, and innovation the game intends to support. The learning outcomes are formulated rather vaguely. In the model framework, the authors include the assumption that income drives decisions. In the discussion, you argue that the enabling environment is the key* |

*driver and no behavioral change can be expected if regulations etc. are not adjusted. Considering all these thoughts and assumptions, what did you intend to change by letting stakeholders play the game?*

Answer:
Thank you for your concern. We agree that we have not written down specifically what form of knowledge we want to convey in this form of game. We revise the sentence in section 4.2 to:

" As hydrological problems are usually quite complex and fundamental, their solution requires quite a long time for integrated planning, which requires all stakeholders to have the bigger picture (Medema et al., 2019). The H2Ours game tries to simplify the space so that it makes it easier for players to be aware of the conditions of neighboring players and to get the entire perspective of socio-hydrological problems. Improving player knowledge by looking at socio-hydrological problems in a broader context encourages responsible behavior towards the environment which is directly proportional to commitment (Keles et al., 2023).

We also revised the third lesson learned to reduce miss interpretation:

"Third, based on the evaluation and debriefing results, even if they stated they can apply the ideal collaborative actions that are simulated in the game, in real life, it still needs to build the enabling condition that supports it (e.g. regulation, integrated planning strategies, etc.). Because the game is a simplification of the system, forms of collaborative action can be discussed directly by the players. In real conditions, the parties who are expected to collaborate may not find it easy to meet and discuss. Therefore, the implementation of the collaboration in real life requires guidelines and mechanisms that can be followed by the parties to achieve common goals, and the commitment, as referred to in the four knowledge-to-action chains, still cannot be carried out directly unless there are enabling conditions that exceed space limitations"

| 5 | *Based on the provided information, it is difficult to understand the structure of the game. Key aspects are not explained, such as:* |
| --- | --- |
| | *Number of players per session; who was selected to play the game (relevant also for theory of change)* |
| | *Types of roles; in section 3.2.2 is described that there are two roles namely communities and a multi-stakeholder forum. But then one finds statements that there are upstream and downstream actors, different villages etc. Is the role definition linked to the diagnosis? Who in Table 2 creates pressure? Who is involved in the responses and dynamics? Are the relevant actors included as roles? Such game development decisions are critical for the theory of change.* |
| | Answer: |
| | Thank you for your concern. We elaborated Section 2.4 (Game implementation) to add more information about the players in this study. Please see our response to reviewer 2 number 1 about the game participants |
| | We added more explanation on section 3.2.2 (roles): |
| | "Based on the stakeholder identification survey in Rejoso Watershed and Pawan-Kepulu peatland, we defined two key roles for this game, namely local communities and a multi-stakeholder forum. In the Rejoso watershed, local communities can be grouped into people who |

live in the upstream village, midstream village and downstream village based on the village elevation. Meanwhile in Pawan-Kepulu peatland, local communities can be grouped into four groups of people living in 4 neighboring villages (Village 1 – Village 4). The local communities represent land owners and their goal is to live happily by fulfilling their needs (food and taxes). The multi-stakeholder forum is a forum that consists of public stakeholders (NGOs, university, etc.) and policy makers (local governments). Their goals is to prevent natural disasters (water scarcity and floods in Rejoso watershed, and fires and floods in Pawan-Kepulu peatland). The $H_2Ours$ game brings the various interests of these actors together and shows how they make their decisions regarding the management of land and water resources to meet their economic and environmental expectations."

In the section 2.3.2, we already mentioned that how we defined the roles based on the ARDI framework:

"According to the ARDI framework (Sect. 2.2), we defined the roles in this game based on the main actors involved in water management. Related to these roles, we designed goals that players must achieve during each simulation based on discussions and interviews with the related stakeholders according to their actual goal in the reality".

We revised section 3.1 (result: Diagnosis of the study areas and issues) based on your input:

"Based on the results from the DPSIR and ARDI analyses, we found that the Rejoso Watershed and the Pawan-Kepulu peatland have similarities in the socio-hydrology context (Table 2). Expectations on better economic conditions led local communities to changes in land cover, and excessive extraction of water resources (groundwater) caused disruption of the water balance. This disruption resulted in local communities and multi-stakeholder forum experience various hydrological problems, such as water shortages (or decreasing the groundwater level) and flooding. However, these two sites are also different regarding their hydrological contexts, such as hydrological boundaries, topography, and water management, and interactions among stakeholders and landscape (Fig. 3, Appendix C). Two solutions (responses) were identified to restore hydrological functions in watersheds and peatlands, namely better land use management and (ground) water management (Table 2; component 7-Response) by the local communities and multi-stakeholder forum".

| 6 | *Which decisions can be taken by which player role? Table 3 and Figure 3 give the impression that only crop and forestry related choices can be made. But then it is mentioned that players can block channels. The choices of the multi-stakeholder forum are formulated very unclear and are not included in Figure 3.* |
|---|---|
| | *How could the forum make regulations or programs to prevent environmental programs.* |
| | Answer: |
| | Thank you for your concern. We added a paragraph in section 3.1 (p.2) related to Figure 3 and Figure D1 to explain what the types of decisions they make that affect their landscape. |
| | "In more detail, the interaction between stakeholders and the landscape is represented by the types of decisions taken by local communities and the multi-stakeholder forum regarding their |

landscape. Local communities (farmer from upstream, midstream and downstream village in Rejoso Watershed and farmers from Village 1 - Village 4 in Pawan-kepulu peatland) have the authority to make decisions regarding their land which consists of land use types and water management types (artesian wells in Rejoso watershed and canal blocking in Pawan-Kepulu peatland). Multi-stakeholder forums have authority over regulations and programs applied to local communities to achieve their goals. Multi-stakeholder forum can refer to their existing document for the regulation and program or based on the game session with multi-stakeholder forum (Section 2.4) "

We revised Section 3.2.3. (Rules) to give more explanation about the option of land use types and options of regulation and program by multi-stakeholder forum.

"At the start of the game, all group of players (local communities and multi-stakeholder forum) received a limited amount of money. Community members were asked to manage their land by arranging the land use type combination and water management in their area with the money provided, while multi-stakeholder forum was asked to run programs or to help reduce the local community's economic problems. Once players decided on how they would manage their land or community programs, the economic and environmental rules linked to those land use decisions were applied (Table 3). These rules then defined the dynamics of the economic and environmental conditions (Table 3, and for the Pawan-Kepulu peatland in Appendix D)."

*Please also better explain how you decided on the players' choices in the game.*

Answer:

We invite the players through village leaders and farmer group leaders, and let them decided who would attend the simulation. However, in the invitation we gave the criteria of the participants. We already added this information in the section 2.4 (game implementation). Please see our response to reviewer 2 about the participants and their criteria.

*In the discussion is mentioned that the game could not capture the complexity of the system and that few issues had to be selected to be included in the game. Please describe the process from diagnosis to selecting the issues to be included in the game. Such game development decisions are again critical for the theory of change.*

Answer:

In the discussion we mentioned that based on several publications, these two research studies have more complex socio-hydrological problems than played in the H2Ours game. In the beginning, we wanted to include all problems in one game such as erosion, landslides and sedimentation. However, during the game development process, especially in the preparation of rules and the quantification systems, the game becomes very complex. Considering the background of the targeted participants and the game duration, we decided to only highlight one hydrological problem. We determined which issues to raise based on the national priority issue in that location. We add more information on how we chose the issue for the game in the discussion.

| | |
|---|---|
| | *Are decisions taken collectively (a group represents one role and decides based on some agreement) or individually, creating collective outcomes. In line 433 is stated that the game demonstrated the value of collective action. How is collective action implemented in the game?*

Answer:

Yes, a group represent one role. When we started the game, the facilitator asked to choose a leader to represent their group.

*Did the players receive real money? If yes, how was the payout determined? Did it depend on the game dynamics as is common in behavioral games? If not, make clear in Section 3.2.3 that you talk about a game endowment/play money.*

Answer:

I am sorry, i do not really understand your question. I assume your question is about the use of real or play money and how to determine it. We revised section 3.2.5 (game properties) to explain about the play money and water balance miniature.

"To make the game more interesting and stimulate engagement, we prepared some game materials such as a game board to represent the landscape, land-use tiles according to the existing and future land cover types, play money token, and water infrastructures token (Fig. 5). We also created water balance miniatures (Fig. 6) to demonstrate how surface water flows and becomes flood and water infiltration become ground water supply. Each round after calculating the economic condition and environmental conditions based on Table 3, we asked players to pay production costs, taxes, etc. and get income, incentives, etc. using play money and fill the water balance miniature with real water according to the produced surface water and groundwater." |
| 7 | *A more comprehensive explanation of outcomes resulting from choices would be important for understanding the game. For instance, in Table 3, can you provide likelihoods of flooding, water shortage, and fire events for different land use options?*

Answer:

We add more explanation in Section 3.2.3 explaining how we determined environmental impacts (floods, fires and water shortage).

"When the total of surface water in the downstream of Rejoso watershed and in the shallow peat of Pawan-Kepulu peatland exceeding its capacity (>800 ml) during the rainy season, it caused flooding. When the groundwater exceeding its capacity (>700 ml), it flowed to springs in Rejoso watershed and to sea in Pawan-Kepulu peatland. But, when the groundwater was less than its requirement (<200 ml), it caused water shortages in the Rejoso Watershed and potential fires in Pawan-Kepulu peatland. These environmental impacts decreased the community income. As the consequence of this situation, they might not have enough money to manage their land, buy food or pay taxes in the next round of the game. The multi-stakeholder forums with their limited budget could then choose to help them by providing financial help or making regulations/programs to prevent these environmental problems. Through this gameplay, we expected to promote all actors to work together and collaborate to achieve their goals. " |

*The choices in Table 3 indicate that unsustainable options provide highest income. At the same time, your model (Figure 3) works with the assumption that income mainly drives choices. How do you address the risk of unsustainable learning outcomes? A player's answer in the evaluation survey that she can apply the game learnings could mean that she learnt that an upstream-all-crop strategy is best.*

Answer:

I understand your question and concern. When we talk about the best scenario at the landscape level involving many stakeholders, there may be a conflict between the best scenario for whom and for what. Hence, the H2Ours game tries to convey this issue through one of its objectives related to a shared understanding by bringing the socio-hydrological system as one system. By understanding condition, we expect that a commitment will emerge to overcome their common problems through negotiation and coordination between stakeholders. This game was designed to resemble actual conditions, so it could provide an illustration of the solution to their real problems. However, to achieve sustainable learning outcome, other efforts are still needed such as implementation (adapting what they learnt from game simulation to real life), monitoring and evaluation.

| | |
|---|---|
| 8 | I do not understand the step of the game solution space analysis. What does it mean that the solution space was explored based on 3, 10, 30, 100, 300, and 1000 games with random choice? How did these 1000 games differ? Just explain this better for readers not familiar with your modelling and game development approach.

Answer:

We elaborate on section 2.3.4 to explain in more detail about the solution space and the method used to generate it
"Solution space is defined as a set of all possible decisions made by players. The solution space of the game was explored based on the average of economic and environmental conditions obtained from 3, 10, 30, 100, 300 and 1000 games with random-choice. One random-choice game consisted of 10 rounds in which climate conditions and land use decisions made by players are completely random. The random-choice of land use and climate condition were generated in R, then simulated using Excel spreadsheet as an imitation of the real H2Ours game to calculate the economic and environmental conditions. In addition, we assessed the probability of outcomes within the solution space under random decision-making as a point of reference for the actual game implementation." |
| 9 | *Line 74f: I challenge the statement that models and games have rarely been combined. There are numerous examples for instance on the website https://games4sustainability.org/gamepedia/. I am also aware of the following study: Lohmann, D., Falk, T., Geissler, K., Blaum, N., & Jeltsch, F. (2014). Determinants of semi-arid rangeland management in a land reform setting in*

Answer:

Thank you – our text was indeed inaccurate here. We corrected to: |

| | |
|---|---|
| | "Socially interactive games and models that explore larger spatial and temporal horizons have complementary strengths. As reviewed in Villamor et al. 2023, games and models can 1) seek a conceptual triangulation of representing the processes behind complex realties, 2) strive for numerical consistency between games and empirical models, 3) use games in the development of scenario models, or 4) use models in the design of games that trigger players to learn by experiencing manageable complexity. As an example of the letter, Lohman et al. (2014) designed and tested model-based role-plays with Namibian land reform beneficiaries, simulating 10 years of rangeland management.  In this paper, we explore the feasibility of transforming a hydrological model into a serious game to provide socio-hydrological dynamics to stakeholders with diverse backgrounds in order to develop restoration plans." |
| 10 | *Line 80f: As a complementary approach, the authors may want to consider providing guidance on context conditions which need to be fulfilled for meaningfully using the games. Also, can you estimate the size of the areas which share socio-hydrological conditions which are featured in the presented games?*

Answer:

The current games rely on hydrological units without major unrecognized in- or outflows. In many cases groundwater flows into or out of agriculturally used areas may be relevant uncertainties in the use of water balance models. In both the mountain-to-sea and peatland hydrological units groundwater flow issues are internalized in the game. In many landscapes the social systems don't match the hydrological boundaries; spatially explicit social subsystems, such as upper, middle and lower watersheds can be accommodated, while interactions with social systems beyond the game boundaries may be represented by economic (supply, demand, prices) parameters and their (exogenous) fluctuations. |
| 11 | *Line 185f: On which basis do you assume that decisions of players in the game represent their real-life decisions? Did you ask them to do so? Why do you need this assumption?*

Answer:

Sari et al. (2023) tested for the plot-level FORCES game whether farmers land use system and tree choices matched their actual conditions outside the game. For H2OURS we don't have the same detailed comparison yet. However, during the game simulation process we see whether players play according to their actual conditions based on the choice of land cover they choose or not. We compared how they selected land use during the game simulation with the actual land use. Part of the game properties, we provide land use types according to their actual conditions and some other land uses that may be options in the future. We did not directly ask them to play according to reality, but in the beginning we created conditions in accordance with their reality.  In this study, we expected participants to play according to their reality at the beginning because one of the goals of this game is to facilitate negotiation and collaboration between stakeholders to achieve the hydrological restoration target. |

| 12 | *Can you please better link your discussion and conclusions to the objectives of the paper and the presented results.* |
| --- | --- |
| | Answer: |
| | Thank you for your input. We will look again the link between objectives, discussion and conclusion |
| 13 | *Now that you have developed the game, can you share some thoughts on how your games could be applied to support sustainable development? Who could facilitate games with which target groups? What would be required to implement the game with a larger group of stakeholders? Ideally this way forward should be linked to the theory of change.* |
| | Answer: |
| | My research has a significant correlation to SDG 6 (Clean water and sanitation), SDC 13 (climate action) and SDG 15 (Life on land), but also supports SDG 16 (Pace, Justice and strong institutions) and SDG 8 (Decent work and economic growth). The complexity of water resource management arises from many stakeholders who have different backgrounds, interests, targets and knowledge have to make decisions related to the same system. Because they do not know that they are working in the same system, they do not know that their decisions might affect other stakeholders. Therefore, changes in point of view of stakeholders to realize that they are actually working in the same socio-ecological system are needed. H2Ours game simplifies a landscape that has the same hydrological system, allows stakeholders to see the same system and makes it easier for them to collaborate to solve problems or achieve common goals. |
| | For broad stakeholder groups, it requires identification of key stakeholders who are participants in the player. In the game development process, we mentioned in the roles of the game. The facilitator's ability and knowledge in leading the game also plays an important role in briefing, implementation and debriefing |
| 14 | *I hope my comments help to improve the paper. I think my concerns can be addressed in the frame of a major revision. I very much hope to see an improved version of the paper being published.* |
| | Answer: |
| | Thank you and we really appreciate your time. |

Reference:

Belcher, B. M., Rasmussen, K. E., Kemshaw, M. R., and Zornes, D. A.: Defining and assessing research quality in a transdisciplinary context, Res. Eval., 25, 1–17, https://doi.org/10.1093/reseval/rvv025, 2016.

Cash, D., Clark, W. C., Alcock, F., Dickson, N., Eckley, N., and J�ger, J.: Salience, Credibility, Legitimacy and Boundaries: Linking Research, Assessment and Decision Making, SSRN Electron. J., https://doi.org/10.2139/ssrn.372280, 2002.

Jeanes, K., van Noordwijk, M., Joshi, L., Widayati, A., Farida, and Leimona, B.: Rapid Hydrological Appraisal in the Context of Environmental Service Rewards, World Agroforestry, Bogor, 56 pp., 2006.

Keles, H., Yayla, O., Tarinc, A., and Keles, A.: The Effect of Environmental Management Practices and Knowledge in Strengthening Responsible Behavior: The Moderator Role of Environmental Commitment, Sustain., 15, https://doi.org/10.3390/su15021398, 2023.

Leimona, B., Khasanah, N., Lusiana, B., and ...: A business case: co-investing for ecosystem service provisions and local livelihoods in Rejoso watershed, 2018.

Medema, W., Mayer, I., Adamowski, J., Wals, A. E. J., and Chew, C.: The potential of serious games to solve water problems: Editorial to the special issue on game-based approaches to sustainable water governance, Water (Switzerland), 11, https://doi.org/10.3390/w11122562, 2019.

van Noordwijk, M., Lusiana, B., Leimona, B., Dewi, S., and Wulandari, D.: Negotiation-support toolkit for learning landscapes, edited by: van Noordwijk, M., Lusiana, B., Leimona, B., Dewi, S., and Wulandari, D., World Agroforestry Southeast Asia Regional Program, 2013.

van Noordwijk, M., Speelman, E., Hofstede, G. J., Farida, A., Kimbowa, G., Geraud, G., Assogba, C., Best, L., and Tanika, L.: Sustainable Agroforestry Landscape Management :, Land, 9, 1–38, 2020.

Rodela, R and Speelman, E.N., 2023, Serious game in natural resource management: steps toward assessment of their contextualized impacts, Current Opinion in Environmental Sustainability (under review process)

van Noordwijk, M., Speelman, E., Hofstede, G. J., Farida, A., Kimbowa, G., Geraud, G., Assogba, C., Best, L., and Tanika, L.: Sustainable Agroforestry Landscape Management :, Land, 9, 1–38, 2020.

---

## Author Comment (AC4)

| No | Comments and Answers |
|----|----------------------|
| 1 | This manuscript provides a very interesting review of the process used to develop "serious" games for social learning among stakeholders in two different socio-hydrological systems in Indonesia: a mountain slope leading to lowland paddies, and a peatland "dome". The games were developed based on hydrological studies using the Drivers, Pressure, State, Impact, and Responses (DPSIR) framework as well as analysis of the actors and stakeholders as well as the Actors, Resources, Dynamics, and Interaction (ARDI) framework. The introduction provides a good explanation of how games can influence action around water. It is particularly interesting how the study use the credibility, salience, and legitimacy framework to evaluate the two games.

 Answer:

 Thank you very much for your comments and interest in our use of the credibility, salience, and legitimacy framework |
| 2 | *The abstract concludes that "We provide clear steps in designing and adapting the game to another area…". This is the area where more is needed to deliver on this promise.  As currently laid out, there is not enough information about what kind of hydrological or socio-economic study is needed to adapt such a game to new contexts. Was this based on quick assessment or multi-year study of the two areas described?*

 Answer:

 Thank you for requesting further clarity here. The socio-hydrological studies that formed the basis for the game development were a combination of previous studies and rapid assessments using established procedure to obtain information that is not yet available and needed during the game development process. We have added a list of minimum information requirements and approach for game adaptation in the section 2.1 study area:

 "In order to diagnose issues and develop the H$_2$Ours game, three knowledge system are explored and contrasted: local ecological knowledge (LEK), public ecological knowledge (PEK) and modeler/scientist ecological knowledge (MEK). Overlaps indicate starting points for the further work, gaps and contrasts issues that need attention. Minimum information requirements are based on topography and drainage systems, climate (rainfall, potential evapotranspiration), land use/cover types, their reliance on labor and inputs and expected economic returns, phenology and relative evapotranspiration. A coherent set of methods was described as a 'Negotiation-support' toolkit for learning landscapes'; it was developed in and tested for a number of Southeast Asan countries (van Noordwijk et al., 2013). The 'rapid hydrological appraisal' (Jeanes et al., 2006) is part of this toolbox. The basic understanding of the socio-hydrological context can be obtained by combining a hydrological study (climate variation in space and time, boundaries of the hydrological system, hydrological problems such as floods and droughts, and current efforts to deal with the causes and impacts of the problems), land cover study (typology, main locally relevant types, recent land cover change and life-cycle profitability estimates), socio- |

economic study (village conditions, socio-economic issues, alternative (incl. non-land-based livelihood options, institutional history)."

*What number of game simulations and what number of actual players are needed?*

Answer:

As for any tool, the way it is used depends on specific targets that the user may have. Games can be used for raising awareness (agenda setting), for increased and shared understanding of how things work (hydrologically, socially, and in interactions), for framing issues and setting goals to deal with them and or for exploring means of implementation for achieving these goals. The number of game replications, choice of players in homogenous or explicitly mixed groups and the balance between 'experience' (letting many stakeholders play and draw their own conclusions) and 'evidence' (documenting and further analyzing game outcomes by researchers) can be decided by a game user.

For the specific examples of the adaptation of H2OURS to two landscapes in Indonesia, we added explanation in section 3.3 Game implementation:

"The simulation of $H_2$Ours game takes approximately two hours (excluding briefing and debriefing). For the Rejoso watershed version, the $H_2$Ours game consisted of 10 rounds with 6-12 players divided into 3 groups acting as local communities: upstream, midstream and downstream. The PHU Pawan-Kepulu version consisted of 8 rounds with 8-16 players divided into 4 groups, and players are asked to select their village name as first step of creating ownership. In both versions, an additional group of players consisting of 2-4 people can act as public stakeholders (government, companies, NGOs) and interact with the villages."

A future paper will further analyze the specific results obtained in the two landscapes.

*The other gap in the paper is a description of who played the game, and whether there were differences in how different types of players responded in the game, or in their interactions with each other. For example, were all the players men?*

Answer:

We will elaborate Section 2.4 Game implementation line 198:

"In this study, we executed ten game sessions with different participant groups with a total of 93 participants. The ten game sessions consisted of five sessions at each study areas. The five game sessions consisted of a session with a multi-stakeholder forum consisting of representative of governments, NGOs, private sectors, and universities to get ideas on regulations and programs that would be offered to local communities/farmers, and four session with farmer groups to implement the regulations and programs resulting from the game simulation with the multi-stakeholder forum. In each session with farmer groups in Rejoso watershed, we invited a total 9-12 representatives of farmer groups from upstream, midstream and downstream village to a meeting hall where all participants could still reach it. While in each simulation in Pawan-Kepulu peatland, we invited 12-16 representatives of farmer group from four villages in that landscape. In the invitation, we let the group determine who would attend the simulation, provided that the

| | |
|---|---|
| | group representatives were willing to hold discussions and exchange information with participants from other villages. During the game simulation, we asked the invited farmers to behave as farmers in line with the position of their village in the landscape." |
| | "For the four sessions with farmer groups we selected participants according to different criteria. For Rejoso watershed, we conducted two sessions with participants who had experience with a recent Payment for Ecosystem Services (PES) program (Leimona et al., 2018) and two sessions with participants from neighboring villages where the PES program was not active. Meanwhile, at PHU Pawan-Kepulu we conducted a game session with members of the village forest management unit, a session with members of an active farmer field school, and two sessions with people who are not members of village forest management unit and farmer field school. Follow-up manuscripts are planned that will provide further analysis of these contrasts in player background. (Tanika et al, in prep)" |
| | *Did the players from upstream play differently than those from downstream areas, even if they were not playing the parts of their own area?* |
| | Answer |
| | We will integrate in the discussion section 4.2 Game evaluation and lessons learned: |
| | "In this research, we invited people from upstream, midstream and downstream to play according to their location. We did not yet have the opportunity to conduct simulations with role-switching players, but this can be done and can provide further insights. Recommendations for further research that makes use of the H$_2$Ours game are to allow players to switch roles to see how responses and perceptions depend on such shifts." |
| 3 | *Line 173 says "profit is total income minus total capital". But if income is on an annual or seasonal basis, shouldn't that be the annualized cost of the capital (e.g. if there is a major outlay for pumps)? Or should that be "minus total costs" (which is what it says in the next sentence). In economic terms, there is a difference.* |
| | Answer: |
| | We will revise Line 173: "profit is revenue minus all financial expenses (taxes, cost, incidental cost, etc.). The underlying economic analysis applied a life-cycle perspective to the various land use systems, annualizing discounted future cost and benefit flows". |
| 4 | *Figure 4B X axis is labelled Amount of groundwater), but shouldn't that be amount of surface water, or runoff?* |
| | Answer: |
| | Yes, it should be 'amount of Surface water (ml)'. Thank you for your correction. I revised Figure 4B |
| 5 | *Figure 4C and D, what does it say that the actual choices by the participants were so much below the simulated income, and mostly lower groundwater and runoff?* |

| | |
|---|---|
| | Answer:

We will provide a clearer explanation about the comparison between solution space and simulation results in the results section 2.3.4 Game solution space analysis and in the discussion.

"The presence of relationship values between humans and nature and humans and other humans (relational values) influences decision making regarding natural resource management (van Noordwijk et al., 2020). Therefore, the decisions made by players during the game are influenced by various factors (e.g. interactions between players, game settings, level of player ecological knowledge, etc.) (Rodela R and Speelman, E.N., 2023, manuscript in review), whereas random decision making is used to build solution space. For example, when the upstream and midstream groups decided to maintain and improve their economic conditions, they caused a reduction in groundwater supply and increase flooding for downstream area, which caused the downstream group to pay for the losses it experiences. Apart from that, during the simulation the facilitator also provided Payment for Ecosystem Services (PES) scenarios (Appendix A, Game Play number 9: repeat step 6 for the rest of the rounds with additional scenarios such as providing payment for ecosystem services). This scenario offers downstream groups to contribute a certain amount of money to maintain more trees in the upstream and midstream. Therefore, the downstream groups economically always spend more money either as a loss due to the environmental consequences (floods or water scarcity) or as a prevention effort by joining the PES program. |
| 6 | *Figure C1: how do the villages match the peat dome?*

Answer:

Referring to Figure 2A (PHU Pawan-Kepulu), in reality the positions of peat domes are spread across several villages with different distribution. There are villages dominated by peat domes and buffering areas, and some of them are dominated by shallow peat. But, in the game board design, the distribution of peat depth (including peat domes) is distributed evenly in all villages. This is intended to facilitate replication for other locations.

We will add further explanation regarding peat dome distribution across the village in Appendix C "The hydrological boundary of peatland in peatland hydrological unit (PHU) as an area between two rivers. Usually inside this landscape we can find a dome (the deepest peat area), area surrounding dome (buffering dome area) and shallow peat area. The peat depth are spread across villages with different distribution. However, to facilitate the replication of the game, we designed the distribution of peat depth (including peat domes) is distributed evenly in all villages. Figure C1 shows the conceptual game model of H2Ours game for peatland version. However, for further simulation, we can design a game board with different peat depth distributions for each village" |
| 7 | *The paper needs a good copy editor throughout ,including the appendix.*

Answer: |

| | Thank you for pointing this out. We will scrutinize the next version to be submitted. |
|---|---|

---

## Author Response (AR1)

Dear Editor and Reviewer,

First of all I would like to thank the reviewers for their inputs and advice on this manuscript, in a positive spirit of interest in the topic. They emphasized a number of gaps that I needed to fill, clarify and elaborate to provide better flow and information. In general, I have included most of the reviewer comments in the new version of the manuscript. Below I provide point to point comments which correspond to the revised version of the manuscript.

Thank you

Lisa Tanika, on behalf of all authors

**Response to Reviewer 1 (RC1)**

| No | Comments and Answers |
|---|---|
| 1 | This manuscript provides a very interesting review of the process used to develop "serious" games for social learning among stakeholders in two different socio-hydrological systems in Indonesia: a mountain slope leading to lowland paddies, and a peatland "dome". The games were developed based on hydrological studies using the Drivers, Pressure, State, Impact, and Responses (DPSIR) framework as well as analysis of the actors and stakeholders as well as the Actors, Resources, Dynamics, and Interaction (ARDI) framework. The introduction provides a good explanation of how games can influence action around water. It is particularly interesting how the study use the credibility, salience, and legitimacy framework to evaluate the two games.

**Answer:**

*Thank you very much for your comments and interest in our use of the credibility, salience, and legitimacy framework* |
| 2 | a. The abstract concludes that "We provide clear steps in designing and adapting the game to another area…". This is the area where more is needed to deliver on this promise.  As currently laid out, there is not enough information about what kind of hydrological or socio-economic study is needed to adapt such a game to new contexts. Was this based on quick assessment or multi-year study of the two areas described?

**Answer:**

*Thank you for requesting further clarity here. The socio-hydrological studies that formed the basis for the game development were a combination of previous studies and rapid assessments using established procedure to obtain information that is not yet available and needed during the game development process. We have added a list of minimum information requirements and approach for game adaptation in the section 2.2 study area:*

"For systems diagnosis and developing the $H_2$Ours game, the minimum required information composed of: hydrological information (to define boundaries of the hydrological system, hydrological problems and efforts that should be done to overcome the causes and impacts of the problems, rainfall, potential evapotranspiration), land cover information (typology, main |

locally relevant types, recent land cover change and life-cycle profitability estimates), and socio-economic information (village conditions, socio-economic issues, alternative livelihood options, institutional conditions). These information were collected using the Rapid Hydrological Appraisal (RHA) approach, which has been used and tested in a number of Southeast Asian countries (van Noordwijk et al., 2013; Jeanes et al., 2006)(van Noordwijk et al., 2013; Jeanes et al., 2006). In this approach, the information were grouped based on local ecological knowledge (LEK), public ecological knowledge (PEK) and modeller/scientist ecological knowledge (MEK). Mapping these different knowledge systems showed overlap, gaps and contrasts that provided starting points for further exploration"

b.  What number of game simulations and what number of actual players are needed?

**Answer:**

*As for any tool, the way it is used depends on specific targets that the user may have. Games can be used for raising awareness (agenda setting), for increased and shared understanding of how things work (hydrologically, socially, and in interactions), for framing issues and setting goals to deal with them and or for exploring means of implementation for achieving these goals. The number of game replications, choice of players in homogenous or explicitly mixed groups and the balance between 'experience' (letting many stakeholders play and draw their own conclusions) and 'evidence' (documenting and further analyzing game outcomes by researchers) can be decided by a game user.*

*For the specific examples of the adaptation of $H_2OURS$ to two landscapes in Indonesia, we added explanation in section 3.3 Game implementation:*

"The game session with the $H_2Ours$ game takes approximately two hours (excluding briefing and debriefing). For the Rejoso watershed version, the two hours of game session consisted of 10 rounds with 6-12 players divided into 3 groups (or 2-4 people per group) acting as local communities: upstream, midstream, and downstream. The Pawan-Kepulu peatland version, the two hours game session consisted of 8 rounds with 8-16 players divided into 4 groups, and players are asked to select their village name as first step of creating ownership. In both versions, an additional group of players consisting of 2-4 people can act as public stakeholders (government, companies, NGOs) and interact with the villages."

c.  The other gap in the paper is a description of who played the game, and whether there were differences in how different types of players responded in the game, or in their interactions with each other.  For example, were all the players men?

**Answer:**

*We will elaborate Section 2.4 Game implementation (P.1 and P.2):*

"In this study, we executed ten game sessions which a total of 93 people participating, with five sessions in each of the study areas. All game sessions in Rejoso watershed were held in October 2021, while in Pawan-Kepulu peatland were held in August 2022. In each study area, a first one game session was organized with members of a multi-stakeholder forum consisting of representative of governments, NGOs, private sectors, and universities to get ideas on

regulations and programs that would be offered to farmer communities, and four game session were organized with farmer groups to explore the implementation of the regulations and programs resulting from the game session with the multi-stakeholder forum.

For each game session, we invited in total of 9-12 representatives of farmer groups from upstream, midstream and downstream village in Rejoso watershed, and 12-16 representatives of four villages in the Pawan-Kepulu peatland. In the invitation, we let the group determine who would attend the simulation, provided that the group representatives were willing to hold discussions and exchange information with participants from other villages. For the four sessions with farmer groups, we grouped participants according to different criteria to get a variety of decisions. For the Rejoso watershed, we conducted two sessions with participants who had experience with a recent Payment for Ecosystem Services (PES) program (Leimona et al., 2018) and two sessions with participants from neighbouring villages where the PES program was not active. For the Pawan-Kepulu peatland, we conducted a game session with members of the village forest management unit, a session with members of an active farmer field school, and two sessions with people who are not members of village forest management unit and farmer field school. Game sessions took place in a central location in each of the landscapes to allow easy access for all participants. During the game session, the participants were asked we asked to play the game with the role of a farmers from their location within the landscape."

*A future paper will further analyze the specific results obtained in the two landscapes.*

Did the players from upstream play differently than those from downstream areas, even if they were not playing the parts of their own area?

**Answer**

*We integrated in the discussion section 4.2, as part of the recommendation for the future works*

"In this research, we invited participants from upstream, midstream, and downstream to play from the perspective from their location in the landscape. We expect that this impacted how the game was played. We intend to explore the impacts of role switching by asking farmers to play the role of a farmer in another location in the landscape."

| 3 | Line 173 says "profit is total income minus total capital". But if income is on an annual or seasonal basis, shouldn't that be the annualized cost of the capital (e.g. if there is a major outlay for pumps)? Or should that be "minus total costs" (which is what it says in the next sentence). In economic terms, there is a difference. |
|---|---|
| | **Answer:** |
| | *We revised to Line 185*: "…., where profit is revenue minus all financial expenses (taxes, cost, incidental cost, etc.). The underlying economic analysis applied a life-cycle perspective to the various land-use systems, annualizing discounted future cost and benefit flows.". |
| 4 | Figure 4B X axis is labelled Amount of groundwater), but shouldn't that be amount of surface water, or runoff? |
| | **Answer:** |

| | |
|---|---|
| | *Yes, it should be 'amount of Surface water (ml)'. Thank you for your correction. we revised Figure 4B* |
| 5 | Figure 4C and D, what does it say that the actual choices by the participants were so much below the simulated income, and mostly lower groundwater and runoff?

**Answer:**

*We provided a clearer explanation about the comparison between solution space and simulation results in the results section 3.2.4 Game solution space analysis.*

"The presence of relationship values between humans and nature and humans and other humans (relational values) influences decision making regarding natural resource management (van Noordwijk et al., 2023, 2020). Therefore, the decisions made by players during the game are influenced by various factors (e.g. interactions between players, game settings, level of player ecological knowledge, etc.) (Rodela and Speelman, 2023), whereas random decision making is used to build solution space.  For example, when the upstream and midstream groups decided to maintain and improve their economic conditions, they caused a reduction in groundwater supply and increase flooding for downstream area, which caused the downstream group to pay for the losses it experiences. Apart from that, during the game session the facilitator also provided PES scenarios (Appendix B, Game Play number 9: repeat step 6 for the rest of the rounds with additional scenarios such as providing payment for ecosystem services). This scenario offers downstream groups to contribute a certain amount of money to maintain more trees in the upstream and midstream. Therefore, the downstream player groups always spend more money than the mid- and upstream player groups either as a loss due to the environmental consequences (floods or water scarcity) or due to their efforts to prevent negative impacts by joining the PES program." |
| 6 | Figure C1: how do the villages match the peat dome?

**Answer:**

*Referring to Figure 2A (PHU Pawan-Kepulu), in reality the positions of peat domes are spread across several villages with different distribution. There are villages dominated by peat domes and buffering areas, and some of them are dominated by shallow peat. But, in the game board design, the distribution of peat depth (including peat domes) is distributed evenly in all villages. This is intended to facilitate replication for other locations.*

We added further explanation regarding peat dome distribution across the village in Appendix D: "The hydrological boundary of peatland is a Peatland Hydrological Unit (PHU) as an area between two rivers. Usually in this landscape, there is a peat dome (the deepest peat area), an area surrounding the peart dome (i.e. buffering dome area) and an area with shallow peat. Villages are spread over the peat dome and the buffer zone with villages having different proportions of peat dome and buffer zone areas. However, for simplification, peat depth (including that of the peat domes) was distributed evenly between villages (Figure D1). However, for future game adaptations, the peat depth distributions in each village can be adjusted on the game board." |
| 7 | The paper needs a good copy editor throughout ,including the appendix. |

| | |
|---|---|
| | **Answer:** |
| | *Thank you for pointing this out. We scrutinized the revised version to be submitted.* |

**Response to Reviewer 2 (RC2)**

| No | Comments and Answers |
|---|---|
| 1 | In the manuscript, the authors provide a description of the H2Ours serious game developed and tested on two locations in Indonesia with an opinion on the possibility of adapting this game to other areas and conditions. While the reasons for developing such a game is clearly described and explained, the rules of the game and the flow of the game are not so clear for the reader. I found especially hard to follow so many subtitles in sections 2 and 3 (Methods and Results) that interrupt the reading flow and consequently the understanding of the game. Moreover, it is not clearly stated who should be the target group of players (students, farmers, general public etc.). |
| | **Answer** |
| | *Thank you for your comments. We have added more information according to your comments. We hope our response addresses your concerns and makes this manuscript clearer.* |
| | *We provided more detailed explanations of on how to define the role and rules in Section 2.3.2 and section 2.3.3, while the roles and rules of the H2Ours game can be found in Section 3.2.2, Section 3.2.3 and Appendix D as the result of this manuscript.* |
| | *We added more information about the participants in the Section 2.4 (Game implementation), please see our response to RC1 number 2b and 2c.* |
| 2 | For me as the hydrologist, the content of section 2.3 about game solution space analysis is not described clearly enough, more specifically, how did you produce random choices (e.g., using some software, etc.). |
| | **Answer:** |
| | *Thank you for your input. We added more information about the process of producing solution space in section 2.3.4* |
| | "The purpose of game solution space is to define the outcomes of all possible choices made by players in the game (Speelman et al., 2014). The solution space of the $H_2$Ours game was explored based on the average of economic and environmental outcomes obtained from 3, 10, 30, 100, 300 and 1000 games with random choice. One random-choice game consisted of 10 rounds in which climate conditions and land-use decisions made by players were completely random. The random-choice of land-use and climate condition were generated in R, then simulated using Excel spreadsheet as an imitation of the real $H_2$Ours game to calculate the economic and environmental conditions. In addition, we assessed the probability of outcomes within the |

| | |
|---|---|
| | solution space under random decision-making as a point of reference for the actual game implementation. " |
| 3 | Also related to the rules and flow of the game, it is not clearly described how and when the models shown in Figure 6 and Figure C3 take place in the game. Please clarify.

**Answer**
*Thank you for your concern. We revised text in the section 3.2.5 (Game Properties) to add more explanation about Figure 6:*
"To make the game engaging, we prepared game materials such as a game board to represent the landscape, land-use tiles according to the existing and future land use types, play money token, and water infrastructures token (Fig. 5). We also created, water balance miniatures (Fig. 6) to demonstrate how surface water flows and leads to floods and water infiltration increases ground water supply. Each round after calculating the economic condition and environmental conditions based on Table 3, we asked players to pay production costs, taxes, etc. and get income, incentives, etc. using play money.  The water balance was shown via a miniature with real water according to the produced surface water and groundwater " |
| 4 | In line 119 it is not clear what kind of values represent discharges 5 and 3,5 m$^3$/s (average in the mentioned year, some long-term average, something else?). Please clarify.

**Answer:**

*Thank you for question. The value represent the average throughout a year of the daily discharge. We changed the text line 120 to:*
"Land conversion from agroforestry to intensive agriculture in the recharge areas (>700 masl. upstream and midstream area) and massive groundwater extraction using artesian wells in the downstream area for rice field were thought to cause the reduced average discharge of the Umbulan spring,  from 5  m$^3$/s (1980s) to 3.5 m$^3$/s (2020) " |
| 5 | Table 1 in my opinion is too long/big and similarly as multiple subsections break the reading flow. I would suggest adding some summary into the text and moving the table into appendices.

**Answer:**
*Thank you for your suggestion. We moved table of criteria of credibility, salience and legitimacy to Appendix C, and replaced that table with short summary in the sub-section 2.5 line 253)*

"From the long list of criteria (Belcher et al., 2016), we chose four credibility criteria, five salience criteria and two legitimacy criteria which we considered to be the most relevant for evaluating the H2Ours game. Each of these criteria were measured during the game design process and after the game implementation. We included those criteria during the game design using the ARDI and DPSIR frameworks to diagnose issues in the study area (Section 2.2)." |
| 6 | In line 231, a brief explanation about both methods, i.e. Likert scale and q-method, needs to be added.

**Answer:** |

| | |
|---|---|
| | *Thank you for your input. We decided to focus on the rapid evaluation using Likert scale survey and will elaborate the q-method result in the next publication related to the decision making pattern and player's perceptions. We elaborate the text in section 2.5 line 258*
"A rapid evaluations were conducted after the game session to assess the game session process and the game in achieving its objective. We converted those game performace criteria and creadibility, salience and legitimay criteria into Likert used questions and asked all game participants to fill in the survey. In the Likert survey, we used five-point scales (strongly disagree, disagree, neutral, agree, and strongly agree) on six statements to ask participants about their feeling during the game, their understanding of the rules of the game, the length of the game simulation, new knowledge that they got from the game, and implementation the game to their reality " |
| 7 | In Figure 4, Figure D1, Figure D2, and Figure D3 it is not clear what are presenting solid blue, green, and red lines. Additionally, "ml" in legends should be replaced with "masl". In relation to these figures, why are there different thresholds used (e.g., 200, 800) than explained in lines 114–116?

**Answer:**
*Thank you for your suggestion. The legend is correct with 'ml' because it describes the amount of surface water and ground water. We will make the legend clearer. Please see Fig. 4*
*We added more explanation about the threshold value (e.g. 200, 800) in the section 3.2.3 (line 320) because this is related to the rules of the game, and flooding, water shortages and land fires as outcomes from the calculation of Table 3 and Table D2.*
"When during the rainy season the total of surface water in the downstream area of Rejoso watershed and in the shallow peat of Pawan-Kepulu peatland exceeds its capacity (>800 ml), it caused flooding. When the groundwater exceeds its capacity (>700 ml), the excess water flows to the Umbulan springs in Rejoso watershed and to sea in Pawan-Kepulu peatland. But, when the groundwater was less than <200 ml, it caused water shortages for agriculture in the Rejoso Watershed and made peat soil dry which triggered fires in Pawan-Kepulu peatland. These environmental impacts decreased the overall community income. As the consequence of this situation, the players might not have enough money to manage their land, buy food or pay taxes in the next round of the game. The multi-stakeholder forums with their limited budget could then choose to help them by providing financial help or making regulations/programs to prevent these environmental problems. Through this gameplay, we aimed to stimulate players to collaborate to achieve their goals." |
| 8 | In Figure 7, there are missing y-axis titles. Please add.

**Answer:**
*Thank you for your correction. we revised the Figure 7* |
| 9 | Please use en-dash throughout the manuscript in case of ranges and periods. E.g., 0–100 masl instead 0 – 100 masl.

**Answer:**

*Yes, Thank you. we checked all the manuscript and revised it* |

| 10 | Appendices should be mentioned in text in the order in which they appear, e.g., Appendix A before Appendix B. |
|---|---|
| | **Answer:** |
| | *Thank you for your input. We have change the order of the Appendices* |

**Response to Reviewer 3 (RC3)**

| No | Comments and Answers |
|---|---|
| 1 | Thank you for sharing this interesting work with me. First of all, please let me apologize if I have not properly understood those elements of the paper on which I give comments, or if my perception was wrong. |
| | The paper documents the process of designing and applying a serious game based on a system simulation model. The application of serious games as a contribution to sustainable development receives increasing attention also raising high expectations. The presented work documents a game design process and intends to provide evidence on outcomes from playing the game. It can help to better understand the potentials and limitations of the approach and can meaningfully inform future serious game design processes. I strongly support the publication of the paper. Having said this, I want to share some thoughts which I hope can help to improve the paper. |
| | **Answer:** |
| | *Thank you for your feedback and support* |
| 2 | a. The study has two objectives: |
| | 1. Develop an adaptable serious game |
| | 2. Assess if game facilitates knowledge transfer and sharing, and supports negotiation and coordination. |
| | I am not sure whether the paper really addresses these two objectives. Either rethink your objectives or structure the paper more strongly around the ones you formulated. You documented the game development process which I consider an important contribution. It is difficult for me to judge how easy it is to adapt the developed game to a different context. The paper does not really make this clear. |
| | **Answer:** |
| | *Thank you. We rephrased the two objectives to be closer to what we actually discuss*
*1. Develop a serious game that is adaptable to different socio-hydrological contexts and issues*
*2. Clarify how such game can facilitate knowledge transfer and sharing, and can support negotiation and coordination by context-specific use* |

b. I am even more concerned about the second research question. The methods section does not explain the approach in sufficient detail. An input-output process is mentioned as methodology without explaining it. You state that information was collected throughout the game development and implementation process. How was this information collected and analyzed? You mention a post-game survey. Please provide more information on: 1) who was interviewed and how have respondents been selected, 2) how many respondents were interviewed, 3) how was the survey implemented (also addressing the strong risk of social desirability bias), 4) when was it conducted?

**Answer:**

*We add more information about how we collect the data for the evaluation survey in the section 2.5 about the game evaluation.*

"From the long list of criteria (Belcher et al., 2016), we chose four credibility criteria, five salience criteria and two legitimacy criteria which we considered to be the most relevant for evaluating the H2Ours game. Each of these criteria were measured during the game design process and after the game implementation. We included those criteria during the game design using the ARDI and DPSIR frameworks to diagnose issues in the study area (Section 2.2).

A rapid evaluations were conducted after the game session to assess the game session process and the game in achieving its objective. We converted those game performace criteria and creadibility, salience and legitimay criteria into Likert used questions and asked all game participants to fill in the survey. In the Likert survey, we used five-point scales (strongly disagree, disagree, neutral, agree, and strongly agree) on six statements to ask participants about their feeling during the game, their understanding of the rules of the game, the length of the game simulation, new knowledge that they got from the game, and implementation the game to their reality."

c. For answering the second research question, can you share information on the decisions of players in the game? Do you see patterns and changes over the rounds? Can you share information on what has been discussed during the game sessions?

**Answer:**

*As the objective of this manuscript is to document the development and adaptation process of the H2Ours game, so in this paper we focus on the development process. We plan to make a series of publications about H2Ours game in Rejoso watershed and Pawan-kepulu peatland:*
1. *H2Ours game development and adaptation process (this manuscript)*
2. *Result of the H2Ours game.*

We decided to present the results of the H2Ours game in different manuscript to have more space to present the conditions of the research location which are closely related to how players make decisions regarding land and water management, including changes in perception and knowledge. We mentioned it in section 3.3 (game implementation result): "Further analysis to these different perspectives will be presented in follow-up manuscripts (Tanika et al, in prep)."

| 3 | In this context, I recommend giving more weight to Section 3.3. Interesting statements are made regarding the game dynamics. It could be combined with some statements included in the |

discussion (e.g. that setting up the game has built emotions). The value of the information depends, however, on the transparency of how information was generated.

**Answer:**

*We agree with your statement. In accordance with the objective of this manuscript which focuses on the H2Ours development process, the results shown in this manuscript are more about the form of the H2Ours game itself. Implementation results are presented in section 3.3 is only indicative result. We believe that the dynamics of the game results correspond to the character of the socio-hydrological system and changes in participants' perceptions can be an interesting article which provide more space for explanations. So we will deliver more detailed result in another paper.*

| | |
|---|---|
| 4 | On a conceptual level, I am not convinced that collecting data on credibility, salience/relevance, and legitimacy can answer the second research question. This would at least require stronger justification. Going a step further, I do not think that the survey questions are adequate indicators for Belcher's et al. criteria. For instance, asking a player whether she sees the possibility to apply the knowledge acquired in the game does not provide information on how inclusive the process has been. I would propose a more differentiated approach regarding how you provide evidence on different criteria. Players will not be able to give meaningful answers to some of them. For instance, by playing the game, a player does not need to be aware of the underlying theory of change. I would move Table 1 to an appendix to make space for a better explanation of the methodology and the key results.

**Answer:**

*In the theory of change related to this game, the desired changes refer to the 4 stages of the knowledge to action chain (understanding, commitment, operationalization and innovation). Based on the definitions of credibility, salinity and legitimacy in section 2.5 line 250: "From the game development perspective, credibility refers to whether a game is built based on scientifically reliable knowledge, including the data and methods used to build the game. Salience refers to how far the game can show the relevance of goals, rules and finding to the actual situation. Finally, legitimacy refers to how the participant can accept the game by relating the game simulation to their actual situations (Cash et al. 2002)." In order to achieve the second objective of this paper, to clarify how games can facilitate the sharing and transfer of knowledge, we want to ensure relevance of game situations to the reality (salience), the acceptance that game conditions resemble their conditions (legitimacy), and supported by appropriated data and methods (credibility).*

*Thank you for your suggestion. We moved table of criteria of credibility, salience and legitimacy to Appendix C, and replaced that table with short summary in the sub-section 2.5, please see our response to Reviewer 2 (RC2) number 5.* |
| 5 | A clear theory of change is one of the evaluation criteria and it needs to be expressed in a more explicit way. In lines 42ff, the authors argue that knowledge supports commitment (intention?) supports responsibility, supports innovation. It is not clear which type of knowledge, commitment, responsibility, and innovation the game intends to support. The learning outcomes are formulated rather vaguely. In the model framework, the authors include the assumption that |

income drives decisions. In the discussion, you argue that the enabling environment is the key driver and no behavioral change can be expected if regulations etc. are not adjusted. Considering all these thoughts and assumptions, what did you intend to change by letting stakeholders play the game?

**Answer:**
*Thank you for your concern. We agree that we have not written down specifically what form of knowledge we want to convey in this form of game. We revise the sentence in section 4.2 (line 482) to:*

"As hydrological problems are usually complex and fundamental, any potential solution requires ample time for integrated planning, and all relevant stakeholders to understand the dynamics of the system at large scale (Medema et al., 2019). The H$_2$Ours game tries to present simple representation of the landscape so that it makes it easier for players to be aware of the conditions of neighbouring players and to gain system level perspective of socio-hydrological issues. Improving player knowledge by looking at socio-hydrological problems in a broader context encourages responsible behaviour towards the environment which is directly proportional to commitment (Keles et al., 2023)..

*We also revised the third lesson learned to reduce miss interpretation (line 503):*

"Third, based on the evaluation and debriefing results, even if they stated they can apply the ideal collaborative actions that were explored in the game session, in real life, the enabling conditions  needed to support this still required to be build (e.g. regulation, integrated planning strategies, etc.). As the game is a simplification of the real-life system, forms of collaborative action can be discussed directly by the players. In real life, the parties that are needed for successful collaboration may not easily meet each other to discuss issues openly. Therefore, it is necessary to create a condition where stakeholders can meet and explore collaboration options to jointly address issues and achieve goals. Without such encounters, the commitment that referred to in the four knowledge-to-action chains cannot be attained"

| 5 | Based on the provided information, it is difficult to understand the structure of the game. Key aspects are not explained, such as:

Number of players per session; who was selected to play the game (relevant also for theory of change)

Types of roles; in section 3.2.2 is described that there are two roles namely communities and a multi-stakeholder forum. But then one finds statements that there are upstream and downstream actors, different villages etc. Is the role definition linked to the diagnosis? Who in Table 2 creates pressure? Who is involved in the responses and dynamics? Are the relevant actors included as roles? Such game development decisions are critical for the theory of change.

**Answer:**

*Thank you for your concern. We elaborated Section 2.4 (Game implementation) and added more information about the players in this study. Please see our response to RC1 number 2c*

We added more explanation on section 3.2.2 (roles): |

"Based on the stakeholder identification survey in Rejoso Watershed and Pawan-Kepulu peatland, we defined two key roles for this game, namely a multi-stakeholder forum and local (or farmer) communities. The goal of the multi-stakeholder forum is to prevent natural disasters meaning water scarcity and floods in Rejoso watershed, and fires and floods in Pawan-Kepulu peatland. In the Rejoso watershed, local communities can be grouped into people who live in the upstream village, midstream village and downstream village based on the village elevation. Meanwhile in Pawan-Kepulu peatland, local communities can be grouped into four groups of people living in four neighbouring villages (Village 1 – Village 4). Local communities represent landowners. Their goal is to fulfil their household needs (food and taxes). The $H_2$Ours game brings the various interests of these actors together and shows how they make their decisions regarding the management of land and water resources to meet their economic and environmental expectations."

*In the section 2.3.2, we already mentioned that how we defined the roles based on the ARDI framework:*

"According to the ARDI framework (Sect. 2.2), we defined the roles based on the main stakeholders involved in water management in each study area. Related to these roles, we designed goals that players must achieve during each simulation based on discussions and interviews with the related stakeholders according to their actual goal".

*We revised section 3.1 (result: Diagnosis of the study areas and issues) based on your input:*

"Based on the results from the DPSIR and ARDI analyses, we found that the Rejoso Watershed and the Pawan-Kepulu peatland have similarities in the socio-hydrology context (Table 1). Expectations on better economic conditions led local communities to changes in land cover, and excessive extraction of water resources (groundwater) caused disruption of the water balance. This disruption resulted in local communities and multi-stakeholder forum experience various hydrological problems, such as water shortages (or decreasing the groundwater level) and flooding. However, these two sites are also different regarding their hydrological contexts, such as hydrological boundaries, topography, and water management, and interactions among stakeholders and landscape (Fig. 3, Fig. D1). Two proposed solutions (responses) were identified by ICRAF and Tropenbos Indonesia based on their research findings to restore hydrological functions in watersheds and peatlands, namely better land use management and (ground) water management (Table 1; component 7-Response)."

| 6 | a. Which decisions can be taken by which player role? Table 3 and Figure 3 give the impression that only crop and forestry related choices can be made. But then it is mentioned that players can block channels. The choices of the multi-stakeholder forum are formulated very unclear and are not included in Figure 3. |
|---|---|

How could the forum make regulations or programs to prevent environmental programs.

**Answer:**

*Thank you for your concern. We added a paragraph in section 3.1 (p.2) related to Figure 3 and Figure D1 to explain what the types of decisions they make that affect their landscape.*

"The interaction between stakeholders and the landscape is represented by the type of decisions regarding their landscape taken by the multi-stakeholder forum and local communities. Local communities (farmer from upstream, midstream, and downstream village in Rejoso Watershed and farmers from neighbouring villages: Village 1-Village 4 in Pawan-Kepulu peatland) have the authority to make decisions regarding their land which consists of land-use types and water management types (artesian wells in Rejoso watershed and canal blocking in Pawan-Kepulu peatland). Multi-stakeholder forums have authority over regulations and programs applied to local communities to achieve their goals. Multi-stakeholder forum can refer to their existing or potential regulation and program"

*We revised Section 3.2.3. (Rules) to give more explanation about the option of land use types and options of regulation and program by multi-stakeholder forum.*

"At the start of the game, players (i.e. multi-stakeholder forum or local communities) received a limited amount of play money. Community members were asked to manage their land to meet their household needs by arranging the land-use type combination and water management in their area with the play money provided, while multi-stakeholder forum was asked to run programs or to help reduce the local community's financial problems. Once players decided on how they would manage their land or community programs, the economic and environmental rules linked to those land-use decisions were applied (Table 2). These rules then defined the dynamics of the economic and environmental conditions (Table 2, and Table D1 and D2 for the Pawan-Kepulu peatland). "

*b.* Please also better explain how you decided on the players' choices in the game.

**Answer:**

*We invite the players through village leaders and farmer group leaders, and let them decided who would attend the simulation. However, in the invitation we gave the criteria of the participants. We already added this information in the section 2.4 (game implementation). Please see our response to reviewer 1 (RC1) number 2c..*

c. In the discussion is mentioned that the game could not capture the complexity of the system and that few issues had to be selected to be included in the game. Please describe the process from diagnosis to selecting the issues to be included in the game. Such game development decisions are again critical for the theory of change.

**Answer:**

*In the discussion we mentioned that based on several publications, these two research studies have more complex socio-hydrological problems than played in the H2Ours game. In the beginning, we wanted to include all problems in one game such as erosion, landslides and sedimentation. However, during the game development process, especially in the preparation of rules and the quantification systems, the game becomes very complex. Considering the background of the targeted participants and the game duration, we decided to only highlight one hydrological problem. We determined which issues to raise based on the national priority issue in that location.*

*We add more information on how we chose the issue for the game in the discussion. Please see line 447:*

"The two study sites experience more complex socio-hydrological problems than represented in the H$_2$Ours game. In our game, the water quantity issues were represented in line with national priority issues in that location, which resulted in groundwater scarcity and floods for Rejoso, (Fig. 3) and fire and floods for Pawan-Kepulu peatland (Fig. D1)."

d. Are decisions taken collectively (a group represents one role and decides based on some agreement) or individually, creating collective outcomes. In line 433 is stated that the game demonstrated the value of collective action. How is collective action implemented in the game?

**Answer:**

*Yes, a group represent one role. When we started the game, the facilitator asked to choose a leader to represent their group.*

e. Did the players receive real money? If yes, how was the payout determined? Did it depend on the game dynamics as is common in behavioral games? If not, make clear in Section 3.2.3 that you talk about a game endowment/play money.

**Answer:**

*I am sorry, i do not really understand your question. I assume your question is about the use of real or play money and how to determine it. We revised section 3.2.5 (game properties) to explain about the play money and water balance miniature.*

"To make the game engaging, we prepared game materials such as a game board to represent the landscape, land-use tiles according to the existing and future land use types, play money token, and water infrastructures token (Fig. 5). We also created, water balance miniatures (Fig. 6) to demonstrate how surface water flows and leads to floods and water infiltration increases ground water supply. Each round after calculating the economic condition and environmental conditions based on Table 3, we asked players to pay production costs, taxes, etc. and get income, incentives, etc. using play money. The water balance was shown via a miniature with real water according to the produced surface water and groundwater."
* * *
a. A more comprehensive explanation of outcomes resulting from choices would be important for understanding the game. For instance, in Table 3, can you provide likelihoods of flooding, water shortage, and fire events for different land use options?

**Answer:**

*We add more explanation in Section 3.2.3 explaining how we determined environmental impacts (floods, fires and water shortage).*

"When during the rainy season the total of surface water in the downstream area of Rejoso watershed and in the shallow peat of Pawan-Kepulu peatland exceeds its capacity (>800 ml), it caused flooding. When the groundwater exceeds its capacity (>700 ml), the excess water flows to the Umbulan springs in Rejoso watershed and to sea in Pawan-Kepulu peatland. But, when the groundwater was less than <200 ml, it caused water shortages for agriculture in the Rejoso

Watershed and made peat soil dry which triggered fires in Pawan-Kepulu peatland. These environmental impacts decreased the overall community income. As the consequence of this situation, the players might not have enough money to manage their land, buy food or pay taxes in the next round of the game. The multi-stakeholder forums with their limited budget could then choose to help them by providing financial help or making regulations/programs to prevent these environmental problems. Through this gameplay, we aimed to stimulate players to collaborate to achieve their goals."

b. The choices in Table 3 indicate that unsustainable options provide highest income. At the same time, your model (Figure 3) works with the assumption that income mainly drives choices. How do you address the risk of unsustainable learning outcomes? A player's answer in the evaluation survey that she can apply the game learnings could mean that she learnt that an upstream-all-crop strategy is best.

**Answer:**

*I understand your question and concern. When we talk about the best scenario at the landscape level involving many stakeholders, there may be a conflict between the best scenario for whom and for what. Hence, the H2Ours game tries to convey this issue through one of its objectives related to a shared understanding by bringing the socio-hydrological system as one system. By understanding condition, we expect that a commitment will emerge to overcome their common problems through negotiation and coordination between stakeholders. This game was designed to resemble actual conditions, so it could provide an illustration of the solution to their real problems. However, to achieve sustainable learning outcome, other efforts are still needed such as implementation (adapting what they learnt from game simulation to real life), monitoring and evaluation.*

| 8 | I do not understand the step of the game solution space analysis. What does it mean that the solution space was explored based on 3, 10, 30, 100, 300, and 1000 games with random choice? How did these 1000 games differ? Just explain this better for readers not familiar with your modelling and game development approach.

**Answer:**

*We elaborate on section 2.3.4 to explain in more detail about the solution space and the method used to generate it. Please see our response to Reviewer 2 (RC2) number 2* |
|---|---|
| 9 | Line 74f: I challenge the statement that models and games have rarely been combined. There are numerous examples for instance on the website https://games4sustainability.org/gamepedia/. I am also aware of the following study: Lohmann, D., Falk, T., Geissler, K., Blaum, N., & Jeltsch, F. (2014). Determinants of semi-arid rangeland management in a land reform setting in

**Answer:**

*Thank you – our text was indeed inaccurate here. We corrected to line 73:*

*"Socially interactive games and models that explore larger spatial and temporal horizons have complementary strengths. As reviewed in Villamor et al., (2023), games and models can 1) seek a* |

| | |
|---|---|
| | conceptual triangulation of representing the processes behind complex realties, 2) strive for numerical consistency between games and empirical models, 3) use games in the development of scenario models, or 4) use models in the design of games that trigger players to learn by experiencing manageable complexity. As an example of the letter, Lohmann et al. (2014) designed and tested model-based role plays with Namibian land reform beneficiaries, simulating 10 years of rangeland management. In this paper, we explore the feasibility of transforming a hydrological model into a serious game to provide socio-hydrological dynamics to stakeholders with diverse backgrounds to develop restoration plans." |
| 10 | Line 80f: As a complementary approach, the authors may want to consider providing guidance on context conditions which need to be fulfilled for meaningfully using the games. Also, can you estimate the size of the areas which share socio-hydrological conditions which are featured in the presented games?

**Answer:**

*The current games rely on hydrological units without major unrecognized in- or outflows. In many cases groundwater flows into or out of agriculturally used areas may be relevant uncertainties in the use of water balance models. In both the mountain-to-sea and peatland hydrological units groundwater flow issues are internalized in the game. In many landscapes the social systems don't match the hydrological boundaries; spatially explicit social subsystems, such as upper, middle and lower watersheds can be accommodated, while interactions with social systems beyond the game boundaries may be represented by economic (supply, demand, prices) parameters and their (exogenous) fluctuations.* |
| 11 | Line 185f: On which basis do you assume that decisions of players in the game represent their real-life decisions? Did you ask them to do so? Why do you need this assumption?

**Answer:**

*Sari et al. (2023) tested for the plot-level FORCES game whether farmers land use system and tree choices matched their actual conditions outside the game. For H2OURS we don't have the same detailed comparison yet. However, during the game simulation process we see whether players play according to their actual conditions based on the choice of land cover they choose or not. We compared how they selected land use during the game simulation with the actual land use. Part of the game properties, we provide land use types according to their actual conditions and some other land uses that may be options in the future. We did not directly ask them to play according to reality, but in the beginning we created conditions in accordance with their reality. In this study, we expected participants to play according to their reality at the beginning because one of the goals of this game is to facilitate negotiation and collaboration between stakeholders to achieve the hydrological restoration target.* |
| 12 | Can you please better link your discussion and conclusions to the objectives of the paper and the presented results. |

| | |
|---|---|
| | **Answer:**

*Thank you for your input. We will look again the link between objectives, discussion and conclusion* |
| 13 | Now that you have developed the game, can you share some thoughts on how your games could be applied to support sustainable development? Who could facilitate games with which target groups? What would be required to implement the game with a larger group of stakeholders? Ideally this way forward should be linked to the theory of change.

**Answer:**

*My research has a significant correlation to SDG 6 (Clean water and sanitation), SDC 13 (climate action) and SDG 15 (Life on land), but also supports SDG 16 (Pace, Justice and strong institutions) and SDG 8 (Decent work and economic growth). The complexity of water resource management arises from many stakeholders who have different backgrounds, interests, targets and knowledge have to make decisions related to the same system. Because they do not know that they are working in the same system, they do not know that their decisions might affect other stakeholders. Therefore, changes in point of view of stakeholders to realize that they are actually working in the same socio-ecological system are needed. H2Ours game simplifies a landscape that has the same hydrological system, allows stakeholders to see the same system and makes it easier for them to collaborate to solve problems or achieve common goals.*

*For broad stakeholder groups, it requires identification of key stakeholders who are participants in the player. In the game development process, we mentioned in the roles of the game. The facilitator's ability and knowledge in leading the game also plays an important role in briefing, implementation and debriefing* |
| 14 | I hope my comments help to improve the paper. I think my concerns can be addressed in the frame of a major revision. I very much hope to see an improved version of the paper being published.

Answer:

Your input is very useful and valuable for us. We really appreciate your time for this manuscript. |

Reference:

Belcher, B. M., Rasmussen, K. E., Kemshaw, M. R., and Zornes, D. A.: Defining and assessing research quality in a transdisciplinary context, Res. Eval., 25, 1–17, https://doi.org/10.1093/reseval/rvv025, 2016.

Cash, D., Clark, W. C., Alcock, F., Dickson, N., Eckley, N., and J�ger, J.: Salience, Credibility, Legitimacy and Boundaries: Linking Research, Assessment and Decision Making, SSRN Electron. J., https://doi.org/10.2139/ssrn.372280, 2002.

Jeanes, K., van Noordwijk, M., Joshi, L., Widayati, A., Farida, and Leimona, B.: Rapid Hydrological Appraisal in the Context of Environmental Service Rewards, World Agroforestry, Bogor, 56 pp., 2006.

Keles, H., Yayla, O., Tarinc, A., and Keles, A.: The Effect of Environmental Management Practices and Knowledge in Strengthening Responsible Behavior: The Moderator Role of Environmental Commitment, Sustain., 15, https://doi.org/10.3390/su15021398, 2023.

Leimona, B., Khasanah, N., Lusiana, B., and ...: A business case: co-investing for ecosystem service provisions and local livelihoods in Rejoso watershed, 2018.

Medema, W., Mayer, I., Adamowski, J., Wals, A. E. J., and Chew, C.: The potential of serious games to solve water problems: Editorial to the special issue on game-based approaches to sustainable water governance, Water (Switzerland), 11, https://doi.org/10.3390/w11122562, 2019.

van Noordwijk, M., Lusiana, B., Leimona, B., Dewi, S., and Wulandari, D.: Negotiation-support toolkit for learning landscapes, edited by: van Noordwijk, M., Lusiana, B., Leimona, B., Dewi, S., and Wulandari, D., World Agroforestry Southeast Asia Regional Program, 2013.

van Noordwijk, M., Speelman, E., Hofstede, G. J., Farida, A., Kimbowa, G., Geraud, G., Assogba, C., Best, L., and Tanika, L.: Sustainable Agroforestry Landscape Management :, Land, 9, 1–38, 2020.

 Rodela, R and Speelman, E.N., 2023, Serious game in natural resource management: steps toward assessment of their contextualized impacts, Current Opinion in Environmental Sustainability (under review process)

van Noordwijk, M., Speelman, E., Hofstede, G. J., Farida, A., Kimbowa, G., Geraud, G., Assogba, C., Best, L., and Tanika, L.: Sustainable Agroforestry Landscape Management :, Land, 9, 1–38, 2020.

---

## Author Response (AR2)

Dear Editor and Reviewer,

Thank you again for taking your time to review the revised version of our manuscripts. We try our best to address input and comments from the reviewer point to point. Below is our response to those input. In this response, we use the blue text to indicate our changes from the original text.

Thank you

| 1 | You need to rethink your second objective (Assess if game facilitates knowledge transfer and sharing and supports negotiation and Coordination). The way the second objective is currently addressed is superficial and not transparent. In your response to us you write: 'the objective of this manuscript is to document the development and adaptation process of the H2Ours game(…)'. If you want to save your data for another manuscript, then drop or revise your second objective. |
|---|---|
| | Answer: |
| | Thank you for your concern about the objective of this manuscript. Because your first and second comments are very connected, please allow us to response your first comment together with the second comment. |
| | |
| 2 | I disagree conceptually with 'In order to achieve the second objective of this paper, to clarify how games can facilitate the sharing and transfer of knowledge, we want to ensure relevance of game situations to the reality (salience), the acceptance that game conditions resemble their conditions (legitimacy), and supported by appropriated data and methods (credibility).'. Salience, legitimacy, and credibility of games are not valid measurements of knowledge transfer and sharing. You could fix many problems in your paper if you formulated the second objective as 'Assess the salience, legitimacy, and credibility of the H2Ours game.'? You are interested in ensuring salience, legitimacy, and credibility because you assume these are preconditions to allow the game to transfer and share knowledge, and to support negotiation and coordination. You can explain this. But at least the data you present to not allow to assess game outcomes. |
| | Answer: |
| | Thank you for providing us with very strong and reasonable reasons regarding the second objective and the use of credibility, salience and legitimacy in this study. If we referred to Cash 2002, 2020 and Belcher 2016, the use of credibility, salience and legitimacy in research is to assess the sustainability, practicality and transferability of the research to actions. You have captured our idea in using these criteria in the game is to assess whether the game simulation is useful for the players, especially to lead to the objectives of the game. Now we agree that, to assess whether the game fulfills its objective as a tool that facilitates knowledge transfer and coordination, we need to provide the evidences how the game change their, which we cannot include here because it makes this paper very complex. So we decided to follow your suggestion to change the second objective to assess the quality if the game in term of credibility, salience and legitimacy criteria. Due to the revision of the second objective of this manuscript, we revised several sections related to game evaluation. |
| | Introduction line 86: |

"Therefore, the objectives of this study are to develop a serious game that is adaptable to different socio-hydrological contexts and issues, and to evaluate the quality of the game in terms of credibility, salience and legitimacy. To achieve our objectives we developed a generic game with two adaptations to two different locations in Indonesia differing largely in hydrological characteristics. First, we developed the $H_2Ours$ game based on the socio-hydrological characteristics of the Rejoso watershed in East Java. Then, we modified the $H_2Ours$ game according to the conditions of the Pawan-Kepulu peatland, West Kalimantan. The qualities of the game were assessed based on several criteria representing credibility, salience and legitimacy which were included in the game development process and post-game assessment. We organized the paper by presenting as method the stages of how we prepared, designed, tested, implemented and evaluated the $H_2Ours$ games. The game itself is the primary 'result', illustrated by the game dynamics during test settings and early applications with local stakeholders. Feedback by game participants is presented as an evaluation of the current games. We close by discussing the simplification process from reality to game, effectiveness of the game to achieve the goals set, and the lessons learned"

We simplified the whole section of 2.5 about the explanation of game evaluation to adjust the terminology of credibility, salience and legitimacy:
"The aim of the evaluation stage is to assess the game session process and the quality of the game as the basis for the game's performance to fulfil its objectives.. The game session process was evaluated based on game performances criteria in the form of rules that can be understood, fun and playability over time. While the quality of the game is assessed based on the scientific logic and reliable knowledge used to build the game (credibility), its relevance to the societal issues (salience) and the acceptance by the game participants (legitimacy) (Cash et al., 2002; van Voorn et al., 2016). For the effectiveness of the assessment, we followed input-output assessment process, which evaluated the input used in the game during development process and the output after the game session (Bedwell et al., 2012). We followed the latter approach and carried out the evaluation based on several criteria that refer to credibility, salience, and legitimacy (Table C1 in Appendix C), using some criteria developed by Belcher et al. (2016)
Because Belcher's long list of criteria (Belcher et al., 2016) originally was used to assess the quality of research, for this study we chose several criteria that were relevant to game quality. Each of these criteria were measured during the game design process and after the game implementation. We measured these criteria by how it was associated with the condition and diagnosis of the study area (Section 2.1 and 2.2) game development process (Section 2.3). Please see Table C1 to see the parameters and sections associated with each criteria. A rapid evaluations were conducted after the game session to assess the process and the quality of game session. We converted those game performace criteria and creadibility, salience and legitimay criteria into Likert used questions and asked all game participants to fill in the survey. In the Likert survey, we used five-point scales (strongly disagree, disagree, neutral, agree, and strongly agree) on six statements to ask participants about their feeling during the game, their understanding of the rules of the game, the length of the game simulation, new knowledge that they got from the game, and implementation the game to their reality"

Section 4.2 Discussion we add a paragraph line 495 to explain that in this manuscript we only assessed the quality of the game through credibility, salience and legitimacy as the basis for evaluating the game to meet its objective.

"We limit the evaluation in this study only to the quality of the game as a product. As a serious game, the H$_2$Ours carries certain goals that it wants to fulfil (Rodela et al., 2019), namely as a tool that can facilitate the transfer and sharing of knowledge from its players to support the coordination and negotiation process (Section 3.2.1). Evaluating the game in fulfilling its objectives is more complicated than evaluating the game session process. Ideally, the evaluation of the game in achieving its objective can be evaluated after several simulations at various levels of simulation, and should be conducted before, during and after the game sessions (Oprins et al., 2015). The evaluation of the game to meet the objective will be carried out in the next manuscript by providing evidence of changes in participant's perceptions".

| 2 | If section 3.3 provides only indicative results, you still need to explain their data basis in the methods section – even if it is a subjective perception of game designers and facilitators. |

Answer:
The data used in Section 3.3 consist of random-walks which are part of the solution space and the results of the game sessions. We have not explained the results that come from the game session, therefore we added a paragraph in Section 2.4 (line 247) to provide explanation regarding the data used in Section 3.3

"The game explores the trade-off space between economic and environmental outcomes, with the responses from players during the debriefing adding further insights. The economic and environmental outcomes was calculated based on the average economic and environmental conditions as a result of decision making regarding land use combinations during a game simulation over 10 rounds. We present these results together with the results of the solution space analysis to show the position of players' decisions compared to random decision-making. During the debriefing, we asked participants several questions such as whether they enjoyed the game, what knowledge they gained from the game, how they responded to government regulations of the type included in the game, how they felt seeing other group decisions and (for study case Pawan-Kepulu peatland) their strategies as a member of multi-stakeholder forum"

| 3 | Line 255ff: Please explain in the methods section how measurements during game design and thereafter were done. I assume that these measurements are different from the rapid evaluations. |

Answer:
Thank you for this input. We clarified how we follow these criteria in the game design process by explaining in line 267

"Because Belcher's long list of criteria (Belcher et al., 2016) originally was used to assess the quality of research, for this study we chose several criteria that were relevant to game quality. Each of these criteria were measured during the game design process and after the game implementation. We measured these criteria by how it was associated with the condition and diagnosis of the study area (Section 2.1 and 2.2) and game development process (Section 2.3). Please see Table C1 to see the parameters and sections associated with each criteria. A rapid evaluations were conducted after the game session to assess the game session process and the game in achieving its objective. We converted those game performace criteria and creadibility, salience and legitimay criteria into Likert used questions and asked all game participants to fill in the survey. In the Likert survey, we used five-point scales (strongly disagree, disagree,

| | |
|---|---|
| | neutral, agree, and strongly agree) on six statements to ask participants about their feeling during the game, their understanding of the rules of the game, the length of the game simulation, new knowledge that they got from the game, and implementation the game to their reality". |
| 4 | I did not find a revision in the text related to the clarification that a group represented one role and that a leader was asked to be chosen. This should be included in the paper as it is important to understand the game. Games with group decisions have very different dynamics than games with individual decisions.

Answer:
Thank you for your input. We clarified the Roles in Section 2.3.2:

"According to the ARDI framework (Sect. 2.2), we defined the roles based on the main stakeholders involved in water management in each study area. Most of the players were asked to be a villager, representing the largest stakeholder group, but others had specific roles as agents trying to influence villager decisions. Related to these roles, we designed goals that players must achieve during each simulation based on discussions and interviews with the related stakeholders according to their actual goal. Before the game started, we asked each group to choose a leader to facilitate discussion within the internal team and represent the group in communicating with other groups." |
| 5 | For me as an economist the process of generating the solution space based on 3, 10, 30, 100, 300, and 1000 games with random choice is still not clear. I understand that random choices mean random parameter setting. Using the word 'choice' gives the impression that somebody (human) takes a decision. But why 3, 10, 30, 100, 300, and 1000 games? Maybe explain this whole process clearly in an appendix.

Answer:
Thank you for your input. We decided to replace 'choice' with 'random-walk' to provide more general meaning because the random parameters refer to the random climate conditions and random land use combinations. Why we chose 3, 10, 30, 100, 300 and 1000, we added explanation in Section 3.2.4.

We revised Section 2.3.4 regarding game solution space analysis as follows:
"The purpose of game solution space is to define the envelope of possible outcomes within the rules of the game, considering all possible choices made by players in the game (Speelman et al., 2014). In a random-walk any sequence of steps has equal probability, blind to where it may lead. The solution space of the $H_2$Ours game was explored based on the average of economic and environmental outcomes obtained with a random-generator deciding choices for every step. We mapped the estimated solution space after 3, 10, 30, 100, 300 and 1000 random-walk iterations to obtain a reference for the trajectories observed in a limited number of actual, real-player games. The random-walk conditions were generated in R, then simulated using an Excel spreadsheet representation of the $H_2$Ours game and its economic and environmental performance indicators. The 1000 random-walk data set was used to assess the probability density of outcomes within the solution space. The economic and environmental performance indicators of actual game implementation refer to player's land use decisions from four |

different game session in Rejoso Watershed which are calculated using the same Excel spreadsheet"

And Section 3.2.4, result of solution space analysis:
"From the comparison results between 3, 10, 30, 100, 300 and 1000 random-walk iterations, we found that the shape and distribution of economic and environmental outcomes began to stabilize at 300 iterations. Therefore we used 300 games with random conditions as the basis for the solution space of this research. As reference for the player-based game runs, in 300 game runs with a random decision making process, the groundwater distribution varied depending on the location, while the distribution of surface water in the upstream and midstream is almost the same, and in the downstream is wider (Fig. 4A and Fig. 4B). Upstream and midstream had almost the same frequency distribution of surface water flows while runoff from the upstream and midstream areas was dominated by wet years, which then may potentially cause flooding downstream in the same year. Contributions of groundwater from upstream and midstream also responded to wet years, but groundwater utilization by downstream occurs mostly during the dry years. Therefore, the frequency distribution of groundwater contributions were wider than those for surface water".

| 6 | Line 77: I did not mean to ask you to cite our paper. I was just using it as an example. If you do not make the problematic statement, you do not need to include any citation.

 Answer:
 Surface No, we think we have made an inaccurate interpretation regarding the lack of explicit combinations of models and games. Therefore, in the revised version we revised it by providing an explanation of how the model and game can complement each other and including your publication as part of the example. |
| 7 | Line 203ff: There is a large body of literature on external validity of experiments and the discrepancy between stated and revealed behavior. The evidence is too ambiguous to support a general assumption that players in a game take the decisions they take in real life. They may experiment, show social desirable behavior, or want to enact a strategy for various reasons. You should share results if you find a correlation between land use decisions in the game and actual land use. Otherwise, I recommend avoiding making such claims also because it is not at all that important for learning games. Knowledge transfer and sharing, negotiation and coordination can all happen even if players do not reveal their real-life behavior in the game. Important is that the choices in the game reflect critical choices people have in real life.

 Answer:

 We designed game properties in such a way that it resemble reality in so that participants can imagine and treat the landscape in the H2Ours game as their landscape. I agree that decisions during the game do not all reflect their decisions in real life. But at least they can correlate the impact of their decisions with the impact they experienced during the game simulation with the impact they might experience in real conditions with similar decision. We revised line "Because we expected the decisions made by the participants during the game simulation represented their actual decisions, we developed the game as close to the reality as possible" to prevent ambiguity into (Section 2.3.5, line 210): |

| | |
|---|---|
| | The purpose of game development is to bring the game design into a real form that players can play or touch such as a game board, various required tokens, and other attributes that support the simulation of the game. We developed the game to be close to the perceived reality, so that players can relate their decisions with the consequences obtained during the game session with the impacts that they have experienced or will experience with the similar decisions. The game board, the game's land-use options, and water simulation miniature are the key elements of recognition for players. Therefore, we adapted these elements to the conditions of each study area. |
| 8 | Given the structure of your game, I would still appreciate a more self-critical reflection on possible unintended undesirable learning outcomes. I understand the good intentions of the team. Discussing this risk may be valuable for game designers in future. How could the risk be mitigated?

Answer:
Thank you for the suggestion. Based on game simulation in Rejoso and Pawan-Kepulu peatland, we realized that lack of social value that encourages continuity of an activity. We include the following paragraph in Section 4.2 (line 532) as a critical lesson learned:

"The $H_2Ours$ game clearly shows the trade-off between the economy and the environment by calculating economic and environmental performance indicators in each round after the players change the land use combination and water management. As a result, the relational value between humans and human with nature (e.g. trees and water being inherited from their predecessors and will be a legacy for their descendants, the use of certain woods in religious rituals) sometime becomes blurred. A very clear trade-off between the economic and environmental conditions have led players to make decisions based solely on economic value. Therefore, the cost-benefit calculation of conservation activities needs to be done carefully in this game or include social values as part of the scenario in the game." |
| 9 | I would still appreciate some more concrete thoughts on how this game could be used beyond this project. Who could facilitate the game? Who could be target groups? I understand your higher-level ambitions but how can you imagine a concrete use of the game?

Answer:

Thank you for your concern to this issue. Further evidence on the potential adaptation of the H2Ours game to other contexts was recently obtained (Khasanah, Pers. Comm. March 2024) by a World Agroforesty (ICRAF) project in East Nusa Tenggara (NTT) in Indonesia with pastoral land use and shallow groundwater conditions. The primary hydrological issues in that area are groundwater scarcity and drought. In the adaptation process of the H2Ours game, ICRAF adjusted the ARDi-DPSIR table from this study according to the local issues and recommended water-landscape management in NTT. They played the H2Ours game with the local communities and multi-stakeholder forum to simulate the land use management in order to conserve springs to secure their water. We did not include this information in the manuscript because we need a consent from ICRAF to include this information in the publications. |